# Tight junction protein LSR is a host defense factor against SARS-CoV-2 infection in the small intestine

Yanan An [1,2,3,13], Chao Wang [4,5,13], Ziqi Wang[1,2,3,13], Feng Kong[6,7,13], Hao Liu[1,2,3], Min Jiang [8], Ti Liu[9], Shu Zhang[9], Kaige Du[9], Liang Yin [6,7], Peng Jiao[5,10], Ying Li[1,2,3], Baozhen Fan[11], Chengjun Zhou[12], Mingxia Wang[1,2,3], Hui Sun[1,2,3], Jie Lei [9✉], Shengtian Zhao [5,6,10✉] & Yongfeng Gong [1,2,3✉]

## Abstract

The identification of host factors with antiviral potential is important for developing effective prevention and therapeutic strategies against SARS-CoV-2 infection. Here, by using immortalized cell lines, intestinal organoids, ex vivo intestinal tissues and humanized ACE2 mouse model as proof-of-principle systems, we have identified lipolysis-stimulated lipoprotein receptor (LSR) as a crucial host defense factor against SARS-CoV-2 infection in the small intestine. Loss of endogenous LSR enhances ACE2-dependent infection by SARS-CoV-2 Spike (S) protein-pseudotyped virus and authentic SARS-CoV-2 virus, and exogenous administration of LSR protects against viral infection. Mechanistically, LSR interacts with ACE2 both in *cis* and in *trans*, preventing its binding to S protein, and thus inhibiting viral entry and S protein-mediated cell–cell fusion. Finally, a small LSR-derived peptide blocks S protein binding to the ACE2 receptor in vitro. These results identify both a previously unknown function for LSR in antiviral host defense against SARS-CoV-2, with potential implications for peptide-based pan-variant therapeutic interventions.

**Keywords** ACE2; Antiviral; Host Factor; LSR; SARS-CoV-2
**Subject Categories** Cell Adhesion, Polarity & Cytoskeleton; Microbiology, Virology & Host Pathogen Interaction

## Introduction

The Spike proteins of both SARS-CoV and SARS-CoV-2 utilize the angiotensin-converting enzyme 2 (ACE2) receptor to enter human target cells (Jackson et al, 2022). Given that binding of Spike to the host receptor ACE2 is the key step in SARS-CoV-2 viral entry and the later clinical pathologies, interfering with this binding could provide a universal strategy to prevent and treat SARS-CoV-2 infections, applicable to all SARS-CoV-2 variants and other emerging coronaviruses exploiting ACE2 as their cellular receptor (Oudit et al, 2023). Hence, most vaccines and neutralizing antibodies aim to abrogate this interaction. Neutralizing antibodies, primarily target the trimeric Spike glycoproteins, are hindered by problems related to their solubility, unsuitability for oral or inhaled administration, immunogenicity, and mutational escape (Bojadzic et al, 2021). Therefore, therapies that modulate viral Spike–ACE2 interaction by a host-cell protein, whose expression is not likely to be affected by mutations in the virus, may be effective against all current and future SARS-CoV-2 variants (Majdoul and Compton, 2022). Further, ACE2 is a common receptor for multiple coronaviruses, such as SARS-CoV and HCoV-NL63 (Brevini et al, 2023). Confirmation of the efficacy of this strategy may therefore provide a quickly deployable intervention in the event of future coronavirus outbreaks (Brevini et al, 2023).

Here we demonstrate that the lipolysis-stimulated lipoprotein receptor (LSR) reduces the interaction between Spike and ACE2 and viral entry using cell lines, intestinal organoids, ex vivo intestinal tissues, and humanized ACE2 mouse model as proof-of-principle systems. Genetic inactivation of LSR has significantly higher viral loads in intestinal tissue following SARS-CoV-2 infection as well as pseudo viruses carrying SARS-CoV-2 Spike, and this change was accompanied by enhanced expression of multiple pro-inflammatory cytokines within intestinal tissue and increased recruitment of leukocytes to the intestine. Exogenous administration of LSR protects against viral infection. Using several complementary approaches, we find that LSR binds to ACE2 through its extracellular C-terminal region, and that this decreases the ability of ACE2 to productively interact with Spike protein. Subsequently, we demonstrate that inhibition of viral Spike-ACE2 interaction using a peptide corresponding to the region of the first cysteine rich domain (CRD1) directly adjacent to the transmembrane domain of LSR reduces SARS-CoV-2 infection in vitro,

[1]Department of Physiology, Binzhou Medical University, Yantai, Shandong, China. [2]Shandong Engineering Research Center of Molecular Medicine for Renal Diseases, Yantai, Shandong, China. [3]Laboratory of Tight Junction, Binzhou Medical University, Yantai, Shandong, China. [4]Department of Urology, The Second Hospital, Cheeloo College of Medicine, Shandong University, Jinan, Shandong, China. [5]Department of Urology, Qilu Hospital, Cheeloo College of Medicine, Shandong University, Jinan, Shandong, China. [6]Shandong Provincial Engineering Laboratory of Urologic Tissue Reconstruction, Jinan, Shandong, China. [7]Department of Central Laboratory, Shandong Provincial Hospital Affiliated to Shandong First Medical University, Jinan, Shandong, China. [8]Department of Pharmacology, Binzhou Medical University, Yantai, Shandong, China. [9]Shandong Provincial Center for Disease Control and Prevention, Jinan, Shandong, China. [10]Department of Urology, Binzhou Medical University Hospital, Binzhou, Shandong, China. [11]Department of Urology, Yantai Affiliated Hospital of Binzhou Medical University, Yantai, Shandong, China. [12]Department of Pathology, The Second Hospital, Cheeloo College of Medicine, Shandong University, Jinan, Shandong, China. [13]These authors contributed equally: Yanan An, Chao Wang, Ziqi Wang, Feng Kong.✉E-mail: leijie@email.sdfmu.edu.cn; zhaoshengtian@sdu.edu.cn; ygong@bzmc.edu.cn

in vivo, and ex vivo. Therefore, this peptide appears as strong therapeutic candidates against the SARS-CoV-2 infection.

## Results

### SARS-CoV-2 infection alters the expression of LSR in intestine

The intestine is a crucial target organ for SARS-CoV-2 infection (Lamers et al, 2020). In particular, several studies reported that COVID-19 with gastrointestinal involvement had a worse disease course and that gastrointestinal symptoms could even precede respiratory symptoms (Mao et al, 2020; Song et al, 2020; Zhang et al, 2023). The trans-membrane protein LSR was found to be rich in expression in the villi and crypts of intestine (An et al, 2023). We first sought to investigate the effect of SARS-CoV-2 infection on the expression of LSR. We determined the distribution of LSR in Caco-2 cells and HIEC-6 overexpressing human ACE2 (hACE2-HIEC-6) infected with a chimeric VSV virus expressing the fluorescent reporter GFP reporter or the luciferase reporter, where the glycoprotein of VSV was replaced with the wild-type Spike protein of SARS-CoV-2 with the sequence of the original wild type SARS-CoV-2 strain (VSV-SARS-CoV-2). VSV-SARS-CoV-2 infection was able to alter LSR expression and distribution as demonstrated by quantitative reverse transcription polymerase chain reaction (RT-qPCR) and western blot (Figs. 1A and EV1A). When Caco-2 cells and hACE2-HIEC-6 were exposed to 1 multiplicity of infection (MOI) of VSV-SARS-CoV-2, LSR downregulation was observed (Figs. 1B,C and EV1B,C). LSR levels were significantly decreased 12, 24, and 48 h postinfection in infected Caco-2 cells and hACE2-HIEC-6 (Figs. 1D,E and EV1D,E).

To investigate whether SARS-CoV-2 infection leads to downregulation of endogenous LSR in vivo, we infected knock-in mice expressing the human ACE2 receptor driven by the mouse ACE2 promoter, an animal model commonly used for SARS-CoV-2 infection, with VSV-SARS-CoV-2 intraperitoneal injection. Western blot analysis showed clear band corresponding to the Spike protein following VSV-SARS-CoV-2 infection (Fig. 1F), indicating the successful establishment of the in vivo model of SARS-CoV-2 intestinal infection. Histopathological analysis revealed intestinal injury with increased expression of pro-inflammatory cytokines (*Il2, Il6, Il12, Cxcl10, Tnfα, Ifnα*, and *Ifnγ*) and infiltration of neutrophils in humanized ACE2 mice challenged with VSV-SARS-CoV-2 (Fig. EV1F–J). Notably, both protein and mRNA levels of LSR in intestinal tissues of these infected mice were significantly reduced at 6 h postinfection in contrast to control group (Fig. 1F,G), raising the possibility that reduced LSR expression might have a role in SARS-CoV-2–mediated intestine pathologies. In addition, immunofluorescent staining demonstrated LSR protein expression was reduced predominantly in the villus epithelium but was readily detectable in the crypt compartment (Fig. 1H), in agreement with that SARS-CoV-2 productively infects human gut enterocytes (Lamers et al, 2020; Lehmann et al, 2021).

To validate these results using authentic SARS-CoV-2, we infected Caco-2 cells and ex vivo intestinal tissues with patient isolate of the virus corresponding to the original wild-type SARS-CoV-2 strain (WT), SARS-CoV-2 Omicron variant (Omicron), and SARS-CoV-2 XBB variant (XBB). Consistent with the results of the VSV-SARS-CoV-2 chimera infection assay, LSR expression was also reduced relative to uninfected ones on both mRNA and protein level (Fig. 1I–M). We further validated these results using human intestinal organoids derived from induced pluripotent stem cell (iPSC) infected with VSV-SARS-CoV-2, or SARS-CoV-2 WT, Omicron, or XBB (Fig. 1N) where expression of LSR mRNA and protein were significantly decreased as well (Fig. 1O,P). Moreover, immunostaining analysis indicated that LSR was substantially lower in the intestine of COVID-19 patients compared to those without the disease (Fig. 1Q).

In addition, we incorporated propidium iodide staining in the infected intestine and organoids and observed LSR reduction in the propidium iodide-negative cells on microscopy slides (Fig. 1R,S), indicating that the alteration of LSR expression in intestinal tissue and organoids is not merely a consequence of virus-induced cell death.

Collectively, both pseudotyped virus and authentic SARS-CoV-2 virus trigger LSR downregulation in immortalized cell lines, intestinal organoids, ex vivo intestinal tissues, and humanized ACE2 mouse model.

### Spike protein contributes to LSR downregulation induced by SARS-CoV-2

As both VSV-SARS-CoV-2 containing the Spike protein and authentic SARS-CoV-2 induce LSR downregulation, we next attempted to investigate whether Spike protein is involved in the SARS-CoV-2 exposure-mediated LSR downregulation. As expected, Spike protein treatment caused a decrease in LSR protein level in Caco-2 and hACE2-HIEC-6 that expressed ACE2, but not in HIEC-6 with very low or undetectable ACE2 expression (Baggen et al, 2023) (Fig. 2A), suggesting ACE2 is required for LSR downregulation. Overexpression of Spike can also reduce the level of LSR protein in Caco-2 and hACE2-HIEC-6 (Fig. 2B). Given that viral infection and Spike protein could induce the expression of type I and II interferons (IFNs), we treated Caco-2 monolayer cultures with recombinant human IFN-α (IFN-α 2a), IFN-β, IFN-γ, or IFN-λ proteins for 24 h and quantified LSR mRNA levels using RT-qPCR. We noted inhibitory effects of IFN-α, IFN-γ, and IFN-λ on expression of LSR mRNA (Fig. 2C).

SARS-CoV-2 nonstructural protein NSP1 and NSP14, as well as VSV matrix (M) protein, have been proved to employ divergent mechanisms to suppress host protein expression (Desforges et al, 2001; Hsu et al, 2021; Thoms et al, 2020). We next analyzed whether LSR downregulation in infected cells was also a consequence of translational shutoff triggered by infection. To investigate this, Caco-2 cells were transfected with DNA plasmid encoding M protein, NSP1, or NSP14. After 24 h of transfection, the expression of LSR was assessed via immunoblotting. The results showed that LSR levels were significantly decreased in Caco-2 cells overexpressing M protein, NSP1, or NSP14 (Fig. 2D), indicating that translational shutoff triggered by infection also contribute to the LSR downregulation in infected cells. We also observed that the Spike protein elicited a dose-dependent exacerbation of LSR expression inhibition alongside the translation suppression mediated by M protein, NSP1, or NSP14 (Fig. 2E), indicating that LSR downregulation in infected cells is not solely due to translational shutoff but also a specific consequence of Spike protein activity.

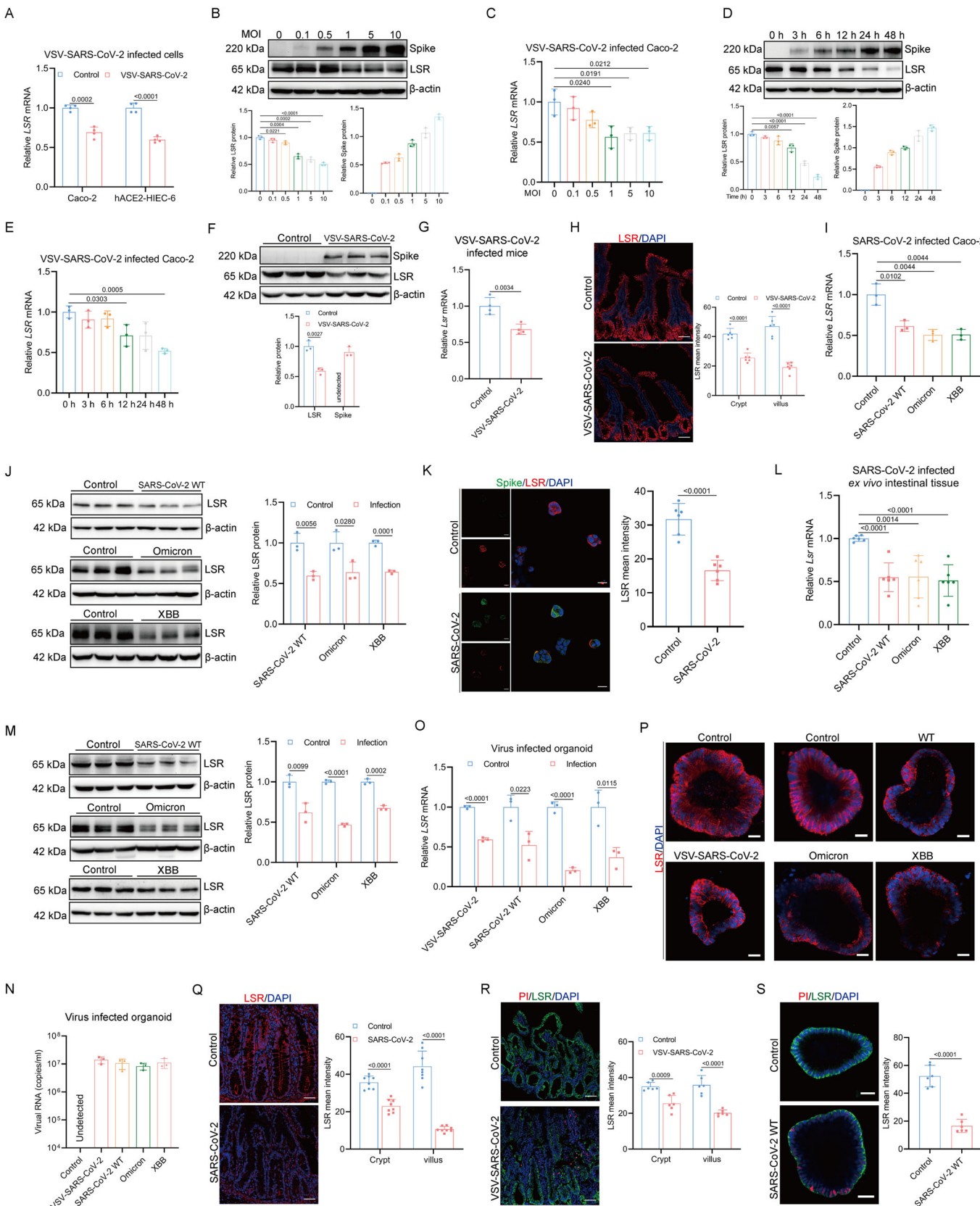

◀ **Figure 1.  SARS-CoV-2 infection alters the expression of LSR in intestine.**

(**A**) Expression of *LSR* mRNA analyzed by RT-qPCR in Caco-2 and hACE2-HIEC-6 cells infected with VSV-SARS-CoV-2 at 1 MOI for 24 h (*n* = 4). (**B, C**) Western blot and densitometric analysis of Spike and LSR (**B**), and expression of *LSR* mRNA analyzed by RT-qPCR (**C**) in Caco-2 cells infected with VSV-SARS-CoV-2 at the indicated MOI for 24 h (*n* = 3). (**D, E**) Western blot and densitometric analysis of Spike and LSR (**D**, *n* = 3), and expression of *LSR* mRNA analyzed by RT-qPCR (**E**, *n* = 4) in Caco-2 cells infected with VSV-SARS-CoV-2 at 1 MOI at indicated time points. (**F–H**) Western blot and densitometric analysis of Spike and LSR (**F**, *n* = 3), expression of *Lsr* analyzed by RT-qPCR (**G**, *n* = 4), and immunofluorescence and quantitative analysis (**H**, *n* = 6) in the duodenum from hACE2-transgenic mice infected with VSV-SARS-CoV-2. (**I, J**) Expression of LSR analyzed by RT-qPCR (**I**), and western blot and densitometric analysis (**J**) in Caco-2 cells infected with SARS-CoV-2 WT, Omicron, and XBB (*n* = 3). (**K**) Immunofluorescence and quantitative analysis of LSR in Caco-2 cells infected with SARS-CoV-2 WT (*n* = 6). (**L, M**) Expression of LSR analyzed by RT-qPCR (**L**), and western blot and densitometric analysis (**M**) in the ex vivo intestinal tissue infected with SARS-CoV-2 WT, Omicron, and XBB (*n* = 3). (**N**) Viral RNA copies analyzed by RT-qPCR using primers targeting VSV-P or SARS-CoV-2 N gene in intestinal organoids infected with VSV-SARS-CoV-2 and SARS-CoV-2 WT, Omicron, and XBB (*n* = 3). (**O, P**) Expression of LSR analyzed by RT-qPCR (**O**, *n* = 3) and immunofluorescence (**P**) in intestinal organoids infected with VSV-SARS-CoV-2 and SARS-CoV-2 WT, Omicron, and XBB. (**Q**) Immunofluorescence and quantitative analysis of LSR in small intestinal biopsy specimens of subjects with and without SARS-CoV-2 infection (*n* = 8). (**R, S**) Immunofluorescence staining images and quantitative analysis of LSR (green) in the PI (red) negative areas of small intestine from hACE2-transgenic mice, infected with or without VSV-SARS-CoV-2 (**R**), and of the intestinal organoids, infected with or without SARS-CoV-2 WT (**S**) (*n* = 6). "*n*" represents number of biological replicates. Scale bars: (**H**), (**K**), (**Q**), and (**R**), 50 μm; (**P**) and (**S**), 20 μm. Data represent mean ± SEM. Unpaired t test was performed. *p* < 0.05, the exact *p*-value is displayed; *p* > 0.05, the *p*-value is not displayed. Source data are available online for this figure.

To determine whether proteasome-mediated degradation or lysosome-mediated degradation mediates LSR downregulation, we pretreated cells with 100 μg/ml cycloheximide (CHX), a well-established protein synthesis inhibitor that functions by blocking the elongation phase of translation to prevent the synthesis of new proteins, and the proteasome inhibitor MG132 or the lysosome inhibitor bafilomycin A1 (Baf-A1) before adding the Spike protein. As shown in Fig. 2F,G, Baf-A1, but not MG132, prevented Spike protein-induced LSR downregulation, suggesting the involvement of the lysosome in this process. The attenuated effect of Baf-A1 treatment on LSR down-regulation was also observed in VSV-SARS-CoV-2-treated and Spike protein-expressing plasmid-transfected Caco-2 cells (Fig. 2H,I), further confirming the involvement of the lysosome in LSR downregulation.

Consequently, the downregulation of LSR induced by SARS-CoV-2 is attributed to the Spike protein, which downregulates LSR via the lysosome pathway.

## LSR acts as a host defense factor against SARS-CoV-2 in intestine

To evaluate the involvement of LSR in SARS-CoV-2 infection of intestinal epithelial cells, we used shRNA against human LSR (LSR-KD) and plasmids encoding human full-length LSR (LSR-OE) to knockdown and overexpress the LSR gene in Caco-2 cells, respectively, then infected the cells with VSV-SARS-CoV-2. LSR knockdown and overexpression were confirmed by RT-qPCR and western blot (Fig. EV2A,B). To assess the infection kinetics of SARS-CoV-2 in the wild type, LSR-KD, and LSR-OE Caco-2 cells, the viral load was quantified by RT-qPCR and a 50% tissue culture infective dose (TCID$_{50}$) assay. As early as 1 h post-infection, viral genome copies diverged concordantly with changes in LSR levels. Notably, LSR silencing resulted in an order of magnitude increase in viral amount, indicating that LSR is essential for inhibiting virus entry (Fig. EV2C). Infectious progeny viruses were found to be released from infected cells as early as 8 h after infection and plateaued around 24 h after infection (Fig. EV2D), in agreement with the previous report that SARS-CoV-2 completes one round of infection, from virus binding and entry to replication and release of de novo infectious particles, within 8 h in Caco-2 cells (Koch et al, 2021). The virus production changes across different LSR level groups, quantified by TCID$_{50}$, were consistent with the observations from the RT-qPCR results (Fig. EV2D).

Infection of Caco-2 cells by GFP-expressing VSV-SARS-CoV-2 was readily assessed qualitatively by fluorescence microscopy and quantitatively by fluorescence intensity. Infection by luciferase-expressing VSV-SARS-CoV-2 provided greater sensitivity. Under either condition, knockdown of LSR led to a significant increase in viral infection, as evidenced by increased luciferase activity (Fig. 3A), GFP intensity (Fig. 3B), VSV phosphoprotein (VSV-P) copy numbers (Fig. 3C), and Spike copy numbers (Fig. EV2F). In contrast, overexpression of human or mouse LSR markedly reduced viral infection (Figs. 3A–C and EV2E,F). In hACE2-293 cells that expresses low endogenous LSR, overexpression of LSR also exerted VSV-SARS-CoV-2 inhibition (Fig. EV2G). Further, measurement of total viral RNA levels using RT-qPCR with SARS-CoV-2 nucleocapsid (N) primer sets revealed that infection by SARS-CoV-2 increased significantly by 10.4-fold (WT), 7.5-fold (Omicron), and 8.2-fold (XBB) in LSR-KD cells (Fig. 3D). These results were further validated by immunofluorescence of the N protein, with nuclei counterstained using DAPI. The percentage of infected Caco-2 cells increased by 2.7-fold (WT), 2.5-fold (Omicron), and 2.8-fold (XBB), and the viral N protein signal increased by 2.9-fold (WT), 3.1-fold (Omicron), and 3.1-fold (XBB) in cells with disrupted LSR alleles (Fig. 3E). In addition, disruption of LSR led to a 10.5-fold increase in viral load in hACE2-A549 lung epithelial cells (Fig. EV2H). In contrast, LSR overexpression reduced N protein levels across all isolates (Figs. 3D,E and EV2H). These findings indicate that LSR exhibits antiviral activity against SARS-CoV-2 and functions as a crucial host defense factor against the virus.

Given the uncertainty in most relevant in vitro correlates of protection, we generated the intestinal-epithelium-specific *Lsr*-deficient mice expressing the human ACE2 (hACE2-*Lsr*$^{\text{vill KO}}$) by crossing *Lsr*$^{\text{vill KO}}$ with human ACE2 knockin mice (Sun et al, 2020) (Fig. 3F). To simulate SARS-CoV-2 infection in the intestine in a more physiologically relevant setting, we used ex vivo intestinal tissues directly extracted from hACE2-expressing mice and immediately infected upon resection (Udden et al, 2017). The ex vivo intestinal tissues were infected with VSV-SARS-CoV-2 and SARS-CoV-2 with an inoculum of 1 MOI. At 6 h postinfection, the tissues were harvested, and immunofluorescence staining, hema-toxylin and eosin (H&E) staining and RT-qPCR were performed to assess viral load, intestinal damage, and immune cell infiltration (Fig. 3F). Our results indicated that the small intestinal tissue from

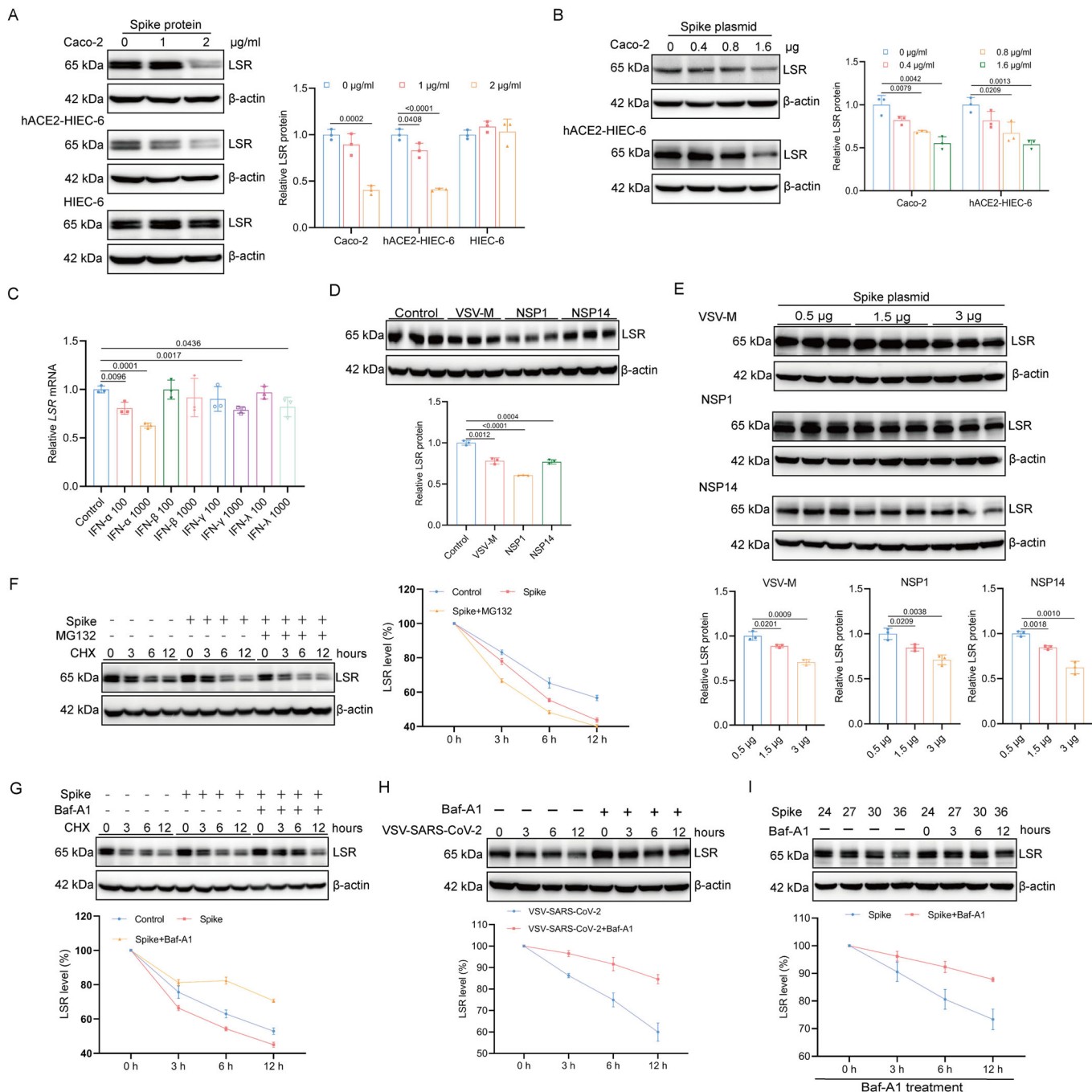

**Figure 2. Spike protein contributes to LSR downregulation induced by SARS-CoV-2.**

(A) Western blot and densitometric analysis of LSR in Caco-2, hACE2-HIEC-6, and HIEC-6 cells treated with the indicated amounts of recombinant Spike protein for 24 h ($n = 3$). (B) Western blot and densitometric analysis of LSR in Caco-2 and hACE2-HIEC-6 cells transfected with indicated amounts of plasmids expressing Spike protein ($n = 3$). (C) Expression of LSR mRNA measured by RT-qPCR in Caco-2 cells treated with or without 100 IU/ml or 1000 IU/ml of recombinant human IFN-α, IFN-β, IFN-γ, or IFN-λ for 24 h ($n = 3$). (D) Western blot and densitometric analysis of LSR in Caco-2 cells transfected with plasmids encoding VSV M protein, NSP1, or NSP14 ($n = 3$). (E) Western blot and densitometric analysis of LSR in Caco-2 cells transfected with plasmids encoding VSV M protein, NSP1, or NSP14, together with indicated amounts of plasmids expressing Spike protein ($n = 3$). (F, G) Western blot and densitometric analysis of LSR in Caco-2 cells pretreated with 10 μM proteasome inhibitor MG132 (F) or 200 nM lysosome inhibitor bafilomycin A1 (Baf-A1) (G) for 2 h, followed by 2 μg/ml of recombinant SARS-CoV-2 Spike protein treatment for 0, 3, 6, and 12 h in the presence of 100 μg/ml CHX ($n = 3$). (H) Western blot and densitometric analysis of LSR in Caco-2 cells pretreated with 200 nM Baf-A1 for 2 h, followed by infection with VSV-SARS-CoV-2 for 0, 3, 6, and 12 h ($n = 3$). (I) Western blot and densitometric analysis of LSR in Caco-2 cells transfected with plasmids expressing Spike protein for 24 h, followed by treatment with 200 nM Baf-A1 for 0, 3, 6, and 12 h ($n = 3$). "$n$" represents number of biological replicates. Data represent mean ± SEM. Unpaired t test was performed. $p < 0.05$, the exact $p$-value is displayed; $p > 0.05$, the $p$-value is not displayed. Source data are available online for this figure.

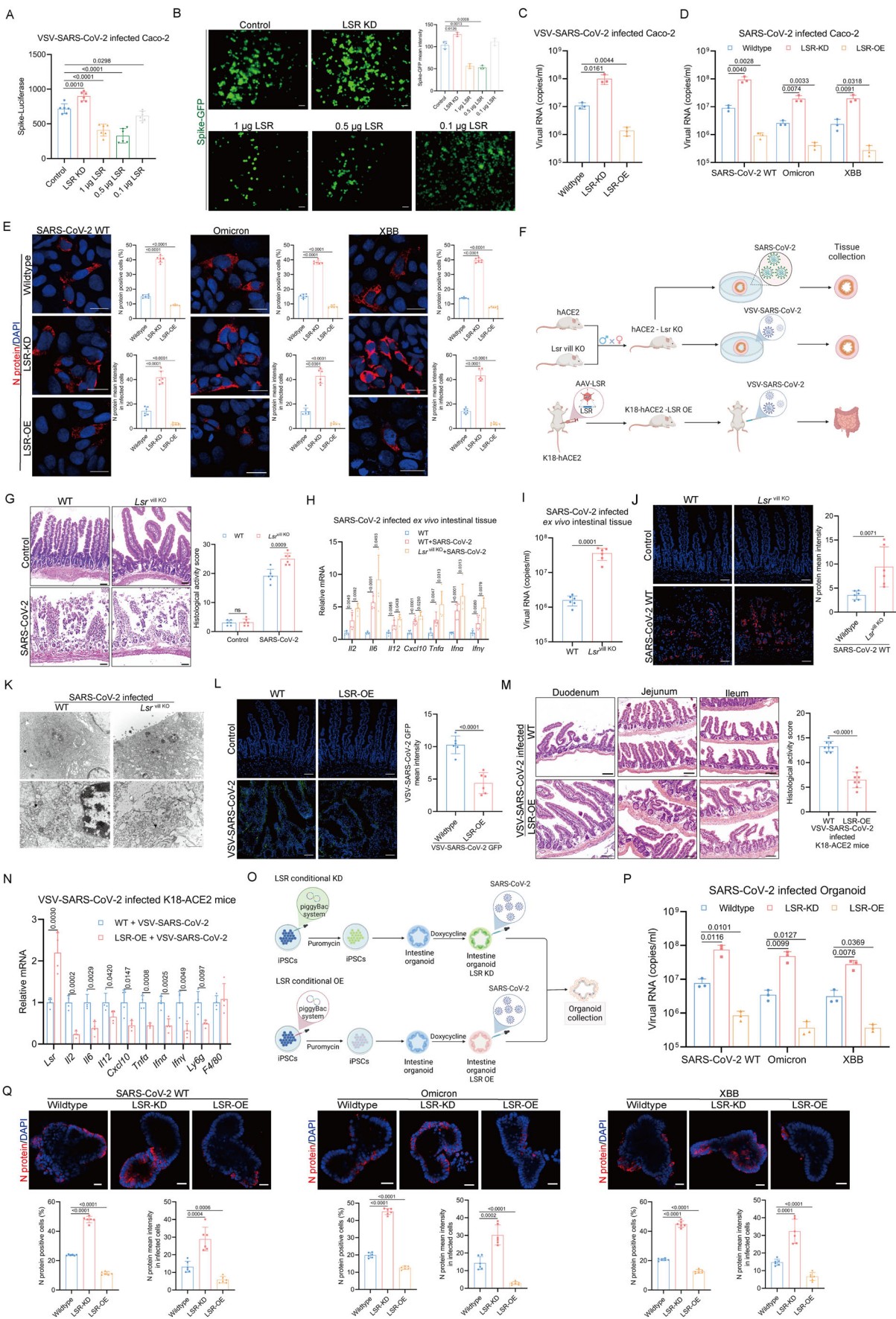

◀   **Figure 3.  LSR acts as a host defense factor against SARS-CoV-2 in intestine.**

(A, B) Luciferase analysis (A, $n = 6$), and fluorescence images and quantitative analysis (B, $n = 3$) of Caco-2 cells transfected with control vector, plasmids containing LSR shRNA, or indicated amounts of plasmids containing human LSR transcript variant 2, and infected with VSV-SARS-CoV-2 at 1 MOI for 24 h. (C) Viral RNA copies analyzed by RT-qPCR using primers targeting VSV-P in wild type, LSR-KD, and LSR-OE Caco-2 cells infected with VSV-SARS-CoV-2 at 1 MOI for 24 h ($n = 3$). (D, E) RT-qPCR analysis of viral RNA copies using primers targeting SARS-CoV-2 N gene (D, $n = 3$), immunofluorescence and quantitative analysis of SARS-CoV-2 N protein (E, $n = 6$) in wild type, LSR-KD, and LSR-OE Caco-2 cells infected with SARS-CoV-2 WT, Omicron, and XBB. (F) Schematic illustration of ex vivo intestinal tissue infection model and mouse infection model. The graphic was created with BioRender.com. (G) H&E staining and histological activity score of the control and SARS-CoV-2 WT infected ex vivo intestinal tissues harvested from hACE2-WT and hACE2-*Lsr*[vill KO] mice ($n = 6$). (H, I) Expression of *Il2, Il6, Il12, Cxcl10, Tnfa, Ifna,* and *Ifny* mRNA measured by RT-qPCR (H), and viral RNA copies analyzed by RT-qPCR using primers targeting VSV-P (I) in the SARS-CoV-2 WT infected ex vivo intestinal tissues harvested from hACE2-WT and hACE2-*Lsr*[vill KO] mice ($n = 6$). (J) Immunofluorescence staining images and quantitative analysis of N protein in SARS-CoV-2 WT infected ex vivo intestinal tissues harvested from hACE2-WT and hACE2-*Lsr*[vill KO] mice ($n = 6$). (K) TEM in the SARS-CoV-2 WT infected ex vivo intestinal tissues harvested from hACE2-WT and hACE2-*Lsr*[vill KO] mice (white triangles indicate intercellular tight junctions, black stars indicate microvilli, black triangles indicate virus). (L) Fluorescence images and quantitative analysis of GFP in small intestines from K18-hACE2-WT (WT) and K18-hACE2-LSR-OE (LSR-OE) mice infected with or without VSV-SARS-CoV-2 ($n = 6$). (M, N) H&E staining and histological activity score (M, $n = 8$), and the expression of *Il2, Il6, Il12, Cxcl10, Tnfa, Ifna, Ifny, Ly6g,* and *F4/80* mRNA analyzed by RT-qPCR (N, $n = 6$) in the small intestines from K18-hACE2-WT (WT) and K18-hACE2-LSR-OE (LSR-OE) mice infected with VSV-SARS-CoV-2. (O) Schematic illustration of organoid infection model. The graphic was created with BioRender.com. (P, Q) RT-qPCR analysis of viral RNA copies using primers targeting SARS-CoV-2 N gene (P, $n = 3$), immunofluorescence staining images and quantitative analysis of SARS-CoV-2 N protein (Q, $n = 6$) in intestinal organoids infected with SARS-CoV-2 WT, Omicron, and XBB. "$n$" represents number of biological replicates. Scale bars: (B), 100 μm; (G), (J), (L), and (M), 50 μm; (K), 1 μm; (E) and (Q), 20 μm. Data represent mean ± SEM. Unpaired t test was performed. $p < 0.05$, the exact $p$-value is displayed; $p > 0.05$, the $p$-value is not displayed. Source data are available online for this figure.

hACE2-*Lsr*[vill KO] mice exhibited more severe villus disruption, separation of lamina propria, and submucosal edema, and with enhanced leukocyte infiltration and upregulated expression of pro-inflammatory cytokines (*Il2, Il6, Il12, Cxcl10, Tnfa, Ifna,* and *Ifny*) compared to hACE2-WT mice (Figs. 3G,H and EV2I,J). Compared to virus-exposed intestines from hACE2-WT mice, the intestines from hACE2-*Lsr*[vill KO] showed a significant increase in viral load, as evidenced by the increased viral RNA and augmented N protein staining or GFP signal (Figs. 3I,J and EV2K). The transmission electron microscopy (TEM) results revealed a higher occurrence of viral particles in the intestinal cells of hACE2-*Lsr*[vill KO] mice compared to hACE2-WT mice, accompanied by a greater disruption of microvilli and reduced intercellular tight junctions in the hACE2-*Lsr*[vill KO] mice (Fig. 3K). Consistent with the findings observed in the ex vivo intestinal tissues, a significant elevation in viral load was observed in the small intestines of hACE2-*Lsr*[vill KO] mice compared to the hACE2-WT group after VSV-SARS-CoV-2 intraperitoneal injection (Fig. EV2L). In contrast, no notable difference in viral load was identified in the lungs (Fig. EV2M). To further evaluate the potential protective effect of LSR over-expression in vivo, we employed knock-in mice expressing the human ACE2 receptor driven by the cytokeratin-18 (K18) gene promoter (K18-hACE2) as a model of severe SARS-CoV-2 infection. LSR overexpression significantly decreased viral load, as evidenced by the reduced mean GFP intensity, resulting in significantly mitigated tissue damage and notable reduction in pro-inflammatory cytokines and neutrophil infiltration in the intestines of K18-hACE2-LSR-OE mice compared to K18-hACE2-WT mice (Figs. 3L–N and EV2N).

We also took advantage of organoid culture to confirm the findings. We have previously shown that LSR plays a key role in the regulation of intestinal stem cell function and differentiation and LSR-mutant iPSC-derived organoids failed to produce extensive budding and contained no Paneth cells. Thus, iPSC lines with a doxycycline (dox)-inducible LSR knockdown and overexpression were generated using the piggyBac system and then propagated as organoids (Fig. 3O). After 15 days of differentiation, organoids were treated with dox for 48 h to induce the expression of LSR or LSR shRNA, then infected with VSV-SARS-CoV-2 or SARS-CoV-2

(Fig. 3O). In accordance with VSV-SARS-CoV-2 infection data (Fig. EV2O), LSR overexpression reduced the levels of infection down to 11%, whereas silencing LSR led to a 10.8-fold increase in SARS-CoV-2 infection by analyzing the viral RNA through RT-qPCR (Fig. 3P). Immunofluorescence staining for the N protein showed that LSR deficiency increased, while LSR overexpression reduced, both the percentage of infected cells and the viral burden in single cells of intestinal organoids exposed to SARS-CoV-2 (Fig. 3Q).

Considered together, these results show that LSR knockout promotes while LSR overexpression inhibits SARS-CoV-2 infection in the intestine in vitro, in vivo, and ex vivo.

## LSR blocks SARS-CoV-2 entry

Next, we sought to identify which step of the viral life cycle is targeted by LSR. Similar to the original SARS-CoV virus, SARS-CoV-2 uses its Spike glycoprotein on the envelope to engage the host ACE2 as the entry receptor, leading to the subsequent membrane fusion or endocytosis for virus entry. We tested whether LSR blocks SARS-CoV-2 binding and internalization. Viral binding at 4 °C for 1 h was significantly increased in LSR-KD Caco-2 cells (Fig. 4A), but not in LSR-KD HEK293 cells which express very low levels of ACE2 (Fig. 4B), demonstrating that LSR modulate SARS-CoV-2 infection in an ACE2-dependent fashion. Shifting cells to 37 °C for 1 h enabled the internalization of bound SARS-CoV-2 followed by trypsin digestion of uninternalized virus. Deficiency of the cell-surface-expressed LSR also significantly increased this parameter (Fig. 4C). Further, we used pseudovirus assays to determine whether LSR regulates viral infection at the level of entry. We challenged Caco-2 cells with a replication-defective HIV-1 vector pseudotyped bearing SARS-CoV-2 Spike and assayed for luciferase activity at 24 h post infection. Loss of LSR markedly increased the infectivity of pseudoviruses containing Spike (Fig. 4D). Moreover, LSR over-expression in Caco-2 cells blocked viral binding and internalization with all isolates (Fig. 4A–C), indicating that LSR is crucial for blocking the entry of virus.

SARS-CoV-2 uses two distinct pathways for cell entry: a cathepsin-dependent endosomal fusion pathway and a TMPRSS2-

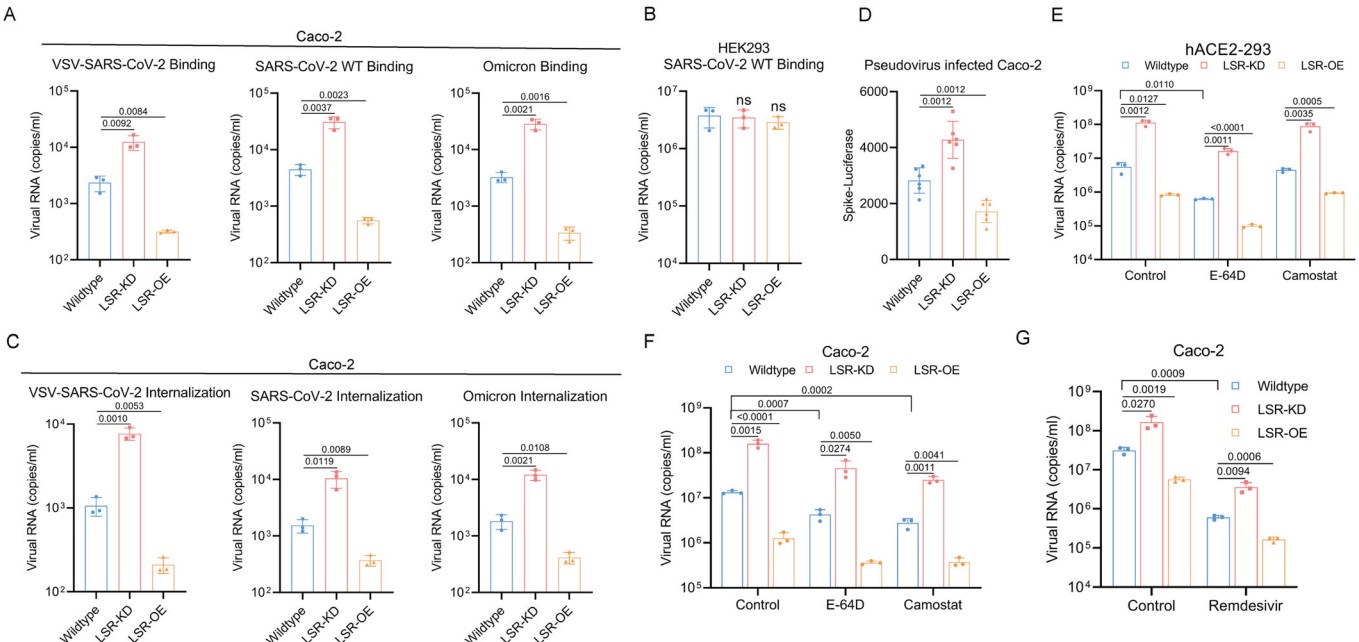

**Figure 4. LSR blocks SARS-CoV-2 entry.**

(A) Virus binding assay measured by RT-qPCR in wild type, LSR-KD, and LSR-OE Caco-2 cells infected with VSV-SARS-CoV-2, and SARS-CoV-2 WT and Omicron at 1 MOI ($n = 3$). (B) Virus binding assay measured by RT-qPCR in wild type, LSR-KD, and LSR-OE HEK293 cells infected with SARS-CoV-2 WT at 1 MOI ($n = 3$). (C) Virus internalization assay measured by RT-qPCR in wild type, LSR-KD, and LSR-OE Caco-2 cells infected with VSV-SARS-CoV-2, and SARS-CoV-2 WT and Omicron at 1 MOI ($n = 3$). (D) The viral load assessed by luciferase analysis in wild type, LSR-KD, and LSR-OE Caco-2 cells infected with replication-defective pseudovirus bearing Spike protein from SARS-CoV-2 WT at 1 MOI for 24 h ($n = 3$). (E, F) Viral RNA copies analyzed by RT-qPCR using primers targeting SARS-CoV-2 N gene in wild type, LSR-KD, and LSR-OE hACE2-293 (E) and Caco-2 (F) cells pretreated with E-64d (20 µM) or Camostat (20 µM) and infected with SARS-CoV-2 WT at 1 MOI for 24 h ($n = 3$). (G) Viral RNA copies analyzed by RT-qPCR using primers targeting SARS-CoV-2 N gene in Caco-2 cells infected with SARS-CoV-2 WT at 1 MOI, then transfected with control vector or plasmids containing LSR shRNA or full-length human LSR, with 3.7 µM remdesivir added in the medium 1 h before infection and during the infection ($n = 3$). "$n$" represents number of biological replicates. Data represent mean ± SEM. Unpaired t test was performed. $p < 0.05$, the exact p-value is displayed; $p > 0.05$, the p-value is not displayed. Source data are available online for this figure.

dependent cell-surface fusion pathway (Jackson et al, 2022). In hACE2-293 cells that lack any detectable TMPRSS2 expression, inhibiting the endosomal fusion pathway with the cysteine protease inhibitor E-64d significantly attenuated infection, whereas the TMPRSS2 inhibitor Camostat had no effect (Fig. 4E). In Caco-2 cells that equally allow cell entry through both the direct membrane fusion pathway and the endocytosis pathway, inhibition of TMPRSS2 using Camostat and inhibition of cathepsin-L using E-64d blocked viral infection of Caco-2 cells (Fig. 4F). However, treatment with Camostat or E-64d did not affect the impact of LSR on SARS-CoV-2 entry (Fig. 4E,F), suggesting that the effect of LSR on endocytosis or membrane fusion might be neglectable in both hACE2-293 and Caco-2 cell lines.

We next examined whether LSR influences viral replication. To do so, Caco-2 cells were pretreated with remdesivir and infected with SARS-CoV-2 for 1 h and then retrovirus carrying the shRNA against human LSR (LSR-KD) and human full-length LSR (LSR-OE) were added to the cells to knockdown and overexpress the LSR gene, respectively, and incubated for 24 h at 37 °C. Remdesivir was kept in the medium during infection. Viral RNA in cells was measured by RT-qPCR, and a similar level of reduction was achieved in the presence of remdesivir despite different expression level of LSR (Fig. 4G). We concluded that the LSR does not play a role in viral replication in the intestinal epithelium.

Collectively, these results suggest that LSR inhibits SARS-CoV-2 infection by blocking the virus binding and the subsequent internalization.

## LSR alters interactions between Spike and ACE2

SARS-CoV-2 Spike binds to the primary receptor ACE2 to mediate virus attachment and entry into the host cells. We first checked the levels of ACE2 on the plasma membrane for viral receptor binding in LSR-KD cells. Surface proteins labeled with membrane-impermeable biotin were subjected to streptavidin pulldown and immunoblotting. LSR deficiency did not alter the amount of ACE2 reaching the total or the surface expression of ACE2 (Fig. EV3A). Next, we tested whether LSR interacts with SARS-CoV-2 Spike to affect virus entry subsequently. We co-expressed full-length Spike and full-length LSR in HEK293 cells and co-IP assay revealed that LSR didn't exhibit pronounced binding to Spike (Fig. EV3B). Preincubation with C. perfringens type V neuraminidase to enzymatically remove sialic acid or with soluble heparin sulfate (HS) to inhibit virus-HS proteoglycan interaction reduced the infection in VSV-SARS-CoV-2-transduced Caco-2 cells and similar decline of virus binding was observed between wild type and LSR-KD group (Fig. EV3C), excluding the possibility that LSR can interfere in the interaction of Spike with sialic acids or HS

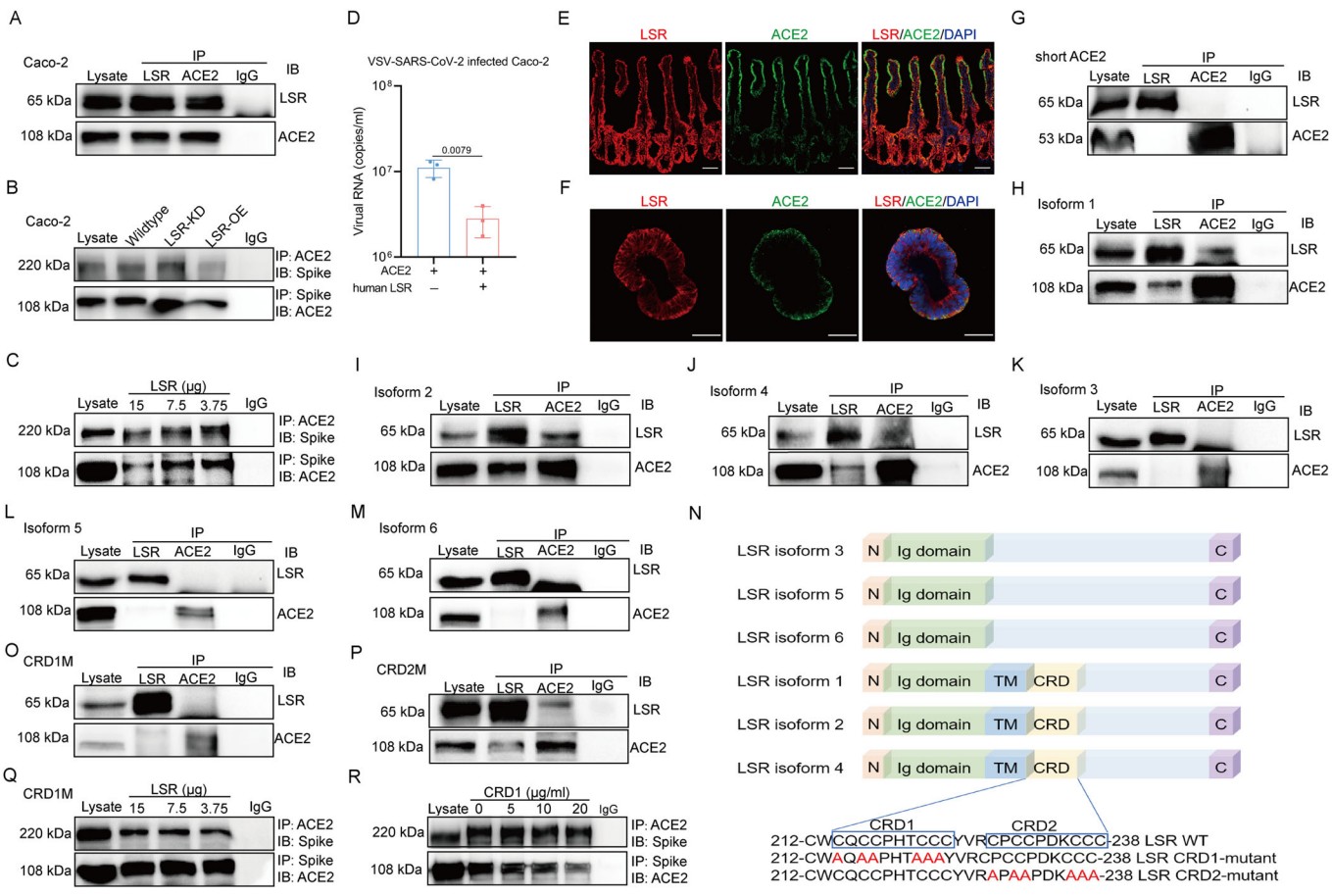

**Figure 5. LSR alters interactions between Spike and ACE2.**

(A) Co-IP showing that human LSR interacts with human ACE2 in Caco-2 cells. (B) Co-IP showing that the interaction between ACE2 and Spike was increased in LSR-KD Caco-2 cells, and the interaction was decreased in LSR-OE Caco-2 cells. (C) Co-IP showing that with the reduction of LSR overexpressing plasmids transfected, the interaction between ACE2 and Spike increased. (D) Viral RNA copies analyzed by RT-qPCR using primers targeting VSV-P in Caco-2 cells transfected with human ACE2 and LSR or human ACE2 alone, and infected with VSV-SARS-CoV-2 at 1 MOI for 24 h ($n = 3$). (E, F) Immunofluorescence analysis showing that ACE2 and LSR colocalize in the brush border of small bowel (E) and apical side of intestinal organoids (F). (G) Co-IP showing that LSR cannot interact with short ACE2. (H–J) Co-IP showing that human LSR isoform 1 (H), isoform 2 (I), and isoform 4 (J) can interact with ACE2. (K–M) Co-IP showing that human LSR isoform 3 (K), isoform 5 (L), and isoform 6 (M) cannot interact with human ACE2. (N) Schematic diagram of different human LSR isoforms and CRD domain mutation. (O) Co-IP showing that LSR with CRD1 mutation domain cannot interact with ACE2. (P) Co-IP showing that LSR with CRD2 mutation domain interacts with ACE2. (Q) Co-IP showing that with the reduction of the plasmids expressing CRD1 mutation LSR, the interaction between ACE2 and Spike had no changes. (R) Co-IP showing that with the increase of CRD1 treatment, the interaction between ACE2 and Spike decreased. All the Co-IP assays were performed in doubly or triply transfected HEK293 cells. IB, immunoblot; IP, immunoprecipitation. "$n$" represents number of biological replicates. Scale bars: (E) and (F), 50 μm. Data represent mean ± SEM. Unpaired t test was performed. $p < 0.05$, the exact $p$-value is displayed; $p > 0.05$, the $p$-value is not displayed. Source data are available online for this figure.

proteoglycans which have been proved to be important for binding and viral entry of SARS-CoV-2 (Bermejo-Jambrina et al, 2021; Nguyen et al, 2022).

Next, we analyzed whether LSR interacts with the ACE2, performed the co-IP assay by transfecting plasmids expressing human ACE2 and LSR into HEK293 cells and observed the interaction between the two proteins (Fig. EV3D). Moreover, we confirmed the endogenous interaction between ACE2 and LSR in Caco-2 cells (Fig. 5A). The simultaneous binding of ACE2 to Spike and LSR suggested the possibility that LSR binding might affect the Spike and ACE2 interaction. To investigate the effect of LSR on the interaction between Spike and ACE2, wild type, LSR-KD, and LSR-OE Caco-2 cells were transfected with Spike expression plasmid, and followed by co-IP analysis. The results revealed that

increased LSR was sufficient to decrease association of ACE2 and Spike (Fig. 5B). Moreover, HEK293 cells were transfected with the same amount of Spike and ACE2 expression plasmid together with 15 μg, 7.5 μg, or 3.75 μg LSR expression plasmid. The co-IP results confirmed that LSR interacted with ACE2, and the addition of LSR impeded Spike binding to ACE2 (Fig. 5C). Further, co-expression of LSR and ACE2 led to a decrease in VSV-SARS-CoV-2 infection compared to ACE2-expressing Caco-2 cells (Fig. 5D). Double immunofluorescent staining revealed colocalization of ACE2 and LSR in the brush border of small bowel and apical side of intestinal organoids (Fig. 5E,F). These data identify that LSR has the potential to bind to ACE2 and competes with the Spike for ACE2. Notably, LSR could not bind to a novel short ACE2 isoform (Fig. 5G), which lacks 356 N-terminal amino acids and is not able to bind SARS-

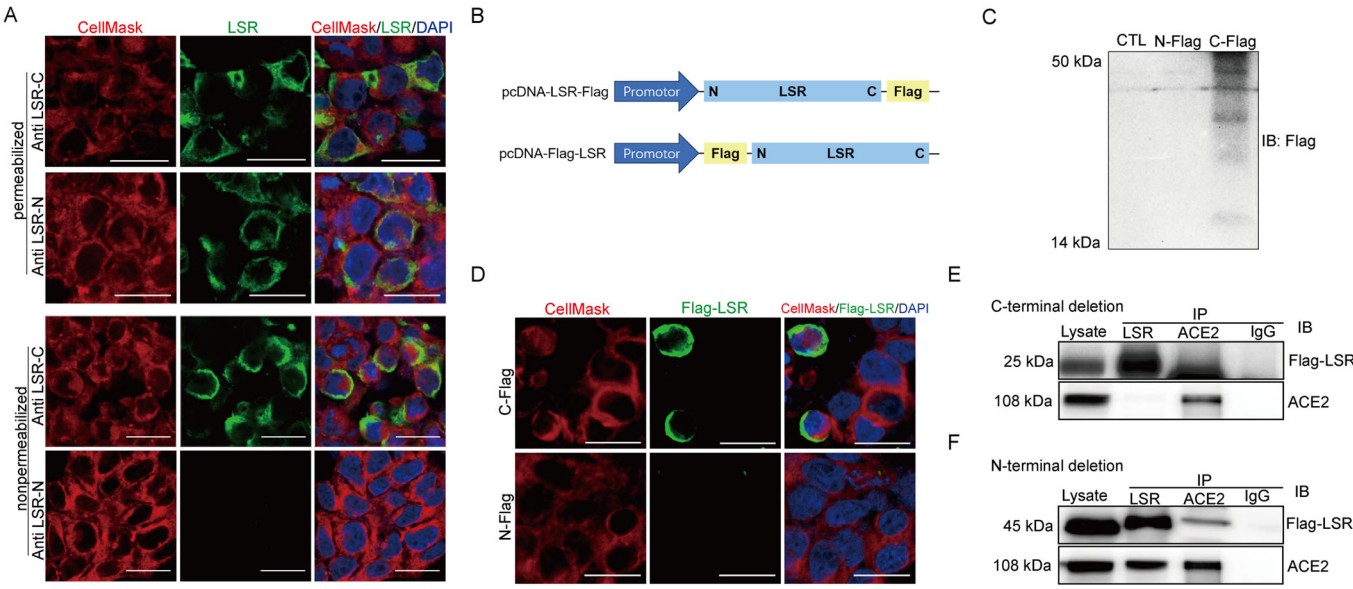

**Figure 6.  C terminus of LSR is located extracellularly.**

(A) Immunofluorescence analysis showing that LSR was detected in both the nonpermeabilized and permeabilized cells by using LSR C-terminal domain antibody, while LSR was only detected in the permeabilized cells by using LSR N-terminal domain antibody. (B) Schematic diagram of the plasmids expressing human LSR with a flag tag in the C-terminal or N-terminal. (C, D) The extracellular segments of LSR by cell surface biotinylating analysis (C) and immunofluorescence analysis (D) showing that anti-Flag antibody detected LSR proteins only in pcDNA-LSR-Flag transfected cells. (E) Co-IP showing that LSR with C-terminal deletion cannot interact with ACE2. (F) Co-IP showing that LSR with N-terminal deletion interacts with ACE2. All the Co-IP assays were performed in doubly or triply transfected HEK293 cells. IB immunoblot, IP immunoprecipitation. Scale bars: (A) and (D), 20 μm. Source data are available online for this figure.

CoV-2 (Blume et al, 2021), suggesting the LSR binding sites are in N-terminal region of ACE2.

Co-expression of full-length ACE2 with all six human LSR isoforms (NM_015925.7, NM_205834.4, NM_205835.4, NM_001260489.2, NM_001260490.2, and NM_001385215.1) individually in HEK293 cells, followed by immunoprecipitation, demonstrated that ACE2 interacted with isoform 1, isoform 2, and isoform 4, but not isoform 3, isoform 5, or isoform 6 which lack transmembrane (TM) domain and CRDs (Fig. 5H–M). LSR contains two CRDs (CQCCPHTCCC and CPCCPDKCCC) and we next mutated each cysteine residue to alanine of CRD region and tested whether these LSR mutants still interact with ACE2 (Fig. 5N). Co-immunoprecipitation experiments revealed that LSR with a CRD1 mutation failed to bind to ACE2 (Fig. 5O), whereas LSR with a CRD2 mutation still retained its ability to bind to ACE2 (Fig. 5P), indicating LSR CRD1, but not the CRD2, is responsible for the interaction and for the inhibitory effect on viral entry into host cells. In addition, the addition of various doses of LSR with CRD1 mutation did not have a significant impact on the binding of Spike to ACE2 (Fig. 5Q). Strikingly, mouse LSR can also bind to human ACE2 (Fig. EV3E) and inhibit SARS-CoV-2 infection to a similar extent as human LSR (Fig. EV3F). The CRD region of LSR is the most conserved region across diverse mammalian species, further implying CRD should be responsible for ACE2 binding. To test whether small LSR-derived CRD peptide can also interfere with Spike binding to ACE2 receptors, we synthesized the CRD1 oligopeptide using the amino acid sequence of CRD1 domain in LSR and evaluated the impact on co-IP of Spike and ACE2. As a result, incubation with the increasing amounts of CRD1 peptide caused a progressive reduction of probed interactions (Fig. 5R).

These data demonstrated that the CRD1 at C-terminus of LSR was necessary for ACE2 binding.

LSR contain one putative transmembrane (TM) domain, the position of the N and C terminus of LSR remains to be studied due to unresolved three-dimensional structure. To differentiate N-terminal or the C-terminal end of LSR ends up exposed to the outside of the cell, pcDNA-LSR was transfected into HEK293 cells and immunofluorescent staining with or without cellular membrane permeabilization was performed using two commercial antibodies targeting N-terminus and C-terminus of LSR, respectively. The fluorescence signal could be detected in both the nonpermeabilized and permeabilized cells by using a LSR C-terminal domain antibody raised against amino acids 364-473, which suggests that C terminus is located extracellularly (Fig. 6A). In contrast, permeabilization was essential by using antibody against the N terminus of LSR (amino acids 35-205), suggesting that N terminus is located in the cytosol of cells (Fig. 6A). Furthermore, a Flag tag was fused to the N or C terminus of LSR to generate two Flag-fused constructs (pcDNA-Flag-LSR and pcDNA-LSR-Flag) (Fig. 6B). These two constructs were then transfected into HEK293 cells and cell surface biotinylation methods was used to verify cell surface expression of the Flag-labeled LSR. Western blot probed with anti-Flag antibody detected LSR proteins in pcDNA-LSR-Flag transfected cells (Fig. 6C), suggesting that the epitope and carboxy tail of LSR are located extracellularly. Location of the N and C termini of human LSR protein was further determined by confocal immunofluorescent staining with Flag antibody of HEK293 cells expressing these two fusion constructs. The data showed that immunofluorescence staining was positive only in the nonpermeabilized cells that were transfected with C-terminal Flag-fused LSR (Fig. 6D). In contrast, in the nonpermeabilized cells that expressed N-terminal Flag-fused LSR, the immunofluorescent

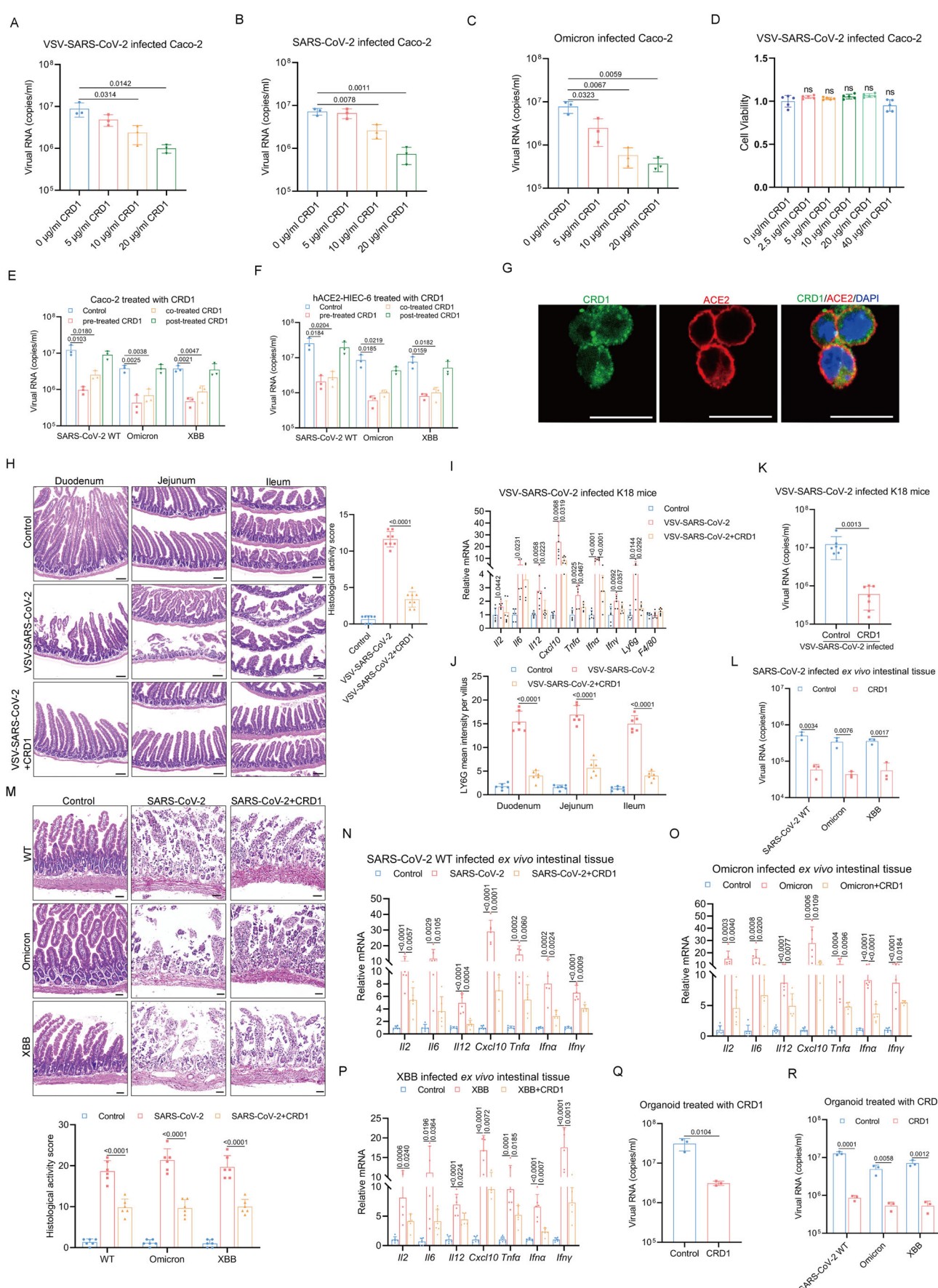

◄   **Figure 7.   LSR-derived CRD1 peptide inhibits infection of SARS-CoV-2.**

(A) Viral RNA copies analyzed by RT-qPCR using primers targeting VSV-P in Caco-2 cells treated with indicated concentrations of CRD1 and infected with VSV-SARS-CoV-2 ($n = 3$). (B, C) Viral RNA copies analyzed by RT-qPCR using primers targeting SARS-CoV-2 N gene in Caco-2 cells treated with indicated concentrations of CRD1 and infected with SARS-CoV-2 WT (B) and Omicron (C) ($n = 3$). (D) Cell viability determined by CCK8 assay in Caco-2 cells treated with indicated concentrations of CRD1 and infected with VSV-SARS-CoV-2 ($n = 5$). (E, F) Viral RNA copies analyzed by RT-qPCR using primers targeting SARS-CoV-2 N gene in Caco-2 cells (E) and hACE2-HIEC-6 cells (F) treated with 10 μg/ml CRD1 −1 h prior to infection, during infection (0 h), and +1 h post infection with SARS-CoV-2 WT, Omicron, and XBB ($n = 3$). (G) Immunofluorescence analysis of ACE2 in hACE2-293 cells treated with FITC-labeled CRD1 peptide for 6 h. (H–K) H&E staining and histological activity score (H, $n = 8$), mRNA expression of *Il2, Il6, Il12, Cxcl10, Tnfa, Ifna, Ifny, Ly6g,* and *F4/80* analyzed by RT-qPCR (I, $n = 6$), quantitative analysis of LY6G mean intensity per villus (J, $n = 6$), and Viral RNA copies analyzed by RT-qPCR using primers targeting VSV-P (K, $n = 6$) in small intestines from control and VSV-SARS-CoV-2 infected K18-hACE2 mice with and without CRD1 treatment before infection. (L–P) Viral RNA copies analyzed by RT-qPCR using primers targeting SARS-CoV-2 N gene (L, $n = 3$), H&E staining and histological activity score (M, $n = 6$), and mRNA expression of *Il2, Il6, Il12, Cxcl10, Tnfa, Ifna,* and *Ifny* analyzed by RT-qPCR (N–P, $n = 6$) in control and SARS-CoV-2 (WT, Omicron, and XBB) infected ex vivo intestinal tissues with and without CRD1 treatment before infection from K18-hACE2 mice. (Q, R) Viral RNA copies analyzed by RT-qPCR using primers targeting VSV-P or SARS-CoV-2 N gene in intestinal organoids treated with PBS or CRD1 at 1 h before being infected with VSV-SARS-CoV-2 (Q) or SARS-CoV-2 (WT, Omicron, and XBB) (R) for 24 h ($n = 3$). "$n$" represents number of biological replicates. Scale bars: (H) and (M), 50 μm; (G), 20 μm. Data represent mean ± SEM. Unpaired t test was performed. $p < 0.05$, the exact *p*-value is displayed; $p > 0.05$, the *p*-value is not displayed. Source data are available online for this figure.

signal was negative (Fig. 6D). In contrast to the C-terminal deletion, the N-terminal deletion mutant of LSR retained their binding to ACE2 in the co-IP assay (Fig. 6E,F), providing further evidence that the C terminus of LSR is located in the extracellular side.

Collectively, the above results suggest that LSR binds to ACE2 through its extracellular C-terminal region, and that this decreases the ability of ACE2 to productively interact with Spike.

## LSR-derived CRD1 peptide inhibits infection of SARS-CoV-2

We next tested whether the peptide CRD1 could inhibit infection by major SARS-CoV-2 variants using both the VSV-SARS-CoV-2 infection assay and the authentic SARS-CoV-2 infection assay. Caco-2 and ACE2-HIEC-6 pretreated with CRD1 were infected with VSV-SARS-CoV-2 or SARS-CoV-2. We found that the viral infectivity decreased in a dose-dependent manner as the concentration of CRD1 increased with little impact on cell viability (Figs. 7A–D and EV4A–D), suggesting that CRD1 plays an essential role in inhibiting viral infectivity. The inhibition of CRD1 for VSV-SARS-CoV-2, SARS-CoV-2 WT, and SARS-CoV-2 Omicron were found to have inhibition of infection in the presence of 5 μg/ml, 10 μg/ml, and 20 μg/ml of CRD1, respectively (Figs. 7A–C and EV4A–C). By performing time-of-addition analysis, we found the antiviral activity of CRD1 required early addition in Caco-2 and hACE2-HIEC-6 cells, confirming that blocking LSR affects the entry stage of the SARS-CoV-2 life cycle (Fig. 7E,F). We also explored the role of LSR on viral infection when the SARS-CoV-2 was bound to cell receptors. For this purpose, Caco-2 cells were infected with VSV-SARS-CoV-2 at 1 MOI at 4 °C for 1 h. Then the cells were washed twice to remove the unattached viruses, and treated with PBS or 10 μg/ml CRD1 for 24 h. Following the addition of CRD1 and a 24-h incubation period at 37 °C, CRD1 continued to demonstrate an inhibitory effect on the virus infection (Fig. EV4E). Moreover, FITC-labeled CRD1 peptide was incubated with hACE2-293 cells, and clear colocalizations of CRD1 with cell membrane ACE2 were observed by confocal microscopy (Fig. 7G). Incubation with CRD1 also inhibited SARS-CoV-2 infection in hACE2-293 cells (Fig. EV4F).

To further investigate the inhibitory effect of CRD1 on SARS-CoV-2 infection in vivo, the K18-hACE2 humanized mice were treated with either vehicle or CRD1 peptide as indicated before VSV-SARS-CoV-2 infection. The histological study showed that

CRD1-treated mice had less pathological damage than the control group (Fig. 7H). Moreover, the infiltration of neutrophil and the expression of *Il2, Il6, Il12, Cxcl10, Tnfα, Ifnα*, and *Ifnγ* were significantly decreased in the CRD1 treatment group compared to the vehicle treatment group (Figs. 7I,J and EV4G). Consistent with this, qPCR showed intestinal VSV-P copy numbers was significantly suppressed by pretreatment with CRD1 peptide (Fig. 7K). To assess the antiviral activity of CRD1 to authentic SARS-CoV-2, we infected ex vivo intestinal tissues isolated from the K18-hACE2 mice with SARS-CoV-2–WT, SARS-CoV-2 Omicron, and SARS-CoV-2 XBB 1 h after treatment with CRD1 or carrier. Notably, the CRD1 treatment reduced the viral load, ameliorated the intestinal pathology, and decreased the levels of pro-inflammatory cytokines when compared to infected tissue treated with vehicle (Fig. 7L–P), consistent with the above data from the chimeric VSV-SARS-CoV-2 virus.

Intestine organoids generated from human iPSC were used to study the effects of CRD1 on SARS-CoV-2 infections. The VSV-P copy numbers was markedly increased in the organoid homogenates after VSV-SARS-CoV-2 infection, but this increase was significantly suppressed by pretreatment with CRD1 (Fig. 7Q). Further, pretreatment of these organoids with CRD1 decreased authentic SARS-CoV-2 infection at 24 h post-infection, as monitored by the levels of SARS-CoV-2 RNA using RT-qPCR (Fig. 7R).

Collectively, these findings indicate that LSR-derived CRD1 peptide act as potential inhibitor capable of efficiently blocking SARS-CoV-2 binding and reduced SARS-CoV-2 entry and infectivity in cell culture, organoid, and humanized mice.

## LSR inhibits SARS-CoV-2 Spike-induced syncytia formation

SARS-CoV-2-infected cells express Spike at their surface and fuse with ACE2-positive neighboring cells, then triggers multinucleated giant cells (syncytium) formation (Buchrieser et al, 2020). We then used a GFP-split complementation system to test whether LSR impact syncytia formation. GFP1-10 and GFP11 cells were co-cultured at 1:1 ratio and cell fusion were quantified by measuring the GFP⁺ area by high-content imaging after 24 h (Fig. 8A). Following LSR knockdown or LSR overexpression, the cells were subsequently infected with VSV-SARS-CoV-2. The results showed that knockdown of LSR significantly enhanced virus infection and Spike protein-mediated syncytium formation,

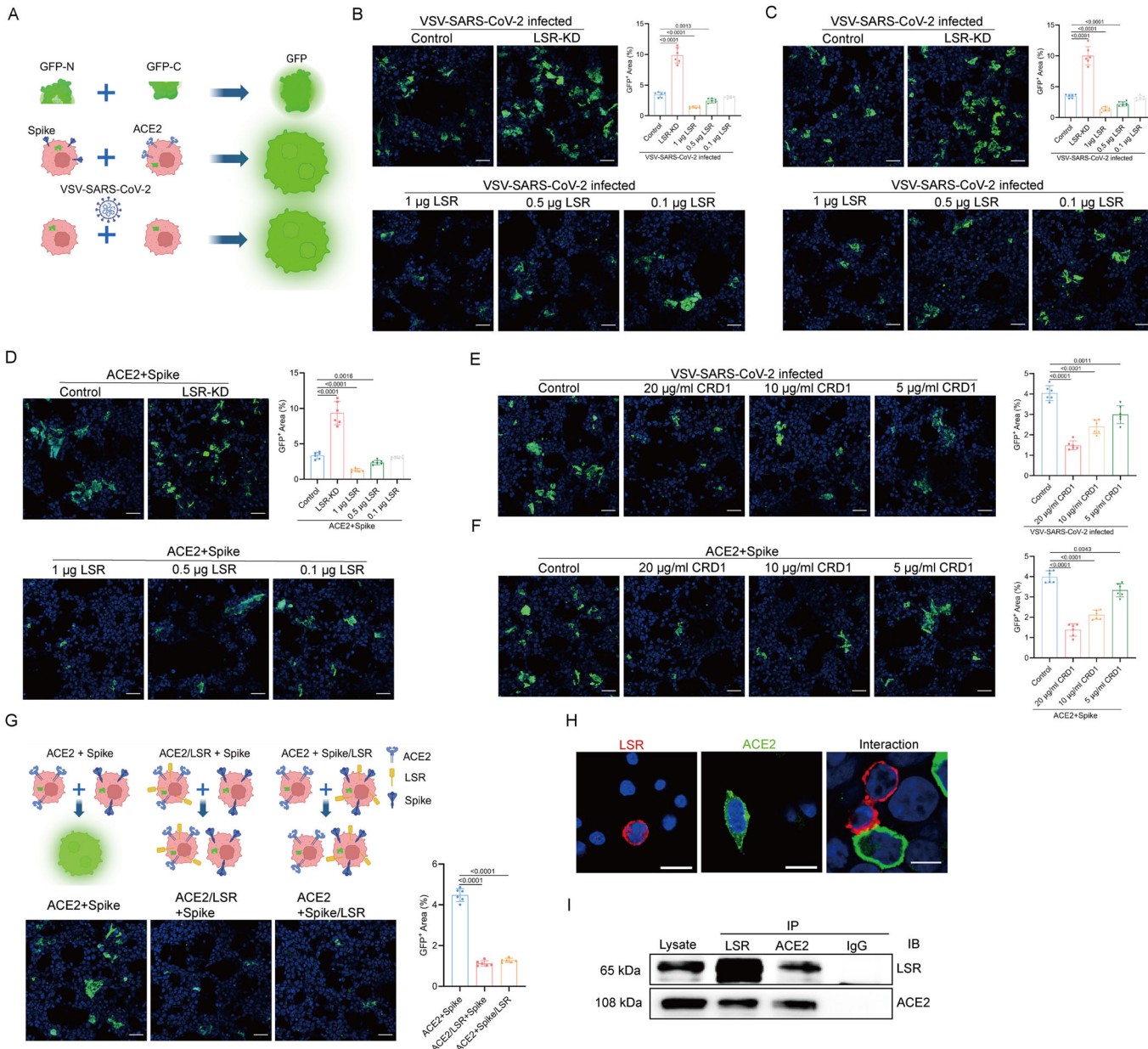

**Figure 8. LSR inhibits SARS-CoV-2 Spike-induced syncytia formation.**

(A) The schematic principle of the GFP-based BiFC assay. The extent of syncytia formation was quantified by measuring the GFP$^+$ area. The graphic was created with BioRender.com. (B, C) Impact of LSR on VSV-SARS-CoV-2-induced syncytia formation in Caco-2 cells (B) and hACE2-HIEC-6 cells (C) ($n = 6$). (D) Impact of LSR on syncytia formation induced by the interaction between ACE2 and Spike in HEK293 ($n = 6$). (E) Impact of LSR on syncytia formation induced by VSV-SARS-CoV-2 in Caco-2 cells ($n = 6$). (F) Impact of CRD1 on syncytia formation induced by the interaction between ACE2 and Spike in HEK293 cells. (G) Impact of LSR on syncytia formation on Spike- or ACE2-expressing HEK293 cells via analysis of the cocultured cells including ACE2-expressing cells cocultured with Spike-expressing cells, LSR transfected ACE2-expressing cells cocultured with Spike-expressing cells, and LSR transfected Spike-expressing cells cocultured with ACE2-expressing cells ($n = 6$). The graphic was created with BioRender.com. (H) Immunofluorescence analysis of ACE2 and LSR in the cocultured HEK293 cells expressing ACE2 or LSR individually. (I) Co-IP of LSR and ACE2 in singly transfected, cocultured HEK293 cells to determine the trans-interaction of LSR with ACE2. Antibodies used for immunoprecipitation are shown above the lanes; antibody for blot visualization is shown on the right. "$n$" represents number of biological replicates. Scale bars: (B), (C), (D), (E), (F), and (G), 50 μm; (H), 10 μm, Data represent mean ± SEM. Unpaired t test was performed. $p < 0.05$, the exact $p$-value is displayed; $p > 0.05$, the $p$-value is not displayed. Source data are available online for this figure.

while overexpression of LSR resulted in a significant decrease in syncytia formation following VSV-SARS-CoV-2 infection in both the Caco-2 (Fig. 8B) and hACE2-HIEC-6 cells (Fig. 8C).

To establish whether LSR blocked membrane fusion itself, we performed a syncytium assay in which cells expressing SARS-CoV-

2 Spike and the GFP1-10 fragment were co-cultured with cells expressing ACE2 and the GFP11 fragment (Fig. 8A). Notably, knockdown of LSR significantly augmented the formation of syncytia, whereas overexpression of LSR effectively suppressed syncytia formation in HEK293 cells (Fig. 8D). CRD1 can also

effectively inhibit syncytium formation and block virus-induced cell-to-cell fusion according to GFP signals and numbers of syncytia when compared to control groups (Fig. 8E,F). Thus, LSR directly prevents membrane fusion triggered by the SARS-CoV-2 Spike.

We next studied whether LSR impact cell–cell fusion by acting in ACE2-expressing cells, in Spike-expressing cells, or in both. To this end, LSR was transfected into either Spike-expressing cells or ACE2-expressing cells, and the transfected cells were then cocultured as shown in Fig. 8G. LSR inhibited fusion when present in the same cell as ACE2, and retained their ability to inhibit fusion when expressed in Spike-expressing cells (Fig. 8G). We speculated that LSR also have the capacity to interact with ACE2 in *trans* and the *trans*-binding of ACE2 by LSR also plays an essential role in the inhibited fusion. Co-cultures of HEK293 expressing LSR or ACE2 were examined using immunofluorescence co-localization and co-immunoprecipitation as assays to determine their *trans*-interaction. We found that LSR and ACE2 co-localized at areas of cell–cell contact (Fig. 8H) and formed a complex isolatable by co-immunoprecipitation (Fig. 8I), suggesting the existence of *trans*-interaction.

Together, these data delineate that interaction of LSR with ACE2 could take place both in *cis* and in *trans*, which can inhibit Spike-mediated cell–cell fusion.

## Discussion

LSR is a transmembrane protein that is expressed in multiple tissues, including liver, lung, heart, blood vessels, kidney, and intestine (Mesli et al, 2004). It functions as a lipoprotein receptor contributing to the liver uptake of triglycerides-rich lipoproteins (Narvekar et al, 2009; Yen et al, 2008), a critical component of the tricellular tight junctions that maintain epithelial and endothelial barrier integrity (Sohet et al, 2015; Sugawara et al, 2021), a host receptor for the binary toxin *Clostridium difficile* transferase (CDT) (Hemmasi et al, 2015; Papatheodorou et al, 2011), an upstream negative regulator of YAP signaling pathway which can modulate intestinal progenitor cell proliferation and Paneth cell differentiation (An et al, 2023), and either a tumor promoter or suppressor (Kohno et al, 2019). Here, we have identified LSR as a crucial host defense factor that may serve as target for antiviral therapies against SARS-CoV-2 infection in intestine.

Firstly, we show that both pseudotyped and authentic SARS-CoV-2 infection downregulates LSR in vivo in animal models, in cultured cells and 3D intestinal organoids in vitro, and in cultured intestinal organ ex vivo. Further, recombinant Spike protein treatment decreased the protein abundance of LSR, which could be due to the following reasons. After Spike interacts with ACE2 on target cells, the internalization of the ACE2 proteins on the cell surface and the interaction between LSR and ACE2 might send the LSR-ACE2 complex to endosome/lysosomes for degradation (Lu et al, 2022). LSR downregulation in infected cells is not only a specific effect induced by the Spike protein, but also a consequence of the translational shutoff triggered by the infection. Intriguingly, mRNA expression of LSR was also downregulated, suggesting that transcriptional/post-transcriptional mechanism may be involved in the LSR downregulation in COVID-19. Activation of the IFN signaling cascade represents a fundamental aspect of the innate

immune response following SARS-CoV-2 infection (Minkoff and tenOever, 2023). Spike protein itself can also activate IFN pathway (Liu et al, 2022; Ssali et al, 2023). Here, we found the inhibitory effects of some of IFNs on expression level of LSR mRNA. IFNs have been reported to suppress gene expression by epigenetic reprogramming, involving the formation and dismantling of enhancers and histone modifications that affect chromatin accessibility and the activity of enhancers and promoters (Barrat et al, 2019; Kang et al, 2017). The mechanisms of IFN-mediated suppression of LSR gene expression, however, require further investigation.

Next, we examined whether the reduced LSR expression contributes to disease progression and pathological outcomes upon SARS-CoV-2 infection. The importance of LSR in limiting virus infection and the therapeutic potential have been reinforced by studies performed in immortalized intestinal cell lines and fully differentiated intestinal organoid models. To validate the relevance of these findings in vivo, we assessed the effect of LSR in knock-in mice expressing the human ACE2 receptor driven by mouse ACE2 promoter (hACE2) and the epithelial cell K18 gene promoter (K18-hACE2), respectively, as models of SARS-CoV-2 infection. Then ex vivo intestinal tissues were infected with both pseudotyped and authentic SARS-CoV-2, viral RNA was detected in the intestine of infected animals by RT-qPCR, and the severity of disease was determined by histopathological changes and levels of pro-inflammatory cytokines and cellular infiltrate in the infected intestine of humanized ACE2 mice following infection. Our results show that hACE2 mice with homozygous intestine epithelial cell–specific loss of function of LSR are more susceptible to infection and have augmented inflammatory response. In contrast, administration of LSR to K18-hACE2 mice decreased their susceptibility to both pseudotyped and authentic SARS-CoV-2 infection. These results collectively suggest that LSR, which is constitutively expressed in intestinal epithelium and exhibits antiviral function, is one of the host factors providing intrinsic defense against virus invasion.

SARS-CoV-2 infection of human cells is initiated by the binding of the viral Spike protein to its cell-surface receptor ACE2 (Blume et al, 2021). ACE2, an 805-amino acid transmembrane protein with an extracellular N-terminal domain and an intracellular C-terminal tail, is a carboxypeptidase with several known physiological functions including regulation of blood pressure, salt and water balance in mammals, amino acid uptake in the small intestine, and glucose homeostasis and pancreatic β-cell function (Blume et al, 2021). In the present study, we revealed that LSR binds ACE2 and prevents it from binding Spike, thereby suppressing SARS-CoV-2 infection. *Cis* and *trans* interactions of LSR with ACE2 are directly observed via co-expressed LSR and ACE2 in the same cell or co-cultures of LSR expressing cells and ACE2 expressing cells. *Cis*-interaction impairs the Spike binding to ACE2 which could potentially play functional roles in inhibiting the entry of SARS-CoV-2. The Spike expressed on the surface of the infected cells interacts with the ACE2 receptor of neighboring cells and triggers the cell–cell fusion process, leading to large muti-nucleated syncytia (Ali et al, 2023). The downregulation of ACE2 using ACE2 blocking antibodies or ACE2-siRNA blunts syncytia formation in Spike-expressing cells (Braga et al, 2021). We next turned our attention to the role of LSR in SARS-CoV-2 syncytia formation and found that LSR inhibited cell–cell fusion triggered by the SARS-CoV-2 Spike

protein, which might be a result of the interaction of LSR with ACE2 occurring in *cis*, in *trans*, or simultaneously. As illustrated by the graphical model in Fig. EV5, LSR not only inhibits the entry of SARS-CoV-2 but also tapers Spike-induced syncytia formation by modulating the interaction between the Spike and ACE2. Strategies that effectively induce LSR expression and function could serve to improve existing anti-SARS-CoV-2 therapies. LSR plays a crucial role in the uptake of triglyceride-rich lipoproteins by binding to these lipoproteins and facilitating their internalization into cells (Mann et al, 1997). Thus, when considering LSR as a therapeutic target to inhibit SARS-CoV-2 infection, it is essential to consider the potential impact of targeting LSR on lipid uptake and metabolism.

The finding that LSR interacts with ACE2 is not entirely surprising. It has been shown that of the thousands of human transmembrane proteins, the top three 'Spike-like' proteins, featuring both high membrane-proximal cysteine and aromatic content, are transmembrane proteins found at tricellular tight junctions, including LSR, ILDR1, and ILDR2 (Sanders et al, 2021). Two cysteine-rich amino acid clusters exist in the membrane proximal region of both Spike and LSR. The CRD in the juxtamembrane region of the SARS-CoV Spike was found to be palmitoylated and is important for Spike-mediated cell–cell fusion and syncytia formation (Petit et al, 2007; Ramadan et al, 2022; Sanders et al, 2021; Wu et al, 2023), however, the detailed mechanism by which CRD participates in the Spike-mediated cell fusion still needs further investigation. Moreover, CRD of the SARS-CoV Spike has been proved to be able to bind sorting nexin 27 (SNX27) (Cattin-Ortola et al, 2021), that possesses a PDZ domain through which it could also interact with PDZ binding motif of ACE2 (Yang et al, 2022). In mammalian, cysteine residues in the membrane proximal region are common post-translational modification sites in transmembrane proteins, and these modifications are important for intermolecular interactions and assembly of protein complexes (Chappell et al, 2015; Wang et al, 2022; Xu et al, 2006; Zeronian et al, 2021). Here, despite the remarkable similarities between SARS-CoV-2 Spike protein and tricellular tight junction proteins with respect to the cysteine-rich motif, the membrane-proximal CRD1 of LSR is necessary for LSR binding to ACE2 and for the inhibitory effect on viral entry into host cells. Furthermore, our data showed that CRD1 treatment still inhibited viral infection when SARS-CoV-2 was already bound to cell receptors. However, at this time, we cannot conclude that LSR CRD1 has a higher binding affinity to ACE2, since we cannot rule out its role in other processes involving virus infection such as the transmission of offspring viruses or the virus-induced syncytia formation, and future comprehensive studies will be necessary to quantitatively assess the binding kinetics and affinities of these interactions. In contrast, CRD of the SARS-CoV Spike protein is not required for ACE2 receptor binding (Ramadan et al, 2022). This is expected, as ACE2 binding occurs via the RBD domain of the Spike and the CRD situated in or close to the viral membrane. Furthermore, previous genetic studies identified *Lsr* (*Lisch7*) as one of the candidate genes for salt-induced hypertension in the Sabra rat (Yagil et al, 2005), however, no further studies confirm its role in blood pressure regulation. Experimental hypertension studies have reported that deletion or inhibition of ACE2 leads to hypertension (Gurley et al, 2006), whilst enhancing ACE2 protects against the development of hypertension (Rentzsch et al, 2008; Yamazato et al, 2007). Thus, the interaction between LSR and ACE2 might also be involved in the regulation of blood pressure homeostasis.

The data obtained from Co-IP revealed interaction between the extracellular N-terminal region of ACE2 and the C-terminal region of LSR. C-terminal deletions of LSR resulted in loss of its binding to ACE2, and N-terminal deletions of LSR retained binding to ACE2. The results described above indicate that the N terminus of LSR protein should protrude into the cytoplasm of the cells and that the C terminus is located extracellularly. In contrast, previous work suggested that LSR is a type I transmembrane protein with extracellular N-terminus domain in mouse mammary epithelial cell line EpH4 and human endometrial cancer cell line Sawano cells (Oda et al, 2020; Saito et al, 2023). In this study, confocal immunofluorescence microscopy and membrane biotinylation of epitope-tagged N and C termini of LSR provide evidence of an extracellular C terminus and an intracellular N terminus in intestinal epithelial cells. These data are basically in support of the topological model predicted with Phobius (EMBL-EBI, https://www.ebi.ac.uk/) (Madeira et al, 2022). Furthermore, several motifs related to endocytosis were detected in full-length LSR, including a phosphorylation site NPGY that potentially represents a clathrin-binding site (Chen et al, 1990), as well as a dileucine lysosomal targeting signal (Santos et al, 2013). Both domains are located on the N-terminal side with respect to the putative transmembrane domain, also suggesting that the N-terminal was exposed intracellularly. However, future integrative approach will be required to further determine whether LSR protein sequence gives rise to different topologies in the membrane to exert different functions and how different combinations of topogenic determinants cooperate in a given membrane environment to finalize its membrane topology.

The continued identification of host factors that block virus entry is important for developing effective prevention as well as therapeutic strategies against viruses (Majdoul and Compton, 2022). Compared with traditional antivirals targeting viral components, these host-directed therapies may prove less likely to lead to drug resistance and to be highly promising strategies to prevent virus infection in the future (Majdoul and Compton, 2022). For SARS-CoV virus, blocking the binding of Spike to ACE2 appears as a rationale route to block viral entry. Herein, we have identified a small LSR-derived peptide that displays binding to ACE2 that can efficiently interfere with the virus Spike/ACE2 interaction, indicating this soluble fragment could potentially be optimized for therapeutic purposes. Treatment of cells, organoids, or humanized ACE2 mice with this peptide markedly reduced viral entry and block infection of cells by pseudo-SARS-CoV-2 virus, as well as authentic SARS-CoV-2. Given their potent inhibition of viral activity and lack of cytotoxicity, these peptides provide an attractive avenue for the development of new prophylactic and therapeutic agents against SARS-CoV-2. Future studies to further optimize the soluble receptor fragments may lead to an effective therapy for clinical management of SARS-CoV-2 infection.

The data presented in this work identify LSR as a therapeutic target in the management of COVID-19 and open up new avenues for blocking the binding of Spike to ACE2 through LSR for the prevention of infection with SARS-CoV-2 as well as other viruses that use ACE2 for cell entry. However, we acknowledge several limitations of the present study. First, although we roughly map an interaction between the CRD of LSR and N-terminal region of ACE2, we were unable to 3D model binding of LSR to the candidate sites present on ACE2 and how such attachment might relate to the

binding of the trimeric Spike to ACE2. A high-resolution structural characterization of this interaction is lacking. Second, LSR have the capacity to interact with ACE2 in both *cis* and *trans*, it is not known whether they are using the identical binding sites or adopting two distinct conformations to mediate *cis–trans* recognition. Last, we show that this mechanism applies in SARS-CoV-2-affected gastrointestinal epithelia. LSR is present not only in the gastro-intestinal tract, but also in other COVID-19-relevant cell types such as the lungs, the liver, and the endothelium et al. Thus, whether regulation of Spike and ACE2 interaction through LSR represents a general mechanism, extending beyond intestinal epithelial cells, are important questions that need to be addressed in future studies.

# Methods

## Experimental model and subject details

### Mice

C57BL/6J mice were obtained from Vital River Laboratory (China). The *Lsr*^loxP/loxP^/villin-Cre^+/−^ mice were generated as previously described (An et al, 2023). hACE2 mice (Strain NO. T037659) and K18-hACE2 mice (Strain NO. T037657) were purchased from GemPharmatech (Nanjing, China). All animal studies were conducted in accordance with the recommendations in the Guide for the Care and Use of Laboratory Animals of the National Institutes of Health. The protocols were approved by the Institutional Animal Care and Use Committee at Binzhou Medical University (protocol number 2021-011). Mice were housed in a pathogen-free, climate-controlled facility and given food and water ad libitum. Female and male mice at 6 to 8 weeks old were used for experiments unless otherwise specified.

### Patient sample

The intestinal specimens of patients with COVID-19 were kindly provided by Dr. Xiu-Wu Bian from Third Military Medical University. As controls, intestinal tissues were collected from patients with healthy tissue margins resected for indications intestinal obstruction or spontaneous intestinal perforation. The study was performed in accordance with the Declaration of Helsinki and conducted with the approval of the medical Ethics Committee of Binzhou Medical University.

### Cell lines

Vero E6, HEK293, A549 cells, Caco-2, and HIEC-6 cell lines were obtained from ATCC (USA) and iPSCs were obtained from National Collection of Authenticated Cell Cultures (China), and cultured according to the distributor's recommendations.

### Antibodies

The antibodies used in this study are summarized in Table EV1.

### Reagents

The reagents used in this study are summarized in Table EV2.

### Virus strains

Chimeric VSV virus where the glycoprotein of VSV was replaced with the Spike protein of SARS-CoV-2 (VSV-SARS-CoV-2) was developed by Aihua Zheng and colleagues (Li et al, 2020) and

propagated in Vero E6 cells. High-titer stocks of authentic SARS-CoV-2 virus (isolate WT, Omicron BA.5.2, and XBB) were obtained by passage in Vero E6 cells. Pseudotyped replication-defective lentiviruses bearing the SARS-CoV-2 Spike glycoprotein was obtained from Genomeditech (China). Viral titers were determined by plaque assay on Vero E6 cells. All work with infectious virus (propagation, titration, and plaque reduction assays) was done in the biosafety level 3 (BSL3) facility at Shandong Provincial Center for Disease Control and Prevention (CDC).

## Method details

### Cells overexpressing ACE2 or LSR

The coding sequences of full-length human ACE2 gene (NM_001371415) and LSR gene (NM_205834.4) were cloned into the retroviral vector pQCXIP (Clontech), respectively. Retroviruses expressing human ACE2 were used to infect HEK293 cells, A549 cells, and HIEC-6 cells to generate hACE2-293 cells, hACE2-A549 cells, and hACE2-HIEC-6 cells. Retroviruses expressing human LSR were used to infect the aforementioned original cell lines. Cells were subsequently selected with 5 μg/ml puromycin for 14 days before further treatments.

### Generation of LSR knockdown cell lines

The siRNA hairpin oligonucleotides (TTTGAAGGAACACT-GATGA), complementary to all the six variants of human LSR mRNA sequences (NM_015925.7, NM_205834.4, NM_205835.4, NM_001260489.2, NM_001260490.2, and NM_001385215.1), were synthesized by Sangon Biotechnology Inc., annealed, and cloned into the retrovirus vector pSIREN (Clontech) to create the LSR shRNA constructs. Retroviruses were used to infect the hACE2-293, Caco-2, hACE2-A549 cells, and hACE2-HIEC6. Cells were subsequently selected with 5 μg/ml puromycin for 14 days before further treatments.

### Construction of LSR-OE and LSR-KD intestinal organoid using PiggyBac transposon system

Full-length LSR gene (NM_205834.4) and siRNA hairpin oligonu-cleotides as described above were de novo synthesized into the PiggyBac transposon expression plasmids under control of the CMV promoter (VectorBuilder Inc., China). The plasmids also contained the puromycin resistant gene to facilitate colony selection and the Tet-On element to control the timing of expression of LSR. Human iPSCs were seeded in a 6-well plate at the density of $6 \times 10^5$ cells/well and then co-transfected with the constructed transposon plasmids along with the helper plasmid encoding hyperactive PiggyBac transposase system by using Lipofectamine 3000. The transfected cells were maintained in growth media supplemented with 5 μg/ml puromycin for 14 days before further treatments. Intestinal organoids were generated according to our previous report (An et al, 2023). To investigate the effects of LSR, mature intestinal organoids were treated with 1 μg/ml doxycycline for 48 h to induce the transgene expression. The WT organoids were subjected to the same doxycycline treatment as the experimental groups.

### In vitro viral infection

Cells were infected with VSV-SARS-CoV-2 or SARS-CoV-2 at 1 MOI for 24 h, and harvested for RNA isolation, western blot, and

immunofluorescence staining analysis. An approach that has been demonstrated to facilitate productive virus infection in organoids was employed for virus infection experiments in organoids (Dutta et al, 2017; Hashimi et al, 2023). Cell recovery solution was used to harvest organoids and mechanically dissociated in basal media using a narrowed Pasteur pipette for 5–7 times up and down to obtain uniformly sized organoid fragments prior to virus infection. The organoid fragments were centrifuged at 4 °C for 5 min at $250 \times g$, resuspend with 400 μl full growth medium containing $2 \times 10^6$ p.f.u. VSV-SARS-CoV-2 or SARS-CoV-2 (WT/Omicron/XBB), and then mixed with 100 μl of matrigel. The mixture was incubated at 37 °C for 24 h, mixing every 30 min in the first 6 h. Later, the mixture was centrifuged at 4 °C for 5 min at $250 \times g$ and harvested for RNA isolation and immunofluorescence analysis.

### Ex vivo viral infection

As previously mentioned (Udden et al, 2017), the freshly obtained intestinal tissues from 7-week-old hACE2-$Lsr^{\text{vill ctrl}}$ (WT), hACE2-$Lsr^{\text{vill KO}}$ ($Lsr^{\text{vill KO}}$), and K18-hACE2 mice were processed into small rectangular pieces and cultured in DMEM/F12 medium containing 5% FBS, 1% Penicillin-Streptomycin, and Gentamycin (20 μg/ml) at 37 °C and 5% $CO_2$. The specimens were infected with $2 \times 10^6$ p.f.u. VSV-SARS-CoV-2 or SARS-CoV-2 for 6 h. The tissues were harvested for RNA isolation and histologic analysis.

### In vivo viral infection

An infection approach known to recapitulate the pathological alterations observed in the intestines of COVID-19 patients was employed to generate the mouse model of viral infection (Zeng et al, 2022). 7- to 9-week-old hACE2-$Lsr^{\text{vill ctrl}}$ (WT), hACE2-$Lsr^{\text{vill KO}}$ ($Lsr^{\text{vill KO}}$), K18-hACE2, and K18-hACE2 with LSR overexpression mice were starved for 24 h and then received a 0.5 ml of 1% (v/v) acetic acid enema for 2 min using the tip of the catheter positioned 5 cm proximal to the anus. After acid challenge, mice were given 100 μl control-media or VSV-SARS-CoV-2 ($2 \times 10^8$ p.f.u./ml) intraperitoneally. Six hours later, intestinal tissues were harvested for RT-qPCR, western blot, histological staining, and immunofluorescence staining analysis.

### The effect of Spike on LSR expression

For the Spike protein treatment studies, Caco-2, HIEC-6, and hACE2-HIEC-6 were incubated with 0, 1, and 2 μg/ml recombinant SARS-CoV-2 Spike protein for 24 h and LSR levels were measured by western blot. For the Spike protein plasmid transfection studies, Caco-2 in 6-well plates were transiently transfected with 0, 0.4, 0.8, and 1.6 μg pCMV3-SARS-CoV-2-Spike vector kindly provided by Dr. Peihui Wang (Shandong University, China) using Lipofecta-mine 3000 for 48 h and then subjected to western blot with anti-LSR antibody. For the proteasome and lysosome inhibition studies, Caco-2 cells were pretreated with 10 μM proteasome inhibitor MG132 or 200 nM lysosome inhibitor Baf-A1 for 2 h, followed by 2 μg/ml of recombinant SARS-CoV-2 Spike protein treatment for 0, 3, 6, and 12 h in the presence of 100 μg/ml CHX and LSR proteins in the cell lysates were examined by western blot. For the VSV-SARS-CoV-2 infection study, Caco-2 cells were pretreated with 200 nM Baf-A1 for 2 h and then infected with VSV-SARS-CoV-2 for 0, 3, 6, and 12 h. LSR protein levels in the cell lysates were examined by western blotting.

### The effect of VSV M protein, NSP1, and NSP14 on LSR expression

Caco-2 cells in 6-well plates were transfected with either the pCAG-VSV-M vector (MiaoLingBio, China) or the pCAG-SARS-CoV-2-NSP1 and pCAG-SARS-CoV-2-NSP14 vectors (kindly provided by Dr. Peihui Wang, Shandong University, China) using Lipofecta-mine 3000 for 48 h. After transfection, cell lysates were then subjected to western blot analysis using an anti-LSR antibody. In addition, Caco-2 cells in 6-well plates were co-transfected with the pCAG-VSV-M vector, the pCAG-SARS-CoV-2-NSP1 vector, or the pCAG-SARS-CoV-2-NSP14 vector, along with 0.5, 1.5, or 3 μg of the pCMV3-SARS-CoV-2-Spike vector using Lipofectamine 3000 for 48 h. Subsequently, cell lysates were subjected to western blot analysis with an anti-LSR antibody.

### The effect of IFNs on LSR expression

Caco-2 cells monolayer cultures were stimulated with 100 IU ml$^{-1}$ or 1000 IU ml$^{-1}$ of recombinant human IFN-α 2a, IFN-β, IFN-γ, or IFN-λ protein for 24 h and LSR mRNA levels were quantified using RT-qPCR.

### Virus binding and internalization assay

For the virus binding assay, Caco-2, HEK293, and hACE2-293 cells were pre-chilled to 4 °C for 15 min, followed by incubation with VSV-SARS-CoV-2, pseudotyped virus bearing SARS-CoV-2 Spike, or authentic SARS-CoV-2 at 1 MOI at 4 °C for 1 h. Unbound viral particles were removed by washing with pre-chilled PBS three times. For inhibitor assays in Caco-2, the cells were seeded at $2.5 \times 10^5$ cells per well in 12-well plates. The next day, cells were treated with 200 mU/ml neuraminidase V or 300 μg/ml heparin sulfate at 37 °C for 1 h, then infected with VSV-SARS-CoV-2 at 1 MOI at 4 °C for 1 h. Unattached viruses were removed through washing three times with cold PBS, and viral RNA copies of binding viruses were measured by RT-qPCR. For virus internalization assay, cells were incubated with SARS-CoV-2 using the same condition described above. Cells were then transferred to 37 °C for 1 h to allow internalization of bound virus. Uninternalized viral particles were removed by washing three times with acidic buffer (50 mM glycine and 100 mM NaCl, pH 3.0) and treating with 0.25% trypsin for 15 min at 4 °C.

### The effect of LSR on viral entry

The wild type, LSR-KD, and LSR-OE Caco-2 and hACE2-293 cells were seeded at $2.5 \times 10^5$ cells per well in 12-well plates. The next day, cells were treated with protease inhibitors at the following concentrations or dilutions: DMSO (1:200), 25 μM E-64d, and 20 μM Camostat. Three hours later, Caco-2 and hACE2-293 cells were infected with SARS-CoV-2 at 1 MOI for 24 h and cells were collected to quantify SARS-CoV-2 infection by RT-qPCR.

### The effect of LSR on viral replication

Caco-2 cells were seeded at $2.5 \times 10^5$ cells per well in 12-well plates. The next day, cells were treated with DMSO (1:200) or 3.7 μM remdesivir for 1 h and infected with SARS-CoV-2 at 1 MOI for 1 h, then transfected with retrovirus carrying LSR shRNA, human LSR transcript variant 2, or control plasmid. Remdesivir were kept in the medium during infection. Twenty-four hours post-infection, cells were collected to quantify SARS-CoV-2 infection by RT-qPCR.

### Viral RNA extraction and RT-qPCR

At indicated time points, total RNA was extracted using Trizol reagent from infected cells, organoids, and intestinal tissues. Extracted RNA was reverse transcribed to cDNA using PrimeScript™ RT reagent Kit with gDNA Eraser. SARS-CoV-2 (WT, Omicron and XBB variant) production was quantified using primers specific to the SARS-CoV-2 N gene (Table EV3). VSV-SARS-CoV-2 was quantified by using primers (Table EV3) specific to VSV-P or Spike. All viral mRNA levels were normalized to β-actin.

### SARS-CoV-2 replication kinetics in Caco-2 cells

The kinetics of viral replication in Caco-2 cells were assessed over 48 h post-infection. For this purpose, Caco-2 cells were seeded at a density of $3 \times 10^5$ cells per well in 12-well plates. The next day, cells were infected with SARS-CoV-2 at an MOI of 1 and cultured at 37 °C for 1 h. Afterward, the inoculum was removed and replaced with fresh medium (DMEM/F12 + 5% FBS), and the cells were incubated at 37 °C. Cell lysates were collected at 1, 4, 8, 24, 36, and 48 h post-infection to quantify viral RNA copy numbers by RT-qPCR. Supernatants from infected cells were collected and assessed for the production of de novo infectious viral particles using TCID$_{50}$ assay on naïve Vero cells.

### TCID$_{50}$ assay

The release of infectious viral particles into the supernatant was assessed using Vero E6 cells cultured in 96-well plates. A tenfold dilution series from each sample was used to infect five independent wells per dilution for the determination of inoculation titers. The cytopathic effect (CPE) was measured by crystal violet staining at day 7 post-infection, and each well was scored as either positive or negative for virus infection. The Spearman-Kärber method was used to calculate the results, which are presented as TCID$_{50}$/ml.

### In vivo administration of recombinant AAV

AAVrh10 has been proved to be the most efficient rAAV serotype for intestine gene delivery (Polyak et al, 2012). 7-week-old K18-hACE2 mice received with $1 \times 10^8$ p.f.u. virus genomic particles of Rh10-pHBAAV-CMV-MCS-3flag-T2A-ZsGreen (WT) or Rh10-pHBAAV-CMV-Lsr-3flag-T2A-ZsGreen (LSR-OE) (prepared by Hanbio Biotechnology Inc, China) by intraperitoneal injection ($n = 8$). Three weeks after transduction, mice were infected with VSV-SARS-CoV-2 as described above. Small intestinal tissues were harvested for RNA isolation and immunofluorescent and histological staining.

### CRD1 peptide treatment

The LSR-derived CRD1 peptide was synthesized by ChinaPeptide (Shanghai, China) and dissolved in PBS. In vitro: Caco-2 and hACE2-HIEC-6 cells were seeded at 60–70% confluency in each well of the plates and virus was diluted into media containing the desired concentration of CRD1 peptide. At 24 h post-infection, supernatant was removed, and cells collected to quantify SARS-CoV-2 infection by RT-qPCR as described above. In vivo: Intraperitoneal administration of CRD1 peptide in K18-hACE2 mice was performed on 1, 2, and 3 days before infection (50 mg per kg each time), followed by virus challenge. CRD1 peptide was delivered using PBS as vehicle. Mice were killed at 6 h after infection for virological and histopathological analyses. Viral yield in the intestinal tissue homogenates was detected by RT–qPCR.

Intestinal tissues obtained from K18-hACE2 mice and organoids were primed with 10 μg/ml CRD1 peptide. One hour later, tissues and organoids were infected with SARS-CoV-2 at 1 MOI for 6 or 24 h to allow effective viral infection. CRD1 peptide was kept in the medium during infection. The tissues and organoids were harvested for RNA isolation and histologic analysis.

### Time of CRD1 addition assay

Caco-2 and hACE2-HIEC-6 cells were seeded in 24-well plates. The next day, 10 μg/ml CRD1 peptide was added to the cells 1 h prior to infection, during infection (0 h), and 1 h post infection. Then the cells were infected with 1 MOI SARS-CoV-2. 1 h after virus inoculation at 37 °C, the medium was replaced with fresh medium devoid of CRD1 or virus. Twenty-four hours post-infection, cells were collected to quantify SARS-CoV-2 infection under different exposure conditions using RT-qPCR. To explore the role of LSR on viral infection when SARS-CoV-2 was bound to cell receptors, Caco-2 cells were infected with VSV-SARS-CoV-2 at 1 MOI at 4 °C for 1 h. The cells were then washed twice to remove unbound viruses and treated with either PBS or 10 μg/ml CRD1 for 24 h. Following the 24-h incubation period at 37 °C after CRD1 addition, cells were collected to quantify SARS-CoV-2 infection under different exposure conditions using RT-qPCR.

### Colocalization assay of CRD1 peptide binding to ACE2 in cells

CRD1 peptide was labeled with the FITC conjugation kit according to the manufacture instructions. hACE2-293 cells were treated with 10 μg/ml FITC-labeled CRD1 peptide at 37 °C for 6 h. Cells were washed twice by PBS and then fixed by 4% paraformaldehyde for 30 min. After washes, the cells were incubated with anti-ACE2 for immunofluorescence staining.

### Cell viability assays

Caco-2 and hACE2-HIEC-6 cells were seeded at $2 \times 10^4$ cells per well in 96-well plates. The following day, CRD1 was added to cells and incubated at 37 °C for 24 h. For CCK8 assays, medium was removed from the cells and replaced by CCK8 reagent diluted in medium and incubated at 37 °C for 1 h. Cell culture wells without peptides were used as the experimental control and medium only served as a blank control. The absorbance was measured with a BMG Labtech microplate reader.

### Cell–cell fusion assays

The coding sequences of GFP1-10 and GFP11 (Kamiyama et al, 2016) were synthesized by Sangon Biotechnology Inc., and cloned into the retroviral vector pQCXIP, respectively. The GFP1-10 or GFP11 were transfected into the human cells Caco-2, HEK293, and hACE2-HIEC-6 using a retroviral delivery system. Cells were subsequently selected with 5 μg/ml puromycin for 14 days before further treatments.

For cell–cell fusion induced by virus, GFP1-10 and GFP11 cells were seeded on coverslips ($1.6 \times 10^5$ cells/well, mixed at a 1:1 ratio) in 24-well plates. The next day, the cells were transfected with plasmids carrying LSR shRNA, the indicated amounts of plasmids carrying human LSR transcript variant 2 (0.1, 0.5, and 1 μg), or control plasmids. Twenty-four hours later, the cells were infected with VSV-SARS-CoV-2 at 1 MOI for 24 h. For CRD1 treatment experiment, the cells were treated with indicated amounts of CRD1 peptide 1 h before infection.

For cell–cell fusion induced by the interaction between ACE2 and Spike, GFP1-10 cell were transfected with pCMV3-SARS-CoV-2-Spike (Spike-GFP1-10 cells), and GFP11 cells were transfected with pQCXIP-ACE2 (hACE2-GFP11 cells). Spike-GFP1-10 cells and hACE2-GFP11 cells were seeded on coverslips at a 1:1 ratio in 24-well plates. The next day, the cells were then transfected with plasmids carrying LSR shRNA, the indicated amounts of plasmids carrying human LSR transcript variant 2 (0.1, 0.5, and 1 μg), or control plasmid. For CRD1 treatment experiment, the cells were treated with indicated amounts of CRD1 peptide 1 h before transfection.

To overexpress LSR in Spike-expressing cells, GFP1-10 HEK293 were co-transfected with pCMV3-SARS-CoV-2-Spike and pQCXIP-LSR-V2, and GFP11 cells were transfected with pQCXIP-ACE2. To overexpress LSR in ACE2-expressing cells, GFP1-10 cells were transfected with pCMV3-SARS-CoV-2-Spike, and GFP11 cells were co-transfected with pQCXIP-ACE2 and pQCXIP-LSR-V2. Twenty-four hours post-transfection, GFP1-10 HEK293 cells and GFP11 HEK293 cells were seeded on coverslips ($1.6 \times 10^5$ cells/well, cells mixed at a 1:1 ratio) in 24-well plates and cultured for 24 h.

Forty-eight hours post-transfection, nuclei were stained with DAPI, and the cell confluence area was analyzed using a confocal laser microscope (LSM 880 Basic Operation, Germany). The GFP area was quantified on ImageJ.

### Histologic analysis

Mouse intestine tissues were fixed overnight at 4 °C in 4% paraformaldehyde and embedded in paraffin. Intestine sections (5 μm thickness) were prepared by a routine procedure, then standard histological staining with H&E. Transmission electron microscopy (TEM) was performed on glutaraldehyde-fixed, epoxy-embedded intestine samples and stained with uranyl acetate and lead citrate.

### mRNA extraction and RT-qPCR

Total mRNA was isolated from cells, tissues, and organoids using a standard Trizol RNA extraction protocol. RNA was reversed transcribed using PrimeScript™ RT reagent Kit with gDNA Eraser. RT-qPCR was performed using SYBR Green PCR Master Mix. RT-qPCR was performed on the Applied Biosystems QuantStudio 3 Real-Time PCR System. The Table EV3 contains the primer sequences used in this study. The expression levels of each mRNA were calculated after normalizing to those of β-actin. Results were expressed as $2^{-\Delta Ct}$ values with $\Delta CT = Ct_{gene} - Ct_{\beta-actin}$.

### Immunofluorescence staining

For immunofluorescence of small intestine tissue, 8-μm frozen sections were fixed in ice-cold methanol or acetone for 20 min at −20 °C. For immunofluorescence of human organoid, organoid on collagen I coated confocal dish were fixed with 4% paraformaldehyde for 30 min at room temperature. For human paraffin section (5 μm thickness), immunostaining was preceded by antigen retrieval in EDTA buffer (pH 8.0). For immunofluorescence of cultured cells, cells were fixed with 4% paraformaldehyde for 20 min at room temperature. The tissue and cell samples were permeabilized with 0.01% Triton X-100 for 5 min and the organoids were permeabilized with 0.3% Triton X-100 for 30 min, followed by blocking with PBS containing 10% FBS and 0.5% Tween 20, and incubation with primary antibodies and FITC- or rhodamine-labeled secondary antibodies. After washing with PBS, slides were mounted with ProLong™ Gold Antifade Mountant. Immunofluorescence images were collected and processed by Zen Software v2.3 (blue edition).

For membrane-impermeant immunostaining, cells were fixed with 4% paraformaldehyde for 20 min at room temperature and blocked with 1% BSA in PBS for 2 h at 4 °C. The nonpermeabilized cells were incubated for 2 h at room temperature with primary antibody and for 1 h at room temperature with secondary antibody. The cell monolayers were washed three times with PBS buffer after each incubation with antibody.

For in vivo propidium iodide staining, mice received an intravenous injection of 10 mg/kg propidium iodide 30 min before euthanasia to label dead cells. Small intestinal tissues were then harvested for immunofluorescent staining. For intestinal organoid propidium iodide staining, organoids were fixed with 4% paraformaldehyde for 30 min at room temperature and then incubated with 1 mg/ml propidium iodide for 30 min at room temperature. Subsequent experiments were conducted according to the aforementioned method for staining.

The number of N-protein-positive cells and the mean fluorescence intensity of N-protein in infected cells were quantified in 10 fields of view for each group ($n = 6$) using Image J software.

### Western blot analysis

Total protein extracts were obtained by lysing the cells or small intestine tissues in RIPA lysis buffer containing PMSF and phosphatase inhibitors on ice, supernatants were collected after centrifugation at 4 °C for 10 min at $12,000 \times g$. Then, the protein concentration was assessed using a BCA protein assay. Proteins were separated in 10% SDS-PAGE and then transferred onto PVDF membranes (0.22 μm), and then blocked with 5% skim milk dissolved in TBST and incubated with the primary antibody at 4 °C overnight. HRP-conjugated secondary antibodies followed by ECL incubation allowed protein band detection. The integration of all blots images was performed on Adobe Illustrator 2021. Image J v1.8.0 was used to quantify western blot results.

### Biotinylated ACE2 pulldown assay

Confluent Caco-2 cells were incubated with 0.25 mg/ml Sulfo-NHS-SS-Biotin in 1x PBS on ice for 30 min. 50 mM Tris (pH 8.0) buffer was used to stop the biotinylation reaction followed by centrifugation at $5000 \times g$. The pellet was dissolved in CSK buffer (150 mM NaCl; 1% Triton X-100; 50 mM Tris, pH 8.0; and protease inhibitors) to extract membrane proteins. The membrane extract was incubated with NeutrAvidin Agarose to bind biotinylated proteins. The NeutrAvidin bound biotinylated proteins were eluted using the SDS sample buffer containing β-mercaptoethanol, followed by SDS-PAGE under denaturing conditions and western blot to analyze the expression of ACE2.

### Biotinylated LSR pulldown assay

The N terminus and C terminus of LSR were fused to the Flag and cloned into eukaryotic expression vector pcDNA 3.1 to generate the two Flag-fused constructs, which were designated pcDNA-Flag-LSR and pcDNA-LSR-Flag, respectively. HEK293 cells seeded into a 15-cm dish were transfected with pcDNA-Flag-LSR or pcDNA-LSR-Flag for 72 h, and cells were incubated with 0.25 mg/ml Sulfo-NHS-SS-Biotin in 1x PBS on ice for 30 min. Membrane proteins were solubilized in 100 mM $NH_4HCO_3$ buffer (pH 8.0) containing 0.1%

(w/v) Rapigest surfactant, 10% acetonitrile, 1 mM iodoacetamide and 1 mM 2,2′-thiodiethanol. Solubilization was assisted by brief pulses of sonication followed by incubation on ice for 30 min. The suspension was incubated with 500 units of PNGaseF for 2 h at 37 °C and then adding trypsin in a 1:50 (w/w) and kept in 37 °C for 16 h. Digestion was stopped by heat inactivation (95 °C for 10 min) followed by the addition of 100 μM TLCK to the reaction mixture. The biotinylated proteins were precipitated with streptavidin-agarose beads and further detected using western blot to analyze the expression of LSR.

### Co-immunoprecipitation (Co-IP)

The coding sequence of all the six variants of human LSR (NM_015925.7, NM_205834.4, NM_205835.4, NM_001260489.2, NM_001260490.2, and NM_001385215.1) were synthesized by Sangon Biotechnology Inc and cloned into pQCXIP vector (pQCXIP-LSR-V1-V6). The six cysteine residues in CRD were replaced by alanine to generate the CRD1 mutant (pQCXIP-LSR-CRD1M) and CRD2 mutant (pQCXIP-LSR-CRD2M) with a PCR-based mutagenesis method using pQCXIP-LSR-V2 as a template. Human LSR isoform 2 with N-terminal deletion (pQCXIP-LSR-N-del) or C-terminal deletion (pQCXIP-LSR-C-del) were amplified by PCR using pQCXIP-LSR-V2 as a template and cloned into pQCXIP vector. Short ACE2 protein isoform (nucleotides encoding amino acids 357–805 of the full-length ACE2) were synthesized by Sangon Biotechnology Inc and cloned into pQCXIP vector (pQCXIP-sACE2).

For protein *cis*-interaction assay, HEK293 cells were doubly or triply co-transfected with the following plasmids: pQCXIP-LSR-V1 and pQCXIP-ACE2; pQCXIP-LSR-V2 and pQCXIP-ACE2; pQCXIP-LSR-V3 and pQCXIP-ACE2; pQCXIP-LSR-V4 and pQCXIP-ACE2; pQCXIP-LSR-V5 and pQCXIP-ACE2; pQCXIP-LSR-V6 and pQCXIP-ACE2; pQCXIP-LSR-CRD1M and pQCXIP-ACE2; pQCXIP-LSR-CRD2M and pQCXIP-ACE2; pQCXIP-LSR-V2 and pQCXIP-sACE2; pQCXIP-LSR-V2, pQCXIP-ACE2, and pCMV3-SARS-CoV-2-Spike; pQCXIP-LSR-CRD1M, pQCXIP-ACE2, and pCMV3-SARS-CoV-2-Spike; pQCXIP-LSR-N-del and pQCXIP-ACE2; pQCXIP-LSR-C-del and pQCXIP-ACE2, respectively. The influence of CRD1 on the Spike-ACE2 interaction was measured by incubating the pQCXIP-LSR-V2 and pQCXIP-ACE2 transfected HEK293 cells with indicated concentrations CRD1 peptide for 12 h. For protein *trans*-interaction assay, two HEK293 cells transfectants singly expressing LSR or ACE2 were cocultured. Then, cells were lysed in 50 mM Tris (pH 8.0) by 25–30 repeated passages through a 25-gauge needle, followed by centrifugation at $5000 \times g$. The membranes of lysed cells were extracted using CSK buffer (150 mM NaCl; 1% Triton X-100; 50 mM Tris, pH 8.0; and protease inhibitors). The membrane extract was precleared by incubation with protein A/G-sepharose prior to co-immunoprecipitation. The precleared membrane extract was incubated for 16 h at 4 °C with anti-LSR, anti-ACE2, or anti-Spike antibodies. Antibody-bound material was pelleted with protein A/G-sepharose, washed 3 times with CSK buffer and subjected to SDS-PAGE, followed by western blot analysis.

### Statistical analysis

The significance of differences between groups was tested by Prism 8 (GraphPad Software Inc.). Statistical analysis was performed using the unpaired *t* test to determine differences between two groups and ANOVA to compare data among groups. *P*-values less than 0.05 were interpreted as statistically significant. All data are presented as mean ± SEM and other details such as the number of replicates and the level of significance is mentioned in figure legends and supplementary information.

## Data availability

All data are available in the main text or the supplementary materials. This study includes no data deposited in external repositories.

The source data of this paper are collected in the following database record: biostudies:S-SCDT-10_1038-S44318-024-00281-4.

## Peer review information

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

## Acknowledgements

We thank Dr. Xiu-Wu Bian from Third Military Medical University for proving the intestine specimens of patients with COVID-19. We thank Dr. Peihui Wang from Shandong University for proving the expression plasmids of Spike, NSP1, and NSP14. We thank Dr. Aihua Zheng from Chinese Academy of Sciences for proving VSV-SARS-CoV-2, VSV-Luciferase-SARS-CoV-2, and VSV-eGFP-SARS-CoV-2. We thank Ping Li for animal husbandry. We thank the electron microscopic core lab of Shandong University for assistance on electron microscopy imaging. This work was funded by the National Natural Science Foundation of China (81670620, 81870485, 81970578, 32371172, 82370706, and 32200758), Taishan Scholars Program of Shandong Province (ts20190953), Natural Science Foundation of Shandong Province (ZR2021QH115 and ZR2021QC105), Shandong First Medical University Academic Promotion Program (2020LI001), and the Major Basic Research Program of the Shandong Province Natural Science Foundation (ZR202206080014).

## Author contributions

**Yanan An**: Conceptualization; Formal analysis; Funding acquisition; Validation; Investigation; Methodology; Writing—original draft. **Chao Wang**: Conceptualization; Formal analysis; Funding acquisition; Validation; Investigation; Methodology; Writing—original draft. **Ziqi Wang**: Formal analysis; Validation; Investigation; Methodology; Writing—original draft. **Feng Kong**: Formal analysis; Validation; Investigation; Methodology; Writing—original draft. **Hao Liu**: Investigation. **Min Jiang**: Validation; Investigation. **Ti Liu**: Conceptualization; Resources. **Shu Zhang**: Validation; Investigation. **Kaige Du**: Validation; Investigation. **Liang Yin**: Resources. **Peng Jiao**: Investigation; Methodology. **Ying Li**: Investigation. **Baozhen Fan**: Investigation. **Chengjun Zhou**: Formal analysis. **Mingxia Wang**: Investigation. **Hui Sun**: Investigation; Methodology. **Jie Lei**: Conceptualization; Resources; Supervision; Funding acquisition; Project administration. **Shengtian Zhao**: Conceptualization; Resources; Supervision; Funding acquisition; Project administration; Writing—review and editing. **Yongfeng Gong**: Conceptualization; Resources; Supervision; Funding acquisition; Project administration; Writing—review and editing.

Source data underlying figure panels in this paper may have individual authorship assigned. Where available, figure panel/source data authorship is listed in the following database record: biostudies:S-SCDT-10_1038-S44318-024-00281-4.

## Disclosure and competing interests statement

The authors declare no competing interests.

# Expanded View Figures

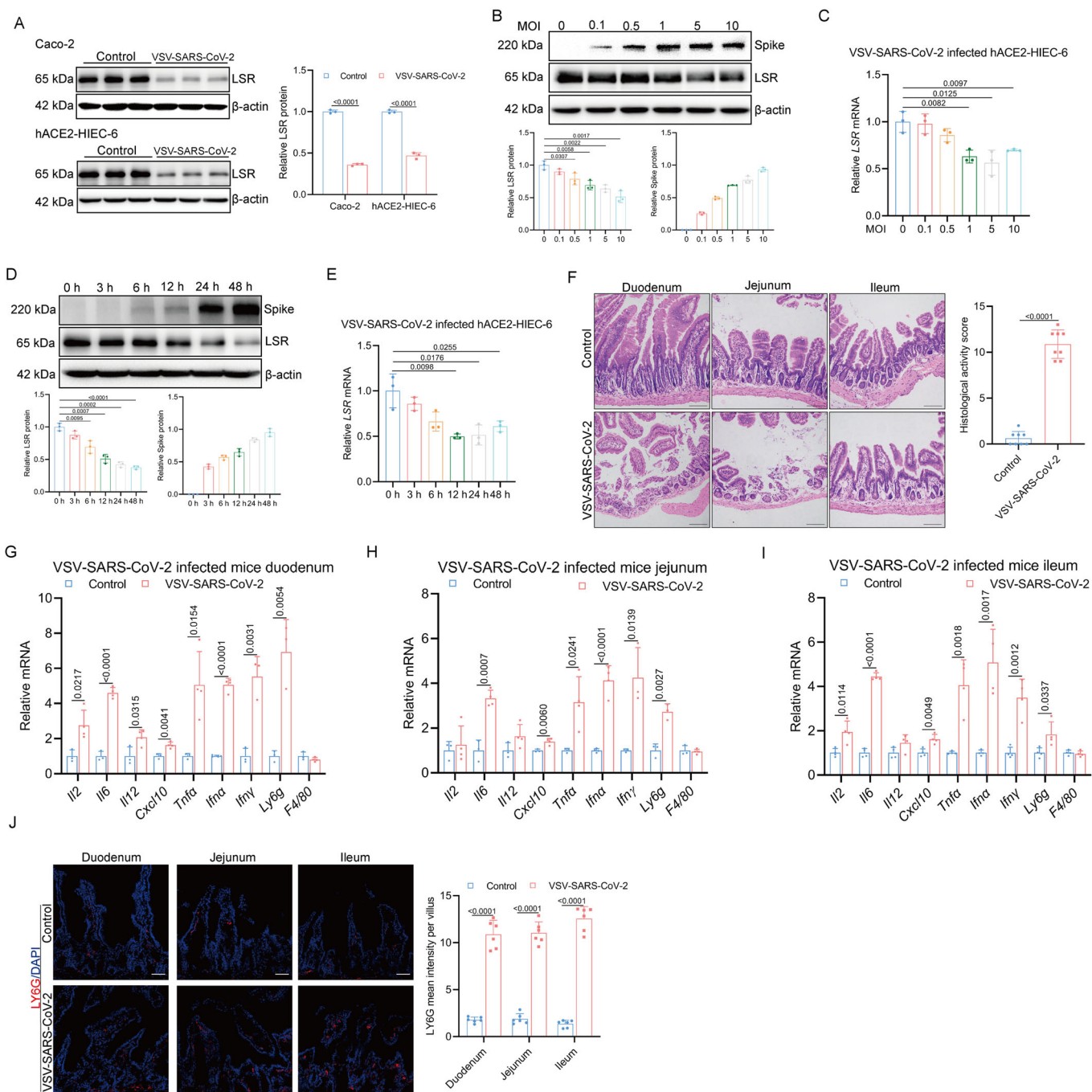

**Figure EV1. SARS-CoV-2 infection alters the expression of LSR in intestine.**

(A) Western blot and densitometric analysis of LSR in Caco-2 and hACE2-HIEC-6 cells with and without VSV-SARS-CoV-2 infection at 1 MOI for 24 h ($n = 3$). (B, C) Western blot and densitometric analysis of Spike and LSR (B), and expression of *LSR* mRNA analyzed by RT-qPCR (C) in hACE2-HIEC-6 cells infected with VSV-SARS-CoV-2 at the indicated MOI for 24 h ($n = 3$). (D, E) Western blot and densitometric analysis of Spike and LSR (D), and expression of *LSR* mRNA analyzed by RT-qPCR (E) in hACE2-HIEC-6 cells infected with VSV-SARS-CoV-2 at 1 MOI for the indicated time points ($n = 3$). (F) H&E staining and histological activity score in small intestines from hACE2-WT mice with and without VSV-SARS-CoV-2 infection ($n = 6$). (G–I) mRNA expression of *Il2, Il6, Il12, Cxcl10, Tnfa, Ifna, Ifnγ, Ly6g,* and *F4/80* analyzed by RT-qPCR in duodenal (G), jejunal (H), and ileal (I) segments from humanized ACE2 mice with and without VSV-SARS-CoV-2 infection ($n = 4$). (J) Immunofluorescence staining images and quantitative analysis of LY6G in small intestines from humanized ACE2 mice with and without VSV-SARS-CoV-2 infection ($n = 6$). "$n$" represents number of biological replicates. Scale bars: (F) and (J), 50 μm. Data represent mean ± SEM. Unpaired t test was performed. $p < 0.05$, the exact *p*-value is displayed; $p > 0.05$, the *p*-value is not displayed. Source data are available online for this figure.

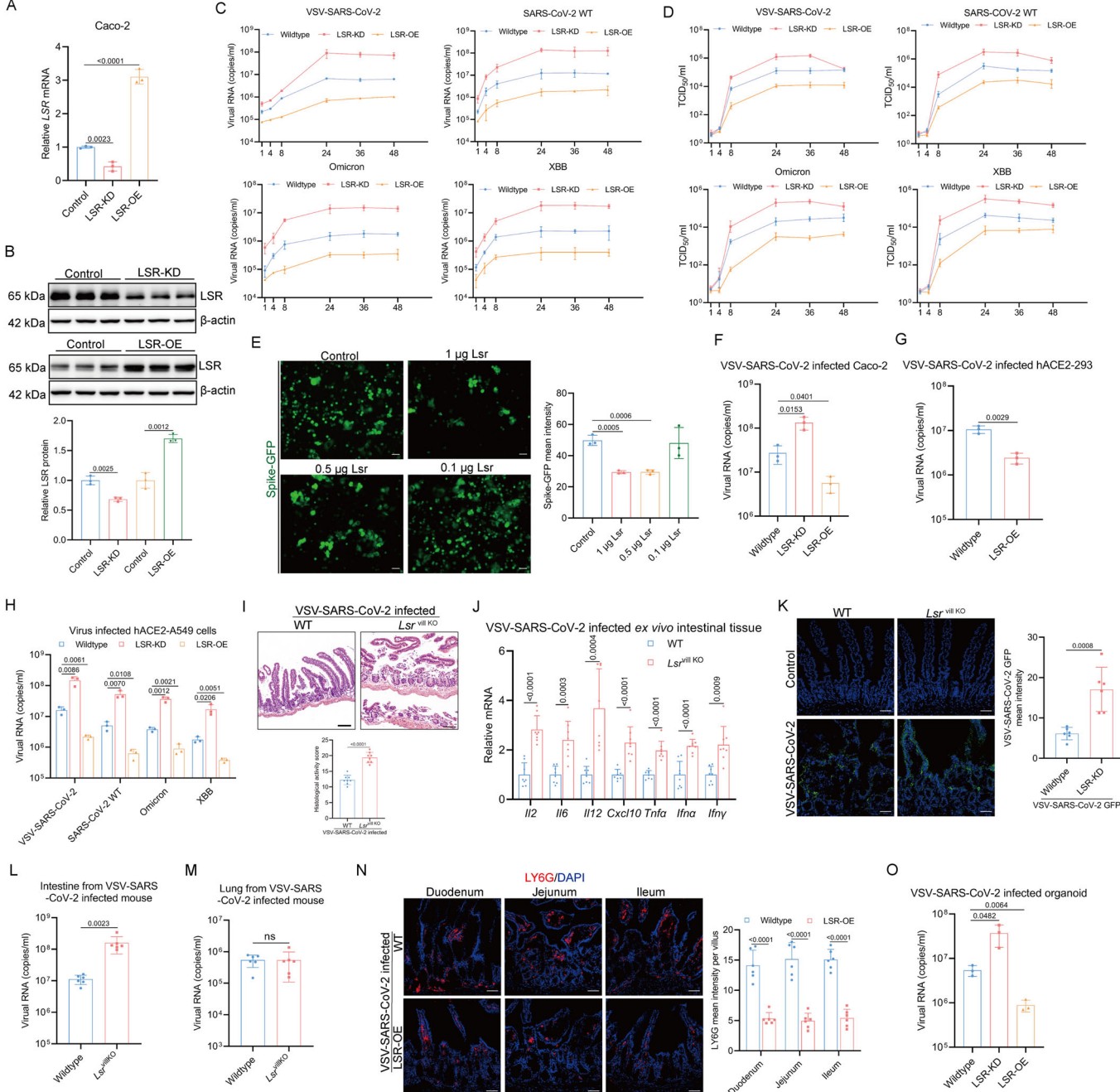

**Figure EV2.  LSR acts as a host defense factor against SARS-CoV-2 in intestine.**

(A, B) Expression of *LSR* mRNA analyzed by RT-qPCR (A), and western blot and densitometric analysis of LSR (B) in Caco-2 cells transduced with retrovirus containing LSR shRNA, human LSR transcript variant 2, or control vector ($n = 3$). (C) Viral RNA copies analyzed by RT-qPCR using primers targeting SARS-CoV-2 N gene in Caco-2 cells infected with VSV-SARS-CoV-2 and SARS-CoV-2 WT, Omicron, and XBB at 1 MOI for 1, 4, 8, 24, 36, and 48 h ($n = 3$). (D) Viral production assessed by $TCID_{50}$ assay on naïve Vero cells treated with supernatants from infected Caco-2 cells collected during the time course in (C) ($n = 5$). (E) Fluorescence images and quantitative analysis of Caco-2 cells transfected with control vector, or indicated amounts of plasmids containing mouse Lsr, and infected with VSV-SARS-CoV-2 at 1 MOI for 24 h ($n = 3$). (F) Viral RNA copies analyzed by RT-qPCR using primers targeting Spike gene in wild type, LSR-KD, and LSR-OE Caco-2 cells infected with VSV-SARS-CoV-2 at 1 MOI for 24 h ($n = 3$). (G) Viral RNA copies analyzed by RT-qPCR using primers targeting VSV-P in hACE2-293 cells infected with VSV-SARS-CoV-2 at 1 MOI for 24 h ($n = 3$). (H) Viral RNA copies analyzed by RT-qPCR using primers targeting VSV-P or SARS-CoV-2 N gene in wild type, LSR-KD, and LSR-OE hACE2-A549 cells infected with VSV-SARS-CoV-2 and SARS-CoV-2 WT, Omicron, and XBB at 1 MOI for 24 h ($n = 3$). (I, J) H&E staining and histological activity score (I), and expression of *Il2, Il6, Il12, Cxcl10, Tnfa, Ifna,* and *Ifny* mRNA analyzed by RT-qPCR (J) in VSV-SARS-CoV-2 infected ex vivo intestinal tissues from hACE2-WT and hACE2-*Lsr*[vill KO] mice ($n = 8$). (K) Fluorescence images and quantitative analysis of GFP in VSV-SARS-CoV-2 infected ex vivo intestinal tissues harvested from hACE2-WT and hACE2-*Lsr*[vill KO] mice ($n = 6$). (L, M) Viral RNA copies analyzed by RT-qPCR using primers targeting VSV-P in intestines (L) and lungs (M) from VSV-SARS-CoV-2 infected hACE2-WT and hACE2-*Lsr*[vill KO] mice ($n = 6$). (N) Immunofluorescence staining images and quantitative analysis of LY6G in small intestines from K18-hACE2-WT (WT) and K18-hACE2-LSR-OE (LSR-OE) mice intraperitoneally infected with 100 µl VSV-SARS-CoV-2 ($2 \times 10^8$ p.f.u./ml) ($n = 6$). (O) Viral RNA copies analyzed by RT-qPCR using primers targeting VSV-P in wild type, LSR-KD, and LSR-OE intestinal organoid infected with VSV-SARS-CoV-2 ($n = 3$). "*n*" represents number of biological replicates. Scale bars: (E), 100 µm; (I), (K), and (N), 50 µm. Data represent mean ± SEM. Unpaired t test was performed. $p < 0.05$, the exact *p*-value is displayed; $p > 0.05$, the *p*-value is not displayed. Source data are available online for this figure.

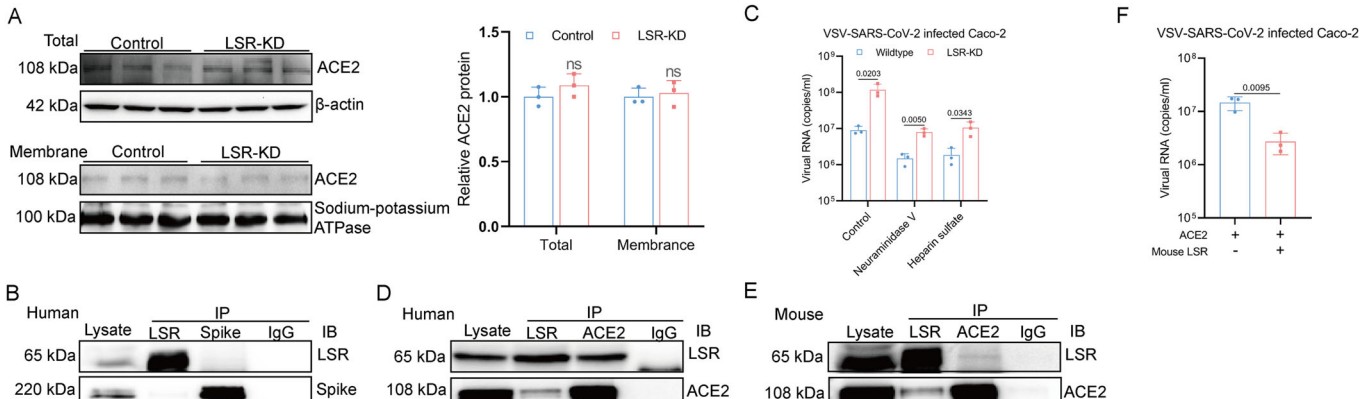

**Figure EV3.   LSR alters interactions between Spike and ACE2.**

(A) Western blot and densitometric analysis of total ACE2 and membrane ACE2 in wild type and LSR-KD Caco-2 cells ($n = 3$). (B) Co-IP showing that human LSR cannot interact with Spike. (C) Viral RNA copies analyzed by RT-qPCR using primers targeting VSV-P in Caco-2 cells treated with DMSO, 200 mU/ml neuraminidase V, or 300 μg/ml heparin sulfate at 37 °C for 1 h, then infected with VSV-SARS-CoV-2 at 1 MOI at 4 °C for 1 h ($n = 3$). (D) Co-IP showing that human LSR interacts with human ACE2 in HEK293 cells. (E) Co-IP showing that mouse LSR interacts with human ACE2. (F) Viral RNA copies analyzed by RT-qPCR using primers targeting VSV-P in Caco-2 cells transfected with human ACE2 and mouse LSR expressing plasmids simultaneously or human ACE2 expressing plasmid alone, and infected with VSV-SARS-CoV-2 at 1 MOI for 24 h ($n = 3$). All the Co-IP assays were performed in doubly transfected HEK293 cells. IB, immunoblot; IP, immunoprecipitation. "$n$" represents number of biological replicates. Data represent mean ± SEM. Unpaired t test was performed. $p < 0.05$, the exact $p$-value is displayed; $p > 0.05$, the $p$-value is not displayed. Source data are available online for this figure.

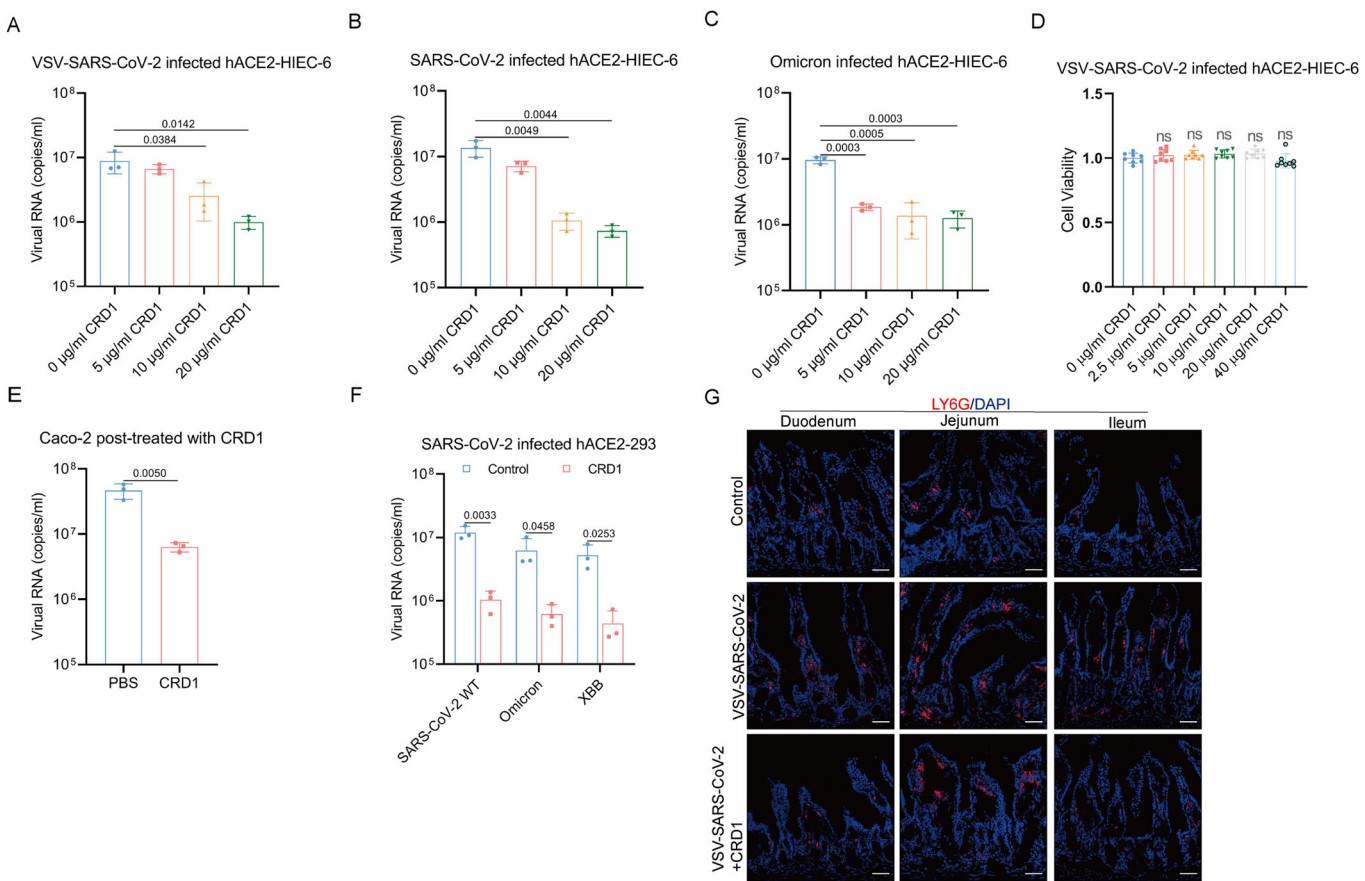

**Figure EV4.  LSR-derived CRD1 peptide inhibits infection of SARS-CoV-2.**

(**A**) Viral RNA copies analyzed by RT-qPCR using primers targeting VSV-P in hACE2-HIEC-6 cells treated with indicated concentrations of CRD1 and infected with VSV-SARS-CoV-2 ($n = 3$). (**B, C**) Viral RNA copies analyzed by RT-qPCR using primers targeting SARS-CoV-2 N gene in hACE2-HIEC-6 cells treated with indicated concentrations of CRD1 and infected with SARS-CoV-2 WT (**B**) and Omicron (**C**) ($n = 3$). (**D**) Cell viability determined by CCK8 assay in hACE2-HIEC-6 cells treated with indicated concentrations of CRD1 ($n = 8$). (**E**) Viral RNA copies analyzed by RT-qPCR using primers targeting VSV-P in Caco-2 cells infected with VSV-SARS-CoV-2 at 1 MOI at 4 °C for 1 h, and treated with PBS or 10 µg/ml CRD1 for 24 h ($n = 3$). (**F**) Viral RNA copies analyzed by RT-qPCR using primers targeting SARS-CoV-2 N gene in hACE2-293 cells treated with PBS or 10 µg/ml CRD1 before being infected with SARS-CoV-2 WT, Omicron, and XBB at 1 MOI for 24 h ($n = 3$). (**G**) Immunofluorescence images of LY6G in small intestines from control and VSV-SARS-CoV-2 infected K18-hACE2 mice with and without CRD1 treatment before infection. "$n$" represents number of biological replicates. Scale bars: (**G**), 50 µm. Data represent mean ± SEM. Unpaired t test was performed. $p < 0.05$, the exact $p$-value is displayed; $p > 0.05$, the $p$-value is not displayed. Source data are available online for this figure.

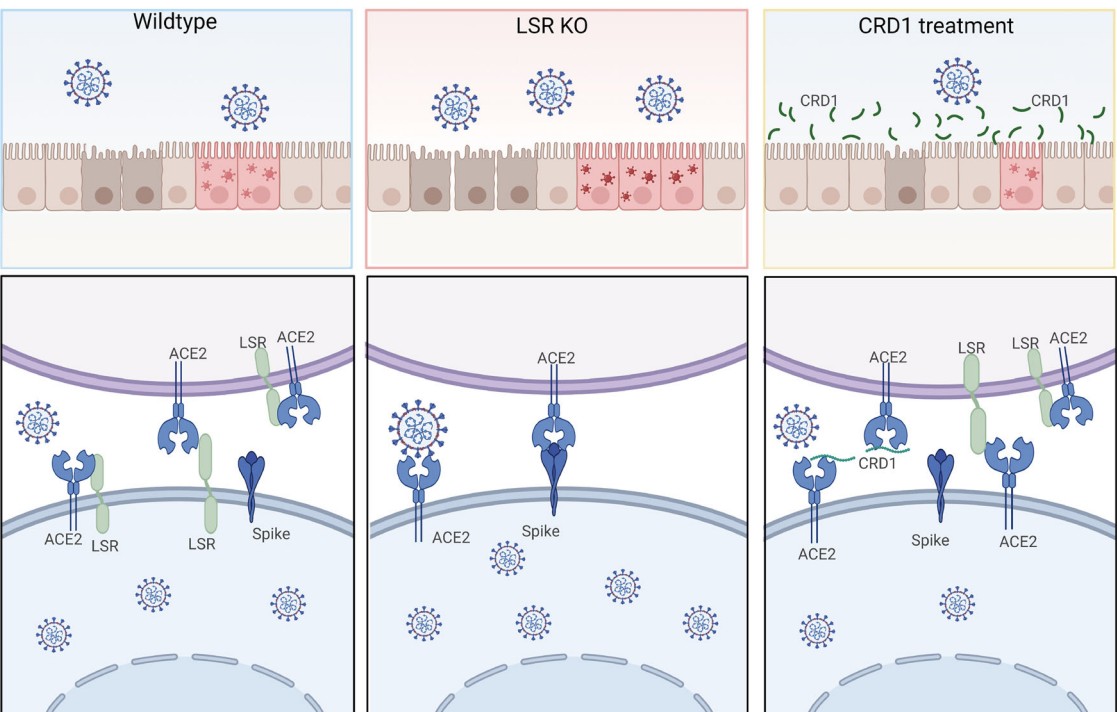

**Figure EV5. Summary.**

LSR interacts with ACE2 both in *cis* and in *trans* through its extracellular C-terminal region, preventing ACE2 binding to Spike protein, and thus blocking viral entry and inhibiting Spike-mediated cell–cell fusion. LSR knockout promotes SARS-CoV-2 infection in the small intestine, while LSR-derived peptide CRD1 blocks Spike binding with ACE2 to protect against SARS-CoV-2 infection. The graphic was created with BioRender.com.

