## [Peer Review File · The EMBO Journal]

Tight junction protein LSR is a host defense factor against SARS-CoV-2 infection in the small intestine

Yanan An, Chao Wang, Ziqi Wang, Feng Kong, Hao Liu, Min Jiang, Ti Liu, Shu Zhang, Kaige Du, Liang Yin, Peng Jiao, Ying Li, Baozhen Fan, Chengjun Zhou, Mingxia Wang, Hui Sun, Jie Lei, Shengtian Zhao, and Yongfeng Gong

Corresponding author(s): Yongfeng Gong (ygong@bzmc.edu.cn) , Shengtian Zhao (zhaoshengtian@sdu.edu.cn), Jie Lei (leijie@email.sdfmu.edu.cn)

Review Timeline:

Submission Date:	28th Feb 24
Editorial Decision:	3rd May 24
Revision Received:	31st Jul 24
Editorial Decision:	20th Sep 24
Revision Received:	29th Sep 24
Accepted:	10th Oct 24

Editor: Ieva Gailite

Transaction Report:

Dear Yongfeng,

Thank you for submitting your manuscript for consideration by the EMBO Journal. We have now received comments from three reviewers, which are included below for your information.

Based on the overall interest expressed in the referee reports and your willingness to engage in a major revision as expressed in the preliminary revision plan provided during the pre-decision consultation, I would like to invite you to address the comments of all reviewers in a revised version of the manuscript. I should add that it is The EMBO Journal policy to allow only a single major round of revision and that it is therefore important to resolve the main concerns at this stage. Please also note that the final decision will depend on the reviewers' assessment of the revised manuscript, which is difficult to predict at this stage.

We generally allow three months as standard revision time, which can be extended if necessary. As a matter of policy, competing manuscripts published during this period will not negatively impact on our assessment of the conceptual advance presented by your study. However, please contact me as soon as possible upon publication of any related work to discuss the appropriate course of action. I have currently extended the deadline to four months as requested. Should you foresee a problem in meeting this deadline, please let me know in advance to discuss an extension.

When preparing your letter of response to the referees' comments, please bear in mind that this will form part of the Review Process File and will therefore be available online to the community. For more details on our Transparent Editorial Process, please visit our website: <https://www.embopress.org/page/journal/14602075/authorguide#transparentprocess>. Please also see the attached instructions for further guidelines on preparation of the revised manuscript.

Please feel free to contact me if you have any further questions regarding the revision. Thank you for the opportunity to consider your work for publication. I look forward to receiving your revised manuscript.

With best regards,

Ieva

We realize that it is difficult to revise to a specific deadline. In the interest of protecting the conceptual advance provided by the work, we recommend a revision within 3 months (1st Aug 2024). Please discuss the revision progress ahead of this time with the editor if you require more time to complete the revisions.

Referee #1:

In this study the authors explore the role of lipolysis-stimulated lipoprotein receptor (LSR) as a host defense factor against SARS-CoV-2 in the small intestine. They used various experimental systems such as intestinal cell lines and intestinal organoids, ex vivo intestinal tissues, and humanized ACE2 mouse models, some with overexpression or KO of LSR. The authors show that LSR interacts with ACE2 both in cis and in trans and this interaction reduces the ability of Spike to bind to ACE2 and the virus to infect the cells. This is relevant in the context of infection and syncytia formation. The authors go on to identify the region of LSR that interacts with ACE2 and use this peptide to inhibit infection at the entry stage. This work is novel and carried out in a robust manner in a range of relevant models. It would be of interest to see the involvement of LSR in the lungs during SARS-CoV-2 infection. There were a few contradictions in the manuscript which are described below as well as several other points

Main comments

1. Most of the inhibitory effects triggered by LSR are modest (2 fold changes). The authors should tone down their interpretation that LSR is a restriction factor or that LSR knock-down promotes infection.
2. LSR down regulation in infected cells. How the authors distinguish between a specific effect and a consequence of the translational shutoff triggered by infection?
3. Line 138 - The authors suggest that binding of Spike to LSR leads to downregulation of LSR. This is quite assumptive as the authors don't show a direct interaction between the two proteins and later experiments confirm through co-IP that there isn't direct binding of these proteins (Fig. EV3B). Could the authors modify this sentence to remove any suggestion that LSR binds to Spike.
4. In figure 2 it is shown that SARS-CoV-2 infection causes increased tissue damage in LSRvill KO ex vivo and in vivo compared to the WT tissues. As LSR is a tight junction protein, may the disruption seen to the intestinal epithelium also be because of LSR KO as well as SARS-CoV-2 infection? (Shimada H., et al. 2017 - Scientific reports). The authors discuss the importance of LSR in maintaining endothelial and epithelial barrier integrity in the discussion (line 411). To clarify this the role of LSR in epithelial integrity, could the authors include additional controls showing histological activity in WT and LSRvill KO uninfected cells in Fig. 2H?
5. Figure 2 - The schematics are nice and helpful in describing the experimental methods but they are too small.
6. Figure 2. In Caco2 cells, the silencing of LSR increases vRNA by only 2-fold. The restriction is not that marked. The authors should show full viral replication curves.
7. Figure 2J. IN LSR KO mice, there is a 7-fold increase of viral RNA in the intestinal tissues. What happens in lung tissues? Levels should be similar in KO and WT mice. The authors could show the actual viral RNA levels in intestines and lungs, rather than a fold change ratio.
8. Figure 3F - The authors conclude that protease inhibitors reduced the quantity of viral RNA similarly in wildtype, LSR-KD, and LSR-OE cells. To better show this, statistics should be made across treatments, for example comparing camostat treated wildtype, LSR-KD, LSR-OE conditions and the same for E-64d and the control groups. Additionally, "mock" should not be used here as mock implies these cells weren't infected.

Minor comments

9. In the "Omicron" conditions, it isn't clarified which subvariant of Omicron is used in the study. Could the authors replace Omicron with the specific subvariant, such as BA.1, or include this information in the methodology.

10. Figure 4 - EV3E-G could be moved into the main of this figure as it is important to show which LSR isoforms don't interact with ACE2 as well as those that do. Furthermore, the layout of this figure is quite hard to follow, could it be rearranged in a more logical order and maybe some subsections can be grouped together, for example, all the LSR isoform co-IP experiments can go under one subsection. Finally, Fig. 4Q-U could form their own figure on LSR structure, separate from the interaction data with Spike and ACE2, and this way Figure 4 will also have less subsections and be easier to read.

11. Figure 6 - "Mock" is used to label the control condition which is misleading as mock suggests that these cells weren't infected.

12. Line 77 - Place "LSR" in brackets as an acronym.

13. Line 238 - Correction to "influences viral replication".

14. Line 279 - Correction to "LSR has the potential to bind".

Referee #2:

The paper seeks to examine whether LSR blocks spike binding to ACE2 and inhibits SARS-CoV-2 infection. This is a highly significant topic with technical as well as conceptual innovations. However, there is a strong lack of rigor in many of the experimental design, making it difficult to draw definite conclusions from the observed findings.

Two major points are listed here:

1. SARS-CoV-2 and VSV-SARS-CoV-2 infection of mouse intestines is a key model system used in the paper. But this model cannot be found in past literature and there is no validation of this seemingly novel system. Without viral antigen staining, there is no evidence of productive infection and damage to the intestine inflicted by the virus. If this paper seeks to characterize a new model, all relevant characterization data and proper controls need to be included.
2. Very small differences were seen with viral titers/RNA with LSR knockdown or expression including but not limited to Fig. 2f and Fig. 5a. Most virologists would discount this difference as being within the error range of the QPCR (less than 1 cycle) and TCID50 assays and not physiologically relevant.

Referee #3:

The authors show that LSR blocks SARS-CoV-2 binding to ACE-2 resulting in decreased infection in various in vitro, ex vivo and in vivo experiments. Although the paper is very interesting and the experiment are very elaborate, I have some concerns.

Concerns

- 1) Figure 1: The authors show that LSR expression is reduced upon VSV-SARS-CoV2 and authentic SARS-CoV-2 strain infection. Here, the authors claim that the reduced expression is the result of binding of the spike protein to ACE2 resulting in controlled downregulation of LSR (137-145). However, how do the authors correct for virus induced cell death which is clearly visible in all microscopy slides? Could the reduction of LSR protein also be the result of cell death (e.g. disrupted villus epithelium and clearly bigger necrotic cores in the organoids), especially since all analysis are performed in bulk (western blot, mRNA expression). In addition, spike protein is known to induce apoptosis (fig Fig. 1Q-R).
- 2) Figure 1. How does Spike protein induce mRNA decrease of LSR? Please explain.
- 3) Figure 1: The authors use a protein synthesis inhibitor cycloheximide (CHX) to proof the mechanism of LSR degradation (147-148). Please clarify the use of CHX and explain the mechanism.
- 4) Figure 1: The authors use bafilomycin (Baf-A1) to inhibit lysosomes. However, Baf-A1 is known to inhibit autophagy in general. Baf-A1 also affects the transport of endocytosed material from early to late endocytic compartments, Both processes are involved in viral entry and replication. Does treatment with Baf-A1 influence virus infection and is the observed level of LSR expression the results of less viral replication? Please included viral titers.
- 5) Figure 2: 5 days for intestinal organoid differentiation seems fast as differentiation can take up 7 - 14 days. Note: in the methods section 606-612 it is mentioned that organoids were mechanically dissociated before VSV-SARS-Cov-2 or SARS-CoV-2 infection. There is no infection protocol for intact organoid infection. In addition, size and shape vary among intestinal organoids. Are the input virus inoculum and also the results of organoids bulk analysis controlled for cell number?
- 6) Figure 3: why is LSR-OE not included in infection experiments? Does reduced binding and internalization in the end result in less infection?
- 7) Figure 5: please explain in more detail (341-346). Higher affinity of LSR CRD1 to ACE2?

8) Figure 5R: Are the calculations of 85% reduction for Omicron variant correct?

minor

9) Figure 2: LSR knock down or over expression is induced by treatment of doxycycline (dox). Are WT organoids also treated with dox? In addition, dox is known to alter intracellular pathways and changes the cell metabolism to glycolytic. This would be a proper control.

10) If considered as therapeutic intervention, how would it affect the regular function in the uptake of triglycerides-rich lipoproteins as the binding site to host-expressed LSR would be blocked. Please discuss briefly.

Responses to editor and reviewers' comments:

Reviewer #1-----Page 2

Reviewer #2-----Page 15

Reviewer #3-----Page 22

Reference -----Page 32

Reviewers' comments:

Referee #1 (Report for Author):

In this study the authors explore the role of lipolysis-stimulated lipoprotein receptor (LSR) as a host defense factor against SARS-CoV-2 in the small intestine. They used various experimental systems such as intestinal cell lines and intestinal organoids, ex vivo intestinal tissues, and humanized ACE2 mouse models, some with overexpression or KO of LSR. The authors show that LSR interacts with ACE2 both in cis and in trans and this interaction reduces the ability of Spike to bind to ACE2 and the virus to infect the cells. This is relevant in the context of infection and syncytia formation. The authors go on to identify the region of LSR that interacts with ACE2 and use this peptide to inhibit infection at the entry stage. This work is novel and carried out in a robust manner in a range of relevant models. It would be of interest to see the involvement of LSR in the lungs during SARS-CoV-2 infection. There were a few contradictions in the manuscript which are described below as well as several other points

Reply to comments by Reviewer #1:

We are pleased that Reviewer #1 found our study interesting and novel and provided an overall positive evaluation. We sincerely appreciate your high assessment of our work. Based on your suggestions, we have revised the manuscript as follows:

Main comments

1. Most of the inhibitory effects triggered by LSR are modest (2 fold changes). The authors should tone down their interpretation that LSR is a restriction factor or that LSR knock-down promotes infection.

Response: We sincerely appreciate both reviewers for this valuable comment (see also Reviewer #2, Comment 2). We agree that the inhibitory effects triggered by LSR are modest, with changes typically around 2-fold. We have optimized the assay

conditions to achieve more pronounced effects and have revised our interpretation to reflect that LSR serves as an antiviral host factor rather than a restriction factor.

2. LSR down regulation in infected cells. How the authors distinguish between a specific effect and a consequence of the translational shutoff triggered by infection?

Response: We thank the Reviewer for bringing up this important point. To address this concern, we have implemented additional controls and experiments to assess the specificity of LSR downregulation in infected cells.

SARS-CoV-2 NSP1 and NSP14 have been proved to employ divergent mechanisms to suppress host protein expression¹⁻⁴. Moreover, vesicular stomatitis virus (VSV) matrix (M) protein also has the capacity to shut off host cell mRNA export^{5,6}. We agree with the Reviewer that the LSR down regulation in infected cells may be a consequence of translational shutoff triggered by infection. To investigate this, Caco-2 cells were transfected with DNA plasmid encoding M protein, NSP1, or NSP14. After 24 hours of transfection, the expression of LSR was assessed via immunoblotting. The results showed that LSR levels were significantly decreased in Caco-2 cells overexpressing M protein, NSP1, or NSP14 (Fig. 2D; Response Figure 1), indicating that translational shutoff triggered by infection contribute to the LSR downregulation in infected cells.

Response Figure 1. Western blot and densitometric analysis of LSR in Caco-2 cells transfected with plasmids encoding VSV M protein, NSP1, or NSP14 (n=3).

In order to distinguish between a specific effect on LSR down-regulation versus a general consequence of the translational shutoff triggered by viral infection, Caco-2 cells were transfected with plasmids encoding M protein, NSP1, or NSP14 along with increasing amounts of plasmid encoding Spike protein for 24 hours. We observed that the Spike protein elicited a dose-dependent exacerbation of LSR expression inhibition alongside the translation suppression mediated by the host shutoff factors M protein, NSP1, and NSP14 (Fig. 2E; Response Figure 2), indicating that LSR downregulation in infected cells is not solely due to translational shutoff but also a specific consequence of Spike protein activity.

Response Figure 2. Western blot and densitometric analysis of LSR in Caco-2 cells transfected with plasmids encoding VSV M protein, NSP1, or NSP14, together with indicated amounts of plasmids expressing Spike protein (n=3).

To better clarify this point, we added some explanations to the Discussion section in revised version of the manuscript (line 470-472).

3. Line 138 - The authors suggest that binding of Spike to LSR leads to downregulation of LSR. This is quite assumptive as the authors don't show a direct interaction between the two proteins and later experiments confirm through co-IP that there isn't direct binding of these proteins (Fig. EV3B). Could the authors modify this sentence to remove any suggestion that LSR binds to Spike.

Response: We appreciate the Reviewer for pointing this mistake out. In response to your suggestion, we have revised the relevant sentence to remove any implication of direct binding between LSR and Spike. The revised sentence now reads: " As both VSV-SARS-CoV-2 containing the Spike protein and authentic SARS-CoV-2 induce LSR downregulation, we next attempted to investigate whether Spike protein is involved in the SARS-CoV-2 exposure-mediated LSR down-regulation." (line 141-143). This modification accurately reflects our findings without implying a direct interaction between the Spike protein and LSR.

4. In figure 2 it is shown that SARS-CoV-2 infection causes increased tissue damage in LSR^{vill^{KO}} ex vivo and in vivo compared to the WT tissues. As LSR is a tight junction protein, may the disruption seen to the intestinal epithelium also be because of LSR KO as well as SARS-CoV-2 infection? (Shimada H., et al. 2017 - Scientific reports). The authors discuss the importance of LSR in maintaining endothelial and epithelial barrier integrity in the discussion (line 411). To clarify this the role of LSR in epithelial integrity, could the authors include additional controls showing histological activity in WT and LSR^{vill^{KO}} uninfected cells in Fig. 2H?

Response: We thank the Reviewer for bringing this to our attention, which made us realized that we did not adequately explain the mouse model we used. Our previous work has demonstrated that the barrier function of tight junctions is not affected by deletion of LSR in intestine epithelial cells ⁷.

In response to the Reviewer's suggestion, we have included H&E staining and histological activity scores for the vehicle-treated *ex vivo* intestinal tissues from both hACE2-WT and hACE2-*Lsr^{villin^{KO}}* mice in Figure 3G (Response Figure 3). The histological analysis revealed that SARS-CoV-2 infection caused more pronounced pathological damage in *ex vivo* intestinal tissues from hACE2-*Lsr^{villin^{KO}}* mice compared to the hACE2-WT mice. Specifically, the SARS-CoV-2 infected small intestinal tissue from hACE2-*Lsr^{villin^{KO}}* mice exhibited more severe villus disruption, separation of the

lamina propria, and submucosal edema than the SARS-CoV-2 infected small intestinal tissue from hACE2-WT mice. Importantly, in the control groups, small intestinal tissue from hACE2-*Lsr^{villin} KO* mice did not show any significant pathological changes compared to hACE2-WT mice.

Response Figure 3. H&E staining and histological activity score of the control and SARS-CoV-2 WT infected *ex vivo* intestinal tissues harvested from hACE2-WT and hACE2-*Lsr^{villin} KO* mice (n=6). Scale bars: 50 μ m.

5. Figure 2 - The schematics are nice and helpful in describing the experimental methods but they are too small.

Response: We thank the Reviewer for this suggestion. We have resized the schematics to enhance their visibility and readability.

6. Figure 2. In Caco2 cells, the silencing of LSR increases vRNA by only 2-fold. The restriction is not that marked. The authors should show full viral replication curves.

Response: We really appreciate both Reviewers (See also Reviewer #2, Comment 2) for this valuable comment. We also agree with the Reviewer that the effects triggered by LSR knockdown or overexpression are modest. However, it is important to note that our research across various experimental models—including intestinal organoids, *ex vivo* intestinal tissue, and *in vivo* animal models—consistently demonstrates significant variations in viral loads associated with LSR knockdown or overexpression. These variations are accompanied by distinct pathological damage and altered profiles of

inflammatory cytokines. These findings suggest that the small differences observed in cell infection experiments may be influenced by experimental approach.

In the previous version of manuscript, we employed the domestically developed Coronavirus (2019-nCoV) Dual Probes qRT-PCR Kit (Beyotime, China), chosen for its lower cost and ease of use, to measure SARS-CoV-2 RNA levels. The commercial assay kits are usually based on various primer–probe sets, each characterized by different analytical specificities and sensitivities, potentially resulting in divergent outcomes. Studies evaluating various kits and comparing the efficiency of different RT-qPCR primer sets for COVID-19 detection revealed large differences in sensitivity^{8,9}. Here, we have investigated the performance of 4 primer-probe sets targeting various SARS-CoV-2 RNA regions, including Spike (S), nucleocapsid (N), envelope (E), and RNA-dependent RNA polymerase (RdRp). Under the same PCR condition, N-gene targeting primer-probe was demonstrated to be the most sensitive primer-probe set in analyzing our samples (Response Figure 4A-E). Given the variable performance of different primer-probe sets, we additionally quantified the SARS-CoV-2 RNA in our samples using the N-gene targeting primer-probe.

Response Figure 4. (A-E) Viral load analyzed by RT-qPCR in wildtype, LSR-KD, and LSR-OE Caco-2 cells infected with SARS-CoV-2 WT, Omicron, and XBB at 1 MOI for 24 hours, using the Novel Coronavirus (2019-nCoV) Dual Probes RT-qPCR Kit (A), or using the primers targeting Spike (B), Nucleocapsid (C), Envelope (D), and RNA-dependent RNA polymerase (RdRp) (E) (n=3).

Furthermore, it has been demonstrated that Caco-2 cells support high levels of viral production but not viral spread¹⁰. As a result, only an average of 10-15% of cells in the monolayer are infected with SARS-CoV-2 at different MOIs, rather than being uniformly infected¹⁰⁻¹³. While this level of infection enables the detection of potential inhibitory or enhancing effects of a perturbant, the observed effects may not be as pronounced. Moreover, all our analysis were performed in bulk. Thus, we further performed immunofluorescence staining with antibody targeting the SARS-CoV-2 N protein to visualize viral antigen expression and quantified the percentage of N protein positive Caco-2 cells and mean fluorescence intensity in the infected Caco-2 cells. Here, measurement of total viral RNA levels using RT-qPCR with the new primer sets revealed that infection by SARS-CoV-2 increased significantly by 10.4-fold (WT), 7.5-fold (Omicron), and 8.2-fold (XBB) in LSR-KO cells (Fig. 3D; Response Figure 5A). These results were further validated by immunofluorescence of the N protein, with nuclei counterstained using DAPI. The percentage of infected Caco-2 cells increased by 2.7-fold (WT), 2.5-fold (Omicron), and 2.8-fold (XBB), and the viral N protein signal increased by 2.9-fold (WT), 3.1-fold (Omicron), and 3.1-fold (XBB) in cells with disrupted LSR alleles (Fig. 3E; Response Figure 5B). Additionally, disruption of LSR led to a 10.5-fold increase in viral load in hACE2-A549 lung epithelial cells (Fig. EV2H; Response Figure 5C). In contrast, LSR overexpression reduced N protein levels across all isolates (Fig. 3D, E and EV2H; Response Figure 5). These findings indicate that LSR exhibits antiviral activity against SARS-CoV-2 and functions as a crucial host defense factor against the virus.

Response Figure 5. (A) Viral RNA copies analyzed by RT-qPCR using primers targeting SARS-CoV-2 N gene in wildtype, LSR-KD, and LSR-OE Caco-2 cells infected with SARS-CoV-2 WT, Omicron, and XBB at 1 MOI for 24 hours (n=3). (B) Immunofluorescence and quantitative analysis of SARS-CoV-2 N protein in Caco-2 cells infected with SARS-CoV-2 WT, Omicron, and XBB at 1 MOI for 24 hours (n=6). (C) Viral RNA copies analyzed by RT-qPCR using primers targeting VSV P gene or SARS-CoV-2 N gene in wildtype, LSR-KD, and LSR-OE hACE2-A549 cells infected with VSV-SARS-CoV-2 and SARS-CoV-2 WT, Omicron, and XBB at 1 MOI for 24 hours (n=3). Scale bars: 20 μ m.

Following the Reviewer's recommendation, we have performed additional experiments to generate full viral replication curves to quantify infection more accurately. Caco-2 cells were infected with SARS-CoV-2 at an MOI of 1 and cultured at 37°C for 1 hour. Afterward, the inoculum was removed and replaced with fresh medium (DMEM/F12 + 5% FBS), and the cells were incubated at 37°C. Cell lysates were collected at 1, 4, 8, 24, 36, and 48 hours post-infection to quantify viral RNA copy numbers by RT-qPCR. Supernatants from infected cells were collected and assessed for the production of de novo infectious viral particles using a 50% tissue culture infective dose (TCID₅₀) assay on naive Vero cells. Already at 1 hour post-infection, viral genome copies diverged concordantly with changes in LSR levels and LSR silencing supported an order of magnitude increase in viral amount, indicating that LSR is essential for inhibiting virus entry (Fig. EV2C; Response Figure 6A). Infectious progeny viruses were found to be released from infected cells as early as 8 hours after infection and plateaued around 24 hours after infection (Fig. EV2D; Response Figure 6B), in agreement with the previous report that SARS-CoV-2 completes one round of infection, from virus binding

and entry to replication and release of de novo infectious particles, within 8 hours in Caco-2 cells¹³. The virus production changes across different LSR level groups, quantified by TCID₅₀, were consistent with the observations from the RT-qPCR results (Fig. EV2D; Response Figure 6B).

Response Figure 6. (A) Viral RNA copies analyzed by RT-qPCR using primers targeting VSV P gene or SARS-CoV-2 N gene in Caco-2 cells infected with VSV-SARS-CoV-2 and SARS-CoV-2 WT, Omicron, and XBB at 1 MOI for 1-, 4-, 8-, 24-, 36- and 48- hours (n=3). (B) Viral production assessed by TCID₅₀ assay on naive Vero cells treated with supernatants from infected Caco-2 cells collected during the time course in A (n=5).

We hope these clarifications and additional data address the Reviewer's concerns and provide a more comprehensive understanding of the role of LSR in SARS-CoV-2 infection.

7. Figure 2J. IN LSR KO mice, there is a 7-fold increase of viral RNA in the intestinal tissues. What happens in lung tissues? Levels should be similar in KO and WT mice. The authors could show the actual viral RNA levels in intestines and lungs, rather than a fold change ratio.

Response: We appreciate the Reviewer's suggestion to provide actual viral RNA levels rather than just fold change ratios. Following the valuable suggestions of the Reviewer, we have performed additional experiments to measure and report the absolute viral RNA levels in both intestinal and lung tissues from LSR KO and WT mice. Our updated data, now included in the revised manuscript (see Fig. EV2L, M), show

that in hACE2-*Lsr^{vill} KO* mice, there was indeed a 10.9-fold increase in viral RNA levels in the intestinal tissues compared to hACE2-WT mice (Fig. EV2L; Response Figure 7A). Additionally, we have measured viral RNA levels in lung tissues, which revealed no statistically significant difference in hACE2-*Lsr^{vill} KO* mice compared to hACE2-WT mice (Fig. EV2M; Response Figure 7B).

Response Figure 7. Viral RNA copies analyzed by RT-qPCR using primers targeting VSV P gene in the intestines (A) or lungs (B) from VSV-SARS-CoV-2 infected hACE2-WT and hACE2-*Lsr^{vill} KO* mice (n=6).

We agree with the Reviewer that exploring the role of LSR in the lungs during SARS-CoV-2 infection is of interest. Although LSR is highly expressed in lung tissue ¹⁴, current research predominantly focuses on its role in tumor tissue ^{15, 16}, leaving the physiological function of LSR in normal lung tissue largely unexplored. Thus, we examined the effect of the LSR knockdown or overexpression on SARS-CoV-2 infection using the human lung cell line A549. Our results revealed that disruption of LSR led to significantly increase in viral load of VSV-SARS-CoV-2 and SARS-CoV-2 WT, Omicron, and XBB in hACE2-A549 cells (Fig. EV2H; Response Figure 8). In contrast, LSR overexpression led to a reduction in viral RNA production across all isolates (Fig. EV2H; Response Figure 8). Thus, LSR is an important host defence factor against SARS-CoV-2 infection.

Response Figure 8. Viral RNA copies analyzed by RT-qPCR using primers targeting VSV P gene or SARS-CoV-2 N gene in wildtype, LSR-KD, and LSR-OE hACE2-A549 cells infected with VSV-SARS-CoV-2 and SARS-CoV-2 WT, Omicron, and XBB at 1 MOI for 24 hours (n=3).

8. Figure 3F - The authors conclude that protease inhibitors reduced the quantity of viral RNA similarly in wildtype, LSR-KD, and LSR-OE cells. To better show this, statistics should be made across treatments, for example comparing camostat treated wildtype, LSR-KD, LDR-OE conditions and the same for E-64d and the control groups. Additionally, "mock" should not be used here as mock implies these cells weren't infected.

Response: We are grateful to the Reviewer for raising this issue. Following the valuable suggestions provided, we have performed statistical analyses across treatments, including comparisons between camostat-treated wildtype, LSR-KD, and LSR-OE conditions, as well as similar comparisons for E-64d and control groups. In hACE2-293 cells, which lack any detectable TMPRSS2 expression, inhibiting the endosomal fusion pathway with the cysteine protease inhibitor E-64d significantly attenuated infection, whereas the TMPRSS2 inhibitor camostat had no effect (Fig. 4E; Response Figure 9A). In Caco-2 cells, which permit viral entry through both the direct membrane fusion pathway and the endocytosis pathway, inhibition of TMPRSS2 using camostat and inhibition of cathepsin-L using E-64d blocked viral infection (Fig. 4F; Response Figure 9B). However, treatment with camostat or E-64d did not affect the impact of LSR on SARS-CoV-2 entry (Fig. 4E, F; Response Figure 9A, B), suggesting

that the effect of LSR on endocytosis or membrane fusion may be negligible in both hACE2-293 and Caco-2 cell lines.

Response Figure 9. Viral RNA copies analyzed by RT-qPCR using primers targeting SARS-CoV-2 N gene in wildtype, LSR-KD, and LSR-OE hACE2-293 cells (A) and Caco-2 cells (B), pretreated with E-64d (20 μ M) or Camostat (20 μ M), followed by infection with SARS-CoV-2 WT at 1 MOI for 24 hours (n=3).

We thank the Reviewer for pointing out that "mock" should not be used here. We apologize for this mistake and have corrected it in the revised version of the manuscript. Specifically, we used "control" to indicate that cells in this group were treated with vehicle or an equivalent control substance but were not infected with the virus.

Minor comments

9. In the "Omicron" conditions, it isn't clarified which subvariant of Omicron is used in the study. Could the authors replace Omicron with the specific subvariant, such as BA.1, or include this information in the methodology.

Response: We thank the Reviewer for raising this important point. We have clarified in the method section that the omicron variant BA.5.2 was used in our study (Line 634).

10. Figure 4 - EV3E-G could be moved into the main of this figure as it is important to show which LSR isoforms don't interact with ACE2 as well as those that do. Furthermore, the layout of this figure is quite hard to follow, could it be rearranged in a more logical order and maybe some subsections can be grouped together, for example, all the LSR isoform co-IP experiments can go under one subsection. Finally, Fig. 4Q-U could form their own figure on LSR structure, separate from the interaction data with

Spike and ACE2, and this way Figure 4 will also have less subsections and be easier to read.

Response: We appreciate the Reviewer's valuable recommendation on Figure 4. Following the suggestion of the Reviewer, we moved EV3E-G into the main of this figure to improve clarity and facilitate reader understanding (Fig. 5K-M). Additionally, we reorganized this figure to enhance readability, grouping similar experiments together under subsections. We also separated Fig. 4Q-U into their own figure to focus solely on LSR structure (Fig. 6A-F), thereby streamlining Figure 4 (Fig. 5 of the revised manuscript) and reducing subsections. Thank you again for your valuable recommendation.

11. Figure 6 - "Mock" is used to label the control condition which is misleading as mock suggests that these cells weren't infected.

Response: We thank the Reviewer for pointing out this mistake, we have used "control" to label the control condition accordingly in the revised manuscript.

12. Line 77 - Place "LSR" in brackets as an acronym.

Response: We appreciate the Reviewer's suggestion. In the revised manuscript, we have placed "LSR" in brackets as an acronym (Line 75).

13. Line 238 - Correction to "influences viral replication".

Response: We thank the Reviewer for bringing this error to our attention. We have corrected it accordingly in the revised manuscript (Line 285).

14. Line 279 - Correction to "LSR has the potential to bind".

Response: We thank the Reviewer for bringing this error to our attention. We have corrected it accordingly in the revised manuscript (Line 326).

Referee #2 (Report for Author)

The paper seeks to examine whether LSR blocks Spike binding to ACE2 and inhibits SARS-CoV-2 infection. This is a highly significant topic with technical as well as conceptual innovations. However, there is a strong lack of rigor in many of the experimental design, making it difficult to draw definite conclusions from the observed findings.

Reply to comments by Reviewer #2:

We would like to express our gratitude for your insightful and constructive comments and acknowledge the importance of enhancing the rigor of our experimental approach. In our revised manuscript, we have incorporated complementary approaches to strengthen our findings on the role of LSR in inhibiting SARS-CoV-2 infection.

Two major points are listed here:

1. SARS-CoV-2 and VSV-SARS-CoV-2 infection of mouse intestines is a key model system used in the paper. But this model cannot be found in past literature and there is no validation of this seemingly novel system. Without viral antigen staining, there is no evidence of productive infection and damage to the intestine inflicted by the virus. If this paper seeks to characterize a new model, all relevant characterization data and proper controls need to be included.

Response: We agree with the Reviewer the importance of validating the mouse intestine infection model used in our study.

Currently, SARS-CoV-2 infection in the intestine is studied based on monolayer cultures of intestinal epithelial cells, organoids, *ex vivo* intestinal tissue, and humanized ACE2 mice¹⁷. However, a model that can accurately reflect the response of the human intestine to the virus is still lacking. Thus, we utilized multiple intestinal infection models to recapitulate human-relevant intestinal pathophysiology induced by SARS-CoV-2 at cellular, tissue, and animal levels.

To simulate SARS-CoV-2 infection in the intestine in a more physiologically relevant setting, we used *ex vivo* intestinal tissues directly extracted from hACE2-expressing mice and immediately infected upon resection. The relevance of these platforms for studying SARS-CoV-2 infection has already been shown in multiple tissues including the lung^{18, 19}, the bronchi²⁰, the vessels²¹, the pancreas²², and the intestine^{18, 23}.

Although human ACE2 is expressed in the gastrointestinal tract of hACE2-expressing mice, no productive infection was observed upon intranasal inoculation^{24, 25}. The development of severe disease following SARS-CoV-2 infection is an important feature of the K18-hACE2 model, however, levels of viral RNA in gastrointestinal tract tissues (duodenum, ileum, and colon) have also been observed to be low following the intranasal route of infection²⁶. Zeng et al. demonstrated SARS-CoV-2 Spike RBD treatment of acid-challenged hACE2-B6J mice could induce intestinal inflammation, with the most severe pathological alterations in the duodenum, similar to the observations in COVID-19 patients²⁷. Therefore, we used this previously reported mouse model, treating the intestinal epithelium by acid enema for 2 min, then infecting mice with virus 16 hours later^{27, 28}. This model was adopted as confirmatory animal model to identify how LSR affect SARS-CoV-2 infection in intestine and similar phenotype was observed.

We agree with the Reviewer that without viral antigen staining, there is no evidence of productive infection and damage to the intestine inflicted by the virus. In response to this comment, we have performed immunofluorescence staining of intestine sections from the infected mice with antibody targeting nucleocapsid to demonstrate the productive infection by the SARS-CoV-2. For the VSV-SARS-CoV-2 infection model, which uses a recombinant virus expressing GFP, we detected GFP-positive signals in the small intestines of infected mice. Immunofluorescence staining for the SARS-CoV-2 nucleocapsid protein revealed intense staining in *ex vivo* intestinal tissue from hACE2-WT mice infected with SARS-CoV-2 WT (Fig. 3J; Response Figure 10A). Notably, increased staining was observed in intestinal tissue from hACE2-*Lsr^{villin} KO*

mice infected with SARS-CoV-2 WT (Fig. 3J; Response Figure 10A). Similar results were obtained with intestinal tissue infected with recombinant VSV-SARS-CoV-2 (Fig. EV2K; Response Figure 10B). In the K18-hACE2 mice infected with VSV-SARS-CoV-2, the mean GFP intensity was significantly reduced in the LSR-OE group compared to the WT group (Fig. 3L; Response Figure 10C).

Response Figure 10. (A) Immunofluorescence staining images and quantitative analysis of SARS-CoV-2 nucleocapsid protein in SARS-CoV-2 WT infected *ex vivo* intestinal tissues harvested from hACE2-WT and hACE2-*Lsr^{vill} KO* mice (n=6). (B) Fluorescence images and quantitative analysis of GFP in VSV-SARS-CoV-2 infected *ex vivo* intestinal tissues harvested from hACE2-WT and hACE2-*Lsr^{vill} KO* mice (n=6). (C) Fluorescence images and quantitative analysis of GFP in the small intestines from K18-hACE2-WT (WT) and K18-hACE2-LSR-OE (LSR-OE) mice with and without VSV-SARS-CoV-2 infection (n=6). Scale bars: A, B, and C, 50 μ m.

We would like to thank the Reviewer for the question. We have incorporated the points raised and the references mentioned above into the result (lines 227-228 and 238-240) and method sections (lines 688-690 and 875-876) regarding the methods for infecting mouse intestines. The findings described above have been included in the revised manuscript to provide the necessary characterization and validation of the mouse intestine infection model (Fig. 3J, L, and EV2K). We hope this additional information addresses the concern raised by the reviewer.

2. Very small differences were seen with viral titers/RNA with LSR knockdown or expression including but not limited to Fig. 2f and Fig. 5a. Most virologists would discount this difference as being within the error range of the QPCR (less than 1 cycle) and TCID50 assays and not physiologically relevant.

Response: We really appreciate both Reviewers (See also Reviewer #1, Major Comments: 1 and 6) for this valuable comment. We also agree with the Reviewer that the effects triggered by LSR knockdown or overexpression are modest. However, it is important to note that our research across various experimental models—including intestinal organoids, *ex vivo* intestinal tissue, and *in vivo* animal models—consistently demonstrates significant variations in viral loads associated with LSR knockdown or overexpression. These variations are accompanied by distinct pathological damage and altered profiles of inflammatory cytokines. These findings suggest that the small differences observed in cell infection experiments may be influenced by experimental approach.

In the previous version of manuscript, we employed the domestically developed Coronavirus (2019-nCoV) Dual Probes qRT-PCR Kit (Beyotime, China), chosen for its lower cost and ease of use, to measure SARS-CoV-2 RNA levels. The commercial assay kits are usually based on various primer–probe sets, each characterized by different analytical specificities and sensitivities, potentially resulting in divergent outcomes. Studies that have evaluated some kits and compared efficiency of different RT-qPCR primer sets for COVID-19 detection revealed large differences in sensitivity^{8, 9}. Here, we have investigated the performance of 4 primer-probe sets targeting various SARS-CoV-2 RNA regions, including Spike (S), nucleocapsid (N), envelope (E), and RNA-dependent RNA polymerase (RdRp). Under the same PCR condition, N-gene targeting primer-probe was demonstrated to be the most sensitive primer-probe set in analyzing our samples (Response Figure 11A-E). Given the variable performance of different primer-probe sets, we additionally quantified the SARS-CoV-2 RNA in our samples using the N-gene targeting primer-probe.

Response Figure 11. (A-E) Viral load analyzed by RT-qPCR in wildtype, LSR-KD, and LSR-OE Caco-2 cells infected with SARS-CoV-2 WT, Omicron, and XBB at 1 MOI for 24 hours, using the Novel Coronavirus (2019-nCoV) Dual Probes RT-qPCR Kit (A), or using the primers targeting Spike (B), Nucleocapsid (C), Envelope (D), and RNA-dependent RNA polymerase (RdRp) (E) (n=3).

Furthermore, it has been demonstrated that Caco-2 cells support high levels of viral production but not viral spread¹⁰. As a result, only an average of 10-15% of cells in the monolayer are infected with SARS-CoV-2 at different MOIs, rather than being uniformly infected¹⁰⁻¹³. While this level of infection enables the detection of potential inhibitory or enhancing effects of a perturbant, the observed effects may not be as pronounced. Moreover, all our analysis were performed in bulk. Thus, we further performed immunofluorescence staining with antibody targeting the SARS-CoV-2 nucleocapsid protein to visualize viral antigen expression and quantified the percentage of nucleocapsid protein positive Caco-2 cells and mean fluorescence intensity in the infected Caco-2 cells. Here, measurement of total viral RNA levels using RT-qPCR with the new primer sets revealed that infection by SARS-CoV-2 increased significantly by 10.4-fold (WT), 7.5-fold (Omicron), and 8.2-fold (XBB) in LSR-KO cells (Fig. 3D; Response Figure 12A). These results were further validated by immunofluorescence of the nucleocapsid protein, with nuclei counterstained using DAPI. The percentage of infected Caco-2 cells increased by 2.7-fold (WT), 2.5-fold (Omicron), and 2.8-fold (XBB), and the viral N protein signal increased by 2.9-fold (WT), 3.1-fold (Omicron), and 3.1-fold (XBB) in infected cells with disrupted LSR alleles (Fig. 3E; Response

Figure 12B). Additionally, disruption of LSR led to a 10.5-fold increase in viral load in hACE2-A549 lung epithelial cells (Fig. EV2H; Response Figure 12C). In contrast, LSR overexpression reduced N protein levels across all isolates (Fig. 3D, E and EV2H; Response Figure 12). These findings indicate that LSR exhibits antiviral activity against SARS-CoV-2 and functions as a crucial host defense factor against the virus.

Response Figure 12. (A) Viral RNA copies analyzed by RT-qPCR using primers targeting SARS-CoV-2 N gene in wildtype, LSR-KD, and LSR-OE Caco-2 cells infected with SARS-CoV-2 WT, Omicron, and XBB at 1 MOI for 24 hours (n=3). (B) Immunofluorescence and quantitative analysis of SARS-CoV-2 nucleocapsid protein in Caco-2 cells infected with SARS-CoV-2 WT, Omicron, and XBB at 1 MOI for 24 hours (n=6). (C) Viral RNA copies analyzed by RT-qPCR using primers targeting VSV P gene or SARS-CoV-2 N gene in wildtype, LSR-KD, and LSR-OE hACE2-A549 cells infected with VSV-SARS-CoV-2 and SARS-CoV-2 WT, Omicron, and XBB at 1 MOI for 24 hours (n=3). Scale bars: 20 μ m.

Following the reviewer's recommendation, we have performed additional experiments to generate full viral replication curves to quantify infection more accurately. Caco-2 cells were infected with SARS-CoV-2 at an MOI of 1 and cultured at 37°C for 1 hour. Afterward, the inoculum was removed and replaced with fresh medium (DMEM/F12 + 5% FBS), and the cells were incubated at 37°C. Cell lysates were collected at 1, 4, 8, 24, 36, and 48 hours post-infection to quantify viral RNA copy numbers by RT-qPCR. Supernatants from infected cells were collected and assessed for the production of de novo infectious viral particles using a 50% tissue culture infective dose (TCID₅₀) assay on naive Vero cells. Already at 1 hour post-infection, viral genome copies diverged concordantly with changes in LSR levels and LSR silencing supported an order of

magnitude increase in viral amount, indicating that LSR is essential for inhibiting virus entry (Fig. EV2C; Response Figure 13A). Infectious progeny viruses were found to be released from infected cells as early as 8 hours after infection and plateaued around 24 hours after infection (Fig. EV2D; Response Figure 13B), in agreement with the previous report that SARS-CoV-2 completes one round of infection, from virus binding and entry to replication and release of de novo infectious particles, within 8 hours in Caco-2 cells¹³. The virus production changes across different LSR level groups, quantified by TCID₅₀, were consistent with the observations from the RT-qPCR results (Fig. EV2D; Response Figure 13B).

Response Figure 13. (A) Viral RNA copies analyzed by RT-qPCR using primers targeting VSV P gene or SARS-CoV-2 N gene in Caco-2 cells infected with VSV-SARS-CoV-2 and SARS-CoV-2 WT, Omicron, and XBB at 1 MOI for 1-, 4-, 8-, 24-, 36- and 48- hours (n=3). (B) Viral production assessed by TCID₅₀ assay on naive Vero cells treated with supernatants from infected Caco-2 cells collected during the time course in A (n=5).

We hope these clarifications and additional data address the Reviewer's concerns and provide a more comprehensive understanding of the role of LSR in SARS-CoV-2 infection.

Referee #3 (Report for Author)

The authors show that LSR blocks SARS-CoV-2 binding to ACE-2 resulting in decreased infection in various in vitro, ex vivo and in vivo experiments. Although the paper is very interesting and the experiments are very elaborate, I have some concerns.

We thank the Reviewer for the valuable feedback. We are pleased that the Reviewer finds our study interesting and recognizes the effort put into our experiments. We appreciate the opportunity to address the Reviewer's concerns and have made the following revisions to enhance the clarity and rigor of our findings:

Concerns

1) Figure 1: The authors show that LSR expression is reduced upon VSV-SARS-CoV2 and authentic SARS-CoV-2 strain infection. Here, the authors claim that the reduced expression is the result of binding of the Spike protein to ACE2 resulting in controlled downregulation of LSR (137-145). However, how do the authors correct for virus-induced cell death which is clearly visible in all microscopy slides? Could the reduction of LSR protein also be the result of cell death (e.g. disrupted villus epithelium and clearly bigger necrotic cores in the organoids), especially since all analyses are performed in bulk (western blot, mRNA expression). In addition, Spike protein is known to induce apoptosis (see Fig. 1Q-R).

Response: We appreciate the Reviewer's comment. We acknowledge the importance of considering the potential contribution of virus-induced cell death to the observed reduction in LSR expression, since SARS-CoV-2 infection is known to induce apoptosis and other forms of cell death^{29,30}. As the reviewer mentioned, treating VSV-SARS-CoV-2-exposed intestinal organoids with the apoptosis inhibitor Z-VAD-FMK enhanced the expression of LSR (Response Figure 14), indicating that virus-induced cell death could contribute to the observed reduction in LSR expression.

Response Figure 14. Western blot and densitometric analysis of LSR in intestinal organoid treated with vehicle or 20 μ M Z-VAD-FMK and infected with or without VSV-SARS-CoV-2 (n=3).

Next, we incorporated propidium iodide (PI) staining in the infected intestine and organoid. PI is a vital dye that selectively enters and stains the nuclei of dead cells ^{31, 32}. By focusing on PI-negative cells, we can more accurately evaluate LSR expression at the cellular level, rather than relying solely on bulk analyses, thereby supporting our claim that the reduction of LSR is indeed linked to Spike protein rather than being solely a result of cell death. Immunofluorescent staining demonstrated reduction of LSR protein expression was also evident in PI-negative cells on microscopy slides, although the decrease was less pronounced than in the bulk samples (Fig. 1R, S; Response Figure 15A, B). These results indicate that the alteration of LSR expression is not solely a consequence of cell death.

Response Figure 15. (A and B) Immunofluorescence staining images and quantitative analysis of LSR (green) in the PI (red) negative areas of the small intestine from hACE2-transgenic mice infected with or without VSV-SARS-CoV-2 (A), and of the intestinal organoids infected with or without SARS-CoV-2 WT (B) (n=6). Scale bars: A, 50 μ m; B, 20 μ m.

2) Figure 1. How does Spike protein induce mRNA decrease of LSR? Please explain.

Response: We appreciate the Reviewer's insightful comment. To date, there is limited understanding of the regulatory mechanisms and signaling pathways implicated in the modulation of LSR gene expression. Activation of the interferon (IFN) signaling cascade represents a fundamental aspect of the innate immune response following SARS-CoV-2 infection³³. Spike protein itself can also activate IFN pathway^{34, 35}. Further, previously published work showed the inhibitory effects of some of the IFNs on expression level of long ACE2 mRNA³⁶. Thus, we assessed whether LSR is an IFN-response gene. Caco-2 monolayer cultures were stimulated with 100 IU ml⁻¹ or 1,000 IU ml⁻¹ of recombinant human IFN- α (IFN- α 2a), IFN- β , IFN- γ , or IFN- λ for 24 hours and LSR mRNA levels were quantified using RT-qPCR. Here, treatment of Caco-2 cells with IFN- α , IFN- γ , or IFN- λ caused downregulation of *LSR*, with IFN- α exhibiting the highest inhibitory effects on LSR expression (Fig. 2C; Response Figure 16). Consequently, the decrease in LSR mRNA induced by Spike may be attributed to the increase of IFN.

Response Figure 16. Expression of *LSR* mRNA measured by RT-qPCR in Caco-2 cells treated with or without 100 IU/ml or 1,000 IU/ml of recombinant human IFN- α (IFN- α 2a), IFN- β , IFN- γ , or IFN- λ for 24 hours (n=3).

IFNs have been reported to mediate suppression of gene expression through extensive remodeling of the epigenome, including the creation and disassembly of

enhancers and the modulation of histone marks that regulate chromatin accessibility and the functions of enhancers and promoters^{37, 38}. The mechanisms of IFN-mediated suppression of LSR gene expression, however, require further investigation.

To better clarify this point, we added some explanations to the Discussion section in revised version of the manuscript (line 474-482).

3) Figure 1: The authors use a protein synthesis inhibitor cycloheximide (CHX) to proof the mechanism of LSR degradation (147-148). Please clarify the use of CHX and explain the mechanism.

Response: We thank the Reviewer for bringing this to our attention. To enhance reader understanding, we have clarified the use of CHX and provided a brief explanation of its mechanism in the Results section of the revised manuscript (line 166-168), as follows:

"...100 µg/ml cycloheximide (CHX), a well-established protein synthesis inhibitor that functions by blocking the elongation phase of translation to prevent the synthesis of new proteins..."

4) Figure 1: The authors use bafilomycin (Baf-A1) to inhibit lysosomes. However, Baf-A1 is known to inhibit autophagy in general. Baf-A1 also affects the transport of endocytosed material from early to late endocytic compartments, Both processes are involved in viral entry and replication. Does treatment with Baf-A1 influence virus infection and is the observed level of LSR expression the results of less viral replication? Please included viral titers.

Response: We thank the Reviewer for pointing this out. The treatment with bafilomycin A1 (Baf-A1) was actually conducted on Caco-2 cells treated with Spike protein, it was mistakenly labeled as VSV-SARS-CoV-2-treated Caco-2 cells in Fig. EV1 M and N. We apologize for the labelling mistakes and have corrected it in revised version of the manuscript (Fig. 2F, G).

We agree with the Reviewer that Baf-A1 is known to inhibit lysosomal function and affect autophagy and endocytic trafficking. These processes are indeed involved in viral entry and replication, raising the question of how Baf-A1 treatment might influence viral infection and subsequently affect LSR expression. To address this, we treated Caco-2 cells with Baf-A1 and then infected them with VSV-SARS-CoV-2. Our results indicate that Baf-A1 treatment significantly reduced viral load and restored LSR expression (Fig. 2H; Response Figure 17), suggesting that the observed recovery in LSR expression might be partially due to decreased viral replication rather than solely due to the inhibition of lysosomal function.

Response Figure 17. (A) Viral RNA copies analyzed by RT-qPCR using primers targeting VSV P gene in Caco-2 cells pretreated with 200 nM lysosome inhibitor bafilomycin A1 (Baf-A1) for 2 hours, followed by infection with VSV-SARS-CoV-2 for 0, 3, 6, and 12 hours (n=3). (B) Western blot and densitometric analysis of LSR in Caco-2 cells pretreated with 200 nM Baf-A1 for 2 hours, followed by infection with VSV-SARS-CoV-2 for 0, 3, 6, and 12 hours (n=3).

To exclude the effect of Baf-A1 on endocytosis, Caco-2 cells were transfected with plasmids expressing Spike protein for 24 hours and then treated with Baf-A1 for 0, 3, 6, and 12 hours to detect the level of LSR. The results were similar to those obtained in Caco-2 cells treated with Spike protein or VSV-SARS-CoV-2, where Baf-A1 treatment inhibited the decrease in LSR (Fig. 2I; Response Figure 18).

Response Figure 18. Western blot and densitometric analysis of LSR in Caco-2 cells transfected with plasmids expressing Spike protein for 24 hours, followed by 200 nM Baf-A1 treatment for 0, 3, 6 and 12 hours (n=3).

5) Figure 2: 5 days for intestinal organoid differentiation seems fast as differentiation can take up 7 - 14 days. Note: in the methods section 606-612 it is mentioned that organoids were mechanically dissociated before VSV-SARS-Cov-2 or SARS-CoV-2 infection. There is no infection protocol for intact organoid infection. In addition, size and shape vary among intestinal organoids. Are the input virus inoculum and also the results of organoids bulk analysis controlled for cell number?

Response: We thank the Reviewer for pointing this out. We agree with the Reviewer that differentiation of intestinal organoid takes up 7–14 days, including 3 days for the hiPSCs to differentiate into definitive endoderm, 5 days for the endodermal cells to differentiate into mid-/hindgut, and 7 days for the embedded mid-/hindgut spheroids to mature into intestinal organoids. We apologize for the mistake regarding the timeline for intestinal organoid differentiation and have corrected it in the revised version of the manuscript (line 248).

We also apologize for the insufficient information regarding the intestine organoid virus infection. We employed an approach that has been demonstrated to facilitate productive virus infection in organoids³⁹⁻⁴¹. The organoids were mechanically dissociated using a narrowed Pasteur Pipette for 5–7 times up and down to expose the apical and basolateral epithelial surface to the virus^{42, 43}. Subsequently, the

organoid fragments were incubated in the full growth medium containing virus and Matrigel for 24 hours at 37 °C, with continuous mixing every 30 min during the initial 6-hour period. The infection protocol and the relevant studies supporting this approach have been added in the Methods section of the revised manuscript (line 672-679).

We agree with the Reviewer that both the input virus inoculum and the results of organoid bulk analysis should be controlled for cell number to account for any variations in organoid morphology. In our experiment, organoids were mechanically dissociated to obtain uniformly sized organoid fragments prior to virus infection. This procedure is expected to enhance experimental reproducibility by minimizing variability in organoid size and morphology. This detail was explicitly mentioned in the Methods section of revised version of the manuscript (line 674-676).

Given that the organoid results were obtained through bulk analysis, we performed immunofluorescence staining on virus-inoculated organoids using an antibody targeting the SARS-CoV-2 N protein. This allowed us to visualize viral antigen expression and quantify the percentage of N protein-positive cells and mean fluorescence intensity in the infected cells. Immunofluorescence staining for the N protein showed that LSR deficiency increased, while LSR overexpression reduced, both the percentage of infected cells and the viral burden in single cells of intestinal organoids exposed to SARS-CoV-2 (Fig. 3Q; Response Figure 19).

Response Figure 19. Immunofluorescence staining images and quantitative analysis of SARS-CoV-2 N protein in intestinal organoids infected with SARS-CoV-2 WT (A), Omicron (B), and XBB (C) (n=6) at 1 MOI for 24 hours. Scale bars: 20 μ m.

6) Figure 3: why is LSR-OE not included in infection experiments? Does reduced binding and internalization in the end result in less infection?

Response: We apologize for the omission. In our revised manuscript, we have now challenged wildtype, LSR-KD, and LSR-OE Caco-2 cells with replication defective HIV-1-based SARS-CoV-2 Spike protein pseudotyped virus and assayed for luciferase activity 24 hours post-infection. Our updated results show that LSR-OE cells exhibit reduced viral infection compared to control cells (Fig. 4D; Response Figure 20), suggesting that LSR overexpression interferes with this replication defective virus's ability to enter the cells.

Response Figure 20. The viral load assessed by luciferase analysis in wildtype, LSR-KD and LSR-OE Caco-2 cells infected with replication-defective pseudovirus bearing Spike protein from SARS-CoV-2 WT at 1 MOI for 24 hours (n=3).

7) Figure 5: please explain in more detail (341-346). Higher affinity of LSR CRD1 to ACE2?

Response: We thank the Reviewer for bringing up this important point and apologize for the lack of clarity. The time of CRD1 addition assay (depicted in Fig. 5E and F) was performed as follows: cells were treated with 10 μ g/ml of CRD1 1 hour before infection, during infection (at 0 hour), and 1 hour post-infection with either SARS-CoV-2 WT, Omicron, or XBB. 1 hour after virus inoculation, the medium was replaced with fresh

medium devoid of CRD1 or virus. Cells were then harvested 24 hours later for virus load assessment. Our observations indicate that the antiviral effect of CRD1 required early addition in both Caco-2 and hACE2-HIEC-6 cells, thereby confirming its influence on the entry phase of SARS-CoV-2 infection. We apologize for missing the experimental description of medium replacement. We have added this part in the Methods section of the revised manuscript (line 792-793).

The details of the experiment described in lines 341-346 of the previous version and lines 390-393 of the revised version of the manuscript are as follows: Caco-2 cells were infected with VSV-SARS-CoV-2 at 1 MOI at 4°C for 1 hour. Then the cells were washed twice to remove the unattached viruses, and treated with PBS or 10 µg/ml CRD1 for 24 hours. Following the addition of CRD1 and a 24-hour incubation period at 37°C, CRD1 continued to demonstrate an inhibitory effect on the virus infection. However, at this time, we cannot conclude that LSR CRD1 has a higher binding affinity to ACE2, since we cannot rule out its role in other processes involving virus infection such as the transmission of offspring viruses or the virus-induced syncytia formation, and future comprehensive studies will be necessary to quantitatively assess the binding kinetics and affinities of these interactions. We have rephrased our statements and included an explanation of these points in our revised manuscript (line 794-799 of the method section and line 544-549 of discussion section).

8) Figure 5R: Are the calculations of 85% reduction for Omicron variant correct?

Response: We thank the Reviewer for bringing this to our attention. Upon re-evaluation, we have identified an error in our calculations. The correct assessment indicates a reduction of 50%, rather than 85%. Following Reviewer #1's suggestion to provide actual viral RNA levels instead of just fold change ratios, we conducted additional experiments to measure and report the viral RNA copy numbers. This description has been revised accordingly in the updated version of the manuscript.

minor

9) Figure 2: LSR knock down or over expression is induced by treatment of doxycycline (dox). Are WT organoids also treated with dox? In addition, dox is known to alter intracellular pathways and changes the cell metabolism to glycolytic. This would be a proper control.

Response: We thank the Reviewer for highlighting the importance of including appropriate controls when utilizing doxycycline (dox) for inducing LSR knockdown or overexpression. In our study, WT organoids were indeed treated with dox to account for any potential effects of dox beyond LSR manipulation. This information was inadvertently omitted from the initial description. We have added this information into the Methods section to explicitly state that WT organoids underwent the same dox treatment as the experimental groups (line 668-669).

10) If considered as therapeutic intervention, how would it affect the regular function in the uptake of triglycerides-rich lipoproteins as the binding site to host-expressed LSR would be blocked. Please discuss briefly.

Response: We agree with the Reviewer that LSR plays a crucial role in the uptake of triglyceride-rich lipoproteins, and manipulating host-expressed LSR could potentially affect its physiological function in lipid uptake. As suggested by the Reviewer, we have included a brief discussion in our revised manuscript regarding the potential impact of LSR-targeted therapies on lipid uptake and metabolism (line 522-525), as follows:

“LSR plays a crucial role in the uptake of triglyceride-rich lipoproteins by binding to these lipoproteins and facilitating their internalization into cells. Thus, when considering LSR as a therapeutic target to inhibit SARS-CoV-2 infection, it is essential to consider the potential impact of targeting LSR on lipid uptake and metabolism.”

Reference:

1. Schubert, K. *et al.* SARS-CoV-2 Nsp1 binds the ribosomal mRNA channel to inhibit translation. *Nat Struct Mol Biol* **27**, 959-966 (2020).
2. Thoms, M. *et al.* Structural basis for translational shutdown and immune evasion by the Nsp1 protein of SARS-CoV-2. *Science* **369**, 1249-1255 (2020).
3. Yuan, S. *et al.* Nonstructural Protein 1 of SARS-CoV-2 Is a Potent Pathogenicity Factor Redirecting Host Protein Synthesis Machinery toward Viral RNA. *Molecular cell* **80**, 1055-1066 e1056 (2020).
4. Hsu, J.C., Laurent-Rolle, M., Pawlak, J.B., Wilen, C.B. & Cresswell, P. Translational shutdown and evasion of the innate immune response by SARS-CoV-2 NSP14 protein. *Proceedings of the National Academy of Sciences of the United States of America* **118** (2021).
5. Desforges, M. *et al.* Different host-cell shutoff strategies related to the matrix protein lead to persistence of vesicular stomatitis virus mutants on fibroblast cells. *Virus Res* **76**, 87-102 (2001).
6. Enninga, J., Levy, D.E., Blobel, G. & Fontoura, B.M. Role of nucleoporin induction in releasing an mRNA nuclear export block. *Science* **295**, 1523-1525 (2002).
7. An, Y. *et al.* LSR targets YAP to modulate intestinal Paneth cell differentiation. *Cell Rep* **42**, 113118 (2023).
8. Vogels, C.B.F. *et al.* Analytical sensitivity and efficiency comparisons of SARS-CoV-2 RT-qPCR primer-probe sets. *Nat Microbiol* **5**, 1299-1305 (2020).
9. Linkowska, K. *et al.* Commercially available SARS-CoV-2 RT-qPCR diagnostic tests need obligatory internal validation. *Sci Rep* **13**, 6991 (2023).
10. Thorne, L.G. *et al.* SARS-CoV-2 sensing by RIG-I and MDA5 links epithelial infection to macrophage inflammation. *The EMBO journal* **40**, e107826 (2021).
11. Shuai, H. *et al.* Differential immune activation profile of SARS-CoV-2 and SARS-CoV infection in human lung and intestinal cells: Implications for treatment with IFN-beta and IFN inducer. *J Infect* **81**, e1-e10 (2020).
12. Zupin, L. *et al.* Effect of Short Time of SARS-CoV-2 Infection in Caco-2 Cells. *Viruses* **14** (2022).
13. Koch, J. *et al.* TMPRSS2 expression dictates the entry route used by SARS-CoV-2 to infect host cells. *EMBO J* **40**, e107821 (2021).
14. Mesli, S. *et al.* Distribution of the lipolysis stimulated receptor in adult and embryonic murine tissues and lethality of LSR-/- embryos at 12.5 to 14.5 days of gestation. *European journal of biochemistry* **271**, 3103-3114 (2004).
15. Zhang, M. & Ma, C. LSR Promotes Cell Proliferation and Invasion in Lung Cancer. *Comput Math Methods Med* **2021**, 6651907 (2021).
16. Arai, W. *et al.* Downregulation of angulin-1/LSR induces malignancy via upregulation of EGF-dependent claudin-2 and TGF-beta-dependent cell metabolism in human lung adenocarcinoma A549 cells. *Oncotarget* **14**, 261-275 (2023).
17. Stanifer, M.L. *et al.* Critical Role of Type III Interferon in Controlling SARS-CoV-2 Infection in Human Intestinal Epithelial Cells. *Cell Rep* **32**, 107863 (2020).

18. Chu, H. *et al.* Host and viral determinants for efficient SARS-CoV-2 infection of the human lung. *Nature communications* **12**, 134 (2021).
19. Hui, K.P.Y. *et al.* Tropism, replication competence, and innate immune responses of the coronavirus SARS-CoV-2 in human respiratory tract and conjunctiva: an analysis in ex-vivo and in-vitro cultures. *Lancet Respir Med* **8**, 687-695 (2020).
20. Hui, K.P.Y. *et al.* SARS-CoV-2 Omicron variant replication in human bronchus and lung ex vivo. *Nature* **603**, 715-720 (2022).
21. Eberhardt, N. *et al.* SARS-CoV-2 infection triggers pro-atherogenic inflammatory responses in human coronary vessels. *Nat Cardiovasc Res* **2**, 899-916 (2023).
22. Tang, X. *et al.* SARS-CoV-2 infection induces beta cell transdifferentiation. *Cell Metab* **33**, 1577-1591 e1577 (2021).
23. Chu, H. *et al.* SARS-CoV-2 Induces a More Robust Innate Immune Response and Replicates Less Efficiently Than SARS-CoV in the Human Intestines: An Ex Vivo Study With Implications on Pathogenesis of COVID-19. *Cell Mol Gastroenterol Hepatol* **11**, 771-781 (2021).
24. Bao, L. *et al.* The pathogenicity of SARS-CoV-2 in hACE2 transgenic mice. *Nature* **583**, 830-833 (2020).
25. Sun, S.H. *et al.* A Mouse Model of SARS-CoV-2 Infection and Pathogenesis. *Cell Host Microbe* **28**, 124-133 e124 (2020).
26. Winkler, E.S. *et al.* SARS-CoV-2 infection of human ACE2-transgenic mice causes severe lung inflammation and impaired function. *Nat Immunol* **21**, 1327-1335 (2020).
27. Zeng, F.M. *et al.* SARS-CoV-2 spike spurs intestinal inflammation via VEGF production in enterocytes. *EMBO Mol Med* **14**, e14844 (2022).
28. Kuba, K. *et al.* A crucial role of angiotensin converting enzyme 2 (ACE2) in SARS coronavirus-induced lung injury. *Nat Med* **11**, 875-879 (2005).
29. Yuan, C. *et al.* The role of cell death in SARS-CoV-2 infection. *Signal Transduct Target Ther* **8**, 357 (2023).
30. Lakshmana, M.K. SARS-CoV-2-induced autophagy dysregulation may cause neuronal dysfunction in COVID-19. *Neural Regen Res* **17**, 1255-1256 (2022).
31. Crowley, L.C. *et al.* Measuring Cell Death by Propidium Iodide Uptake and Flow Cytometry. *Cold Spring Harb Protoc* **2016** (2016).
32. Kramer, C.E., Wiechert, W. & Kohlheyer, D. Time-resolved, single-cell analysis of induced and programmed cell death via non-invasive propidium iodide and counterstain perfusion. *Sci Rep* **6**, 32104 (2016).
33. Minkoff, J.M. & tenOever, B. Innate immune evasion strategies of SARS-CoV-2. *Nat Rev Microbiol* **21**, 178-194 (2023).
34. Ssali, I. *et al.* Spike protein is a key target for stronger and more persistent T-cell responses—a study of mild and asymptomatic SARS-CoV-2 infection. *Int J Infect Dis* **136**, 49-56 (2023).
35. Liu, X. *et al.* SARS-CoV-2 spike protein-induced cell fusion activates the cGAS-STING pathway and the interferon response. *Sci Signal* **15**, eabg8744 (2022).

36. Blume, C. *et al.* A novel ACE2 isoform is expressed in human respiratory epithelia and is upregulated in response to interferons and RNA respiratory virus infection. *Nat Genet* **53**, 205-214 (2021).
37. Barrat, F.J., Crow, M.K. & Ivashkiv, L.B. Interferon target-gene expression and epigenomic signatures in health and disease. *Nat Immunol* **20**, 1574-1583 (2019).
38. Kang, K. *et al.* Interferon-gamma Represses M2 Gene Expression in Human Macrophages by Disassembling Enhancers Bound by the Transcription Factor MAF. *Immunity* **47**, 235-250 e234 (2017).
39. Zhang, C. *et al.* Targeting lysophospholipid acid receptor 1 and ROCK kinases promotes antiviral innate immunity. *Sci Adv* **7**, eabb5933 (2021).
40. Lamers, M.M. *et al.* SARS-CoV-2 productively infects human gut enterocytes. *Science* **369**, 50-54 (2020).
41. Zhou, J. *et al.* Infection of bat and human intestinal organoids by SARS-CoV-2. *Nat Med* **26**, 1077-1083 (2020).
42. Hashimi, M. *et al.* Antiviral responses in a Jamaican fruit bat intestinal organoid model of SARS-CoV-2 infection. *Nature communications* **14**, 6882 (2023).
43. Dutta, D., Heo, I. & Clevers, H. Disease Modeling in Stem Cell-Derived 3D Organoid Systems. *Trends Mol Med* **23**, 393-410 (2017).

Dear Yongfeng,

Thank you for submitting a revised version of your manuscript. I sincerely apologise for the protracted assessment process due to delays in referee report submission and the high manuscript submission rate to our office at the moment.

We have now received input from two of the original reviewers, who now find that their previous concerns have been addressed satisfactorily and broadly recommend acceptance of the manuscript. Therefore, there now remain only a few mainly editorial points that need addressing before I can extend official acceptance of the manuscript:

1. Please address textually the remaining points by reviewer #1.
2. We require an institutional email address for the co-corresponding author Jie Lei.
3. We are missing the ORCID iD for the co-corresponding authors Shi Jiao and Jie Lei. In order to link the ORCID iD to the account in our manuscript tracking system, the author in question has to do the following:
 - Click the 'Modify Profile' link at the bottom of your homepage in our system.
 - On the next page you will see a box halfway down the page titled ORCID*. Below this box is red text reading 'To Register/Link to ORCID, click here'. Please follow that link: you will be taken to ORCID where you can log in to your account (or create an account if you don't have one)
 - You will then be asked to authorise Wiley to access your ORCID information. Once you have approved the linking, you will be brought back to our manuscript system.Unfortunately, we cannot do this linking on the author's behalf for security reasons.
4. Please remove figures from the manuscript text file.
5. CRediT has replaced the traditional author contributions section because it offers a systematic, machine-readable author contributions format that allows for more effective research assessment. Please remove the Authors Contributions from the manuscript and use the free text boxes beneath each contributing author's name in our online submission system to add specific details on the author's contribution. More information is available in our guide to authors.
6. Please rename "Competing interests statement" section into "Disclosure and competing interests statement" (further info: <https://www.embopress.org/page/journal/14602075/authorguide#conflictsofinterest>).
7. In the "Data availability" section, please include the standard sentence "This study includes no data deposited in external repositories". Further information can be found at <https://www.embopress.org/page/journal/14602075/authorguide#dataavailability>
8. During our routine textual plagiarism check, we noted that a sentence in the manuscript that shows high similarity to that from another publication - please see the attached screenshot. Please rephrase the text accordingly.
9. Our data editors have flagged the following issues in figure legends that need correcting:
 - Please note that the figures EV 1k-l are missing in the manuscript, however the legends for the same is provided. This needs to be corrected.
 - Please provide the exact p in the legends of figures 1a-m, o, q-s; 2a-e; 3a-e, g-j, l-n, p-q; 4a-g; 5d; 7a-c, e-f, h-r; 8b-g; EV 1a-j; EV 2a-b, e-l, n-o; EV 3c, f; EV 4a-c, e-f.
 - Please add information on the number and nature of replicates (technical or biological) in the legends of figures 4a-c; 8g; EV 3c; EV 4e.
 - Please describe the nature of replicates (e.g., biological and technical) in the legends of figures 1a-e, i-p; 2a-i; 3a-e, p-q; 4d-g; 5d; 7a-f, h-k, q-r; 8b-f; EV 1a-e; EV 2a-h, o; EV 3a, f; EV 4a-d, f.
10. Please provide a synopsis image with the dimensions of 550x300-600 pixels (width x height, jpeg or png format). You can either show a model or key data in the synopsis image. Please note that the image size is rather small and that the text needs to be readable at the final size.

With best wishes,

Ieva

Ieva Gailite, PhD
Senior Scientific Editor
The EMBO Journal
Meyerhofstrasse 1
D-69117 Heidelberg
Tel: +4962218891309

We realize that it is difficult to revise to a specific deadline. In the interest of protecting the conceptual advance provided by the work, we recommend a revision within 3 months (19th Dec 2024). Please discuss the revision progress ahead of this time with the editor if you require more time to complete the revisions.

Referee #1:

The authors have addressed most of my concerns and the manuscript has been significantly improved. A few minor points are remaining.

1. The assays have been optimized to better reflect the impact of LSR on SARS-CoV-2 infection. There appears around a 10-fold increase or decrease in SARS-CoV-2 replication after LSR KD or OE, respectively. In some systems the impact is modest e.g. Figure EV2G - line 202 refers to this decrease as virus "restriction" however the language should be softened as there remains considerable replication.

2. Figure 8G could still be larger - it is difficult to see the proteins and labels on the cell surfaces.

3. The addition of the A549 model is interesting. However, the authors state that LSR is highly expressed in lung tissue, however KO did not impact viral replication in the lung of the mouse model. What could be the explanation for this?

Referee #2:

The authors have made extensive edits to the text and added several pieces of new experimental data to address the potential concerns. All of the points raised were satisfactorily addressed.

Response to Referee #1:

The authors have addressed most of my concerns and the manuscript has been significantly improved. A few minor points are remaining.

Response: We thank the reviewer for the kind words and positive feedback.

1. The assays have been optimized to better reflect the impact of LSR on SARS-CoV-2 infection. There appears around a 10-fold increase or decrease in SARS-CoV-2 replication after LSR KD or OE, respectively. In some systems the impact is modest e.g. Figure EV2G - line 202 refers to this decrease as virus "restriction" however the language should be softened as there remains considerable replication.

Response: We appreciate the reviewer's suggestion and have now reworded our statement to ensure it reflects the modest impact of LSR on SARS-CoV-2 infection (line 53, line 206, line 460, and line 1453).

2. Figure 8G could still be larger - it is difficult to see the proteins and labels on the cell surfaces.

Response: We appreciate the reviewer's suggestion and have increased the size of Figure 8G to enhance the visibility of proteins and labels on the cell surfaces.

3. The addition of the A549 model is interesting. However, the authors state that LSR is highly expressed in lung tissue, however KO did not impact viral replication in the lung of the mouse model. What could be the explanation for this?

Response: We apologize for any confusion. The animal model was developed using intestine-specific knockout mice, and we observed that the expression of LSR in the lungs of these mice remained unchanged. Consequently, we did not detect any alterations in viral replication in the lung

of the mouse model. We appreciate the reviewer's insights regarding the A549 model, which are indeed intriguing. We agree with the reviewer that further research is essential to explore the potential impact of LSR knockout in the lungs on viral replication and its related functions. In fact, we are currently conducting extensive studies in our laboratory to address this important question. We already established the AT2 cell-specific *Lsr*-deficient mice expressing the human ACE2 by crossing *Lsr*^{flox/flox}/*Sftpc*-Cre mice with human ACE2 knockin mice to evaluate the involvement of LSR in SARS-CoV-2 infection of lung epithelial cells.

Response to Referee #2:

The authors have made extensive edits to the text and added several pieces of new experimental data to address the potential concerns. All of the points raised were satisfactorily addressed.

Response: Thank you for your positive feedback. We appreciate the time the reviewer invested in our work.

Dear Dr. Gong,

Thank you for implementing the final editorial requests in your revised manuscript. I am now pleased to inform you that your manuscript has been accepted for publication.

Before we forward your manuscript to our publishers, I would like to propose some minor edits in the manuscript title, abstract and synopsis (please see below and the attached manuscript text file). I have also written a short blurb that will accompany the title of your manuscript in our online system. Please let me know if any corrections or adjustments are needed.

Title:

Tight junction protein LSR is a host defense factor against SARS-CoV-2 infection in the small intestine

Blurb:

Lipolysis-stimulated lipoprotein receptor (LSR) and its peptide derivative inhibit SARS-CoV-2 entry into host cells by blocking interaction between viral S protein and its ACE2 receptor.

Synopsis:

Host cell factors that affect the interaction of SARS-CoV-2 Spike (S) protein with its receptor ACE2 can modulate SARS-CoV-2 infection. This study shows that lipolysis-stimulated lipoprotein LSR receptor (LSR) acts as a defense factor against SARS-CoV-2 by impairing S protein binding to ACE2.

- LSR suppresses SARS-CoV-2 infection in the small intestine.
- LSR blocks viral entry and restricts S protein-mediated cell-cell fusion.
- LSR interacts with ACE2 both in cis and in trans, preventing its binding to Spike protein.

Finally, we would like to promote your manuscript among the Chinese readership. Therefore, we would like to invite you to prepare a short summary of the manuscript in Chinese (1500-2000 Chinese characters), which we will promote on the WeChat platform 'BioArt' with more than 610,000 followers.

If you are interested in this opportunity, we recommend covering the article very close to its online publication date. Thus, ideally we would very much appreciate if you could send us a draft within the next 7 working days. Please let us know whether or not you would be interested in contributing such a short summary in Chinese.

I have included below some general guidelines on how to prepare a summary and a link to recent examples for your reference. Please let me know if you have any questions about this.

If you have any questions, please do not hesitate to contact the Editorial Office. Thank you for this contribution to The EMBO Journal and congratulations on a nice study!

Best wishes,

Ieva

Ieva Gailite, PhD
Senior Scientific Editor
The EMBO Journal
Meyerhofstrasse 1
D-69117 Heidelberg
Tel: +4962218891309

General WeChat Summary Guidelines

1. These summary articles are meant to be targeting general audience so please limit the use of specialized technical terms, acronyms and jargon.
2. A summary usually starts with brief background information of the reported work, which is followed by explaining the findings in some detail, and ends with a short review of the conclusions as well as the implications of the work and future directions for the research.
3. The summary should at least contain one graphical item, such as a scheme or a figure from the paper.
4. Please provide ONE SINGLE document containing all text and graphical materials, ideally as a Word.docx or .doc file. Please DO NOT provide the document as a .pdf file.
5. Please DO NOT publicly release the document before the paper is officially published online.

Summary Examples

EMBO J | 罗招庆/欧阳松应揭示谷酰胺脱氨酶MvcA的去泛素化功能

EMBO J | 王松灵院士团队揭示组织内应力调控大型哺乳动物乳恒牙替换的新机制
